# Towards Revealing the Mystery behind Chain of Thought: A Theoretical Perspective

**Guhao Feng**[1,5,*]    **Bohang Zhang**[2,*]    **Yuntian Gu**[3,*]    **Haotian Ye**[3,*]
**Di He**[2,✉]    **Liwei Wang**[2,4,✉]

[1]School of EECS, Peking University    [2]National Key Laboratory of General Artificial Intelligence,
School of Intelligence Science and Technology, Peking University    [3]Yuanpei College,
Peking University    [4]Center for Machine Learning Research, Peking University    [5]Pazhou Lab
fenguhao@stu.pku.edu.cn    zhangbohang@pku.edu.cn    guyuntian@stu.pku.edu.cn
haotianye@pku.edu.cn        dihe@pku.edu.cn        wanglw@pku.edu.cn

## Abstract

Recent studies have discovered that Chain-of-Thought prompting (CoT) can dramatically improve the performance of Large Language Models (LLMs), particularly when dealing with complex tasks involving mathematics or reasoning. Despite the enormous empirical success, the underlying mechanisms behind CoT and how it unlocks the potential of LLMs remain elusive. In this paper, we take a first step towards theoretically answering these questions. Specifically, we examine the *expressivity* of LLMs with CoT in solving fundamental mathematical and decision-making problems. By using circuit complexity theory, we first give impossibility results showing that bounded-depth Transformers are unable to directly produce correct answers for basic arithmetic/equation tasks unless the model size grows *super-polynomially* with respect to the input length. In contrast, we then prove by construction that autoregressive Transformers of *constant size* suffice to solve both tasks by generating CoT derivations using a commonly used math language format. Moreover, we show LLMs with CoT can handle a general class of decision-making problems known as Dynamic Programming, thus justifying their power in tackling complex real-world tasks. Finally, an extensive set of experiments show that, while Transformers always fail to directly predict the answers, they can consistently learn to generate correct solutions step-by-step given sufficient CoT demonstrations.

## 1 Introduction

Transformer-based Large Language Models (LLMs) have emerged as a foundation model in natural language processing. Among them, the autoregressive paradigm has gained arguably the most popularity [51, 9, 46, 69, 57, 16, 52, 54], based on the philosophy that all different tasks can be uniformly treated as sequence generation problems. Specifically, given any task, the input along with the task description can be together encoded as a sequence of tokens (called the *prompt*); the answer is then generated by predicting subsequent tokens conditioned on the prompt in an autoregressive way.

Previous studies highlighted that a carefully designed prompt greatly matters LLMs' performance [32, 38]. In particular, the so-called *Chain-of-Thought* prompting (CoT) [61] has been found crucial for tasks involving arithmetic or reasoning, where the correctness of generated answers can be dramatically improved via a modified prompt that triggers LLMs to output intermediate derivations. Practically, this can be achieved by either adding special phrases such as "*let's think step by step*" or by giving few-shot CoT demonstrations [34, 61, 56, 44, 70, 63]. However, despite the striking performance, the underlying mechanism behind CoT remains largely unclear and mysterious. On one hand, are there

---

*Equal contributions.

37th Conference on Neural Information Processing Systems (NeurIPS 2023).

indeed *inherent* limitations of LLMs in directly answering math/reasoning questions? On the other hand, what is the essential reason behind the success of CoT[2] in boosting the performance of LLMs?

This paper takes a step towards theoretically answering the above questions. We begin with studying the capability of LLMs on two basic mathematical tasks: evaluating arithmetic expressions and solving linear equations. Both tasks are extensively employed and serve as elementary building blocks in solving complex real-world math problems [10]. We first provide fundamental impossibility results, showing that none of these tasks can be solved using bounded-depth Transformer models without CoT unless the model size grows super-polynomially with respect to the input length (Theorems 3.1 and 3.2). Remarkably, our proofs provide insights into why this happens: the reason is not due to the (serialized) computational cost of these problems but rather to their *parallel complexity* [2]. We next show that the community may largely undervalue the strength of autoregressive generation: we prove by construction that autoregressive Transformers of *constant* size can already perfectly solve both tasks by generating intermediate derivations in a step-by-step manner using a commonly-used math language format (Theorems 3.3 and 3.4). Intuitively, this result hinges on the recursive nature of CoT, which increases the "effective depth" of the Transformer to be proportional to the generation steps.

Besides mathematics, CoT also exhibits remarkable performance across a wide range of reasoning tasks. To gain a systematic understanding of why CoT is beneficial, we next turn to a fundamental class of problems known as *Dynamic Programming* (DP) [6]. DP represents a golden framework for solving sequential decision-making tasks: it decomposes a complex problem into a sequence (or chain) of subproblems, and by following the reasoning chain step by step, each subproblem can be solved based on the results of previous subproblems. Our main finding demonstrates that, for general DP problems of the form (5), LLMs with CoT can generate the complete chain and output the correct answer (Theorem 4.7). However, it is impossible to directly generate the answer in general: as a counterexample, we prove that bounded-depth Transformers of polynomial size cannot solve a classic DP problem known as Context-Free Grammar Membership Testing (Theorem 4.8).

Our theoretical findings are complemented by an extensive set of experiments. We consider the two aforementioned math tasks plus two celebrated DP problems listed in the "Introduction to Algorithms" book [17], known as *longest increasing subsequence* (LIS) and *edit distance* (ED). For all these tasks, our experimental results show that directly predicting the answers without CoT always fails (accuracy mostly below 60%). In contrast, autoregressive Transformers equipped with CoT can learn entire solutions given sufficient training demonstrations. Moreover, they even generalize well to longer input sequences, suggesting that the models have learned the underlying reasoning process rather than statistically memorizing input-output distributions. These results verify our theory and reveal the strength of autoregressive LLMs and the importance of CoT in practical scenarios.

## 2 Preliminary

An (autoregressive) Transformer [58, 50] is a neural network architecture designed to process a sequence of input tokens and generate tokens for subsequent positions. Given an input sequence $\boldsymbol{s}$ of length $n$, a Transformer operates the sequence as follows. First, each input token $s_i$ ($i \in [n]$) is converted to a $d$-dimensional vector $\boldsymbol{v}_i = \text{Embed}(s_i) \in \mathbb{R}^d$ using an embedding layer. To identify the sequence order, there is also a positional embedding $\boldsymbol{p}_i \in \mathbb{R}^d$ applied to token $s_i$. The embedded input can be compactly written into a matrix $\boldsymbol{X}^{(0)} = [\boldsymbol{v}_1 + \boldsymbol{p}_1, \cdots, \boldsymbol{v}_n + \boldsymbol{p}_n]^\top \in \mathbb{R}^{n \times d}$. Then, $L$ Transformer blocks follow, each of which transforms the input based on the formula below:

$$\boldsymbol{X}^{(l)} = \boldsymbol{X}^{(l-1)} + \text{Attn}^{(l)}(\boldsymbol{X}^{(l-1)}) + \text{FFN}^{(l)}\left(\boldsymbol{X}^{(l-1)} + \text{Attn}^{(l)}(\boldsymbol{X}^{(l-1)})\right), \quad l \in [L], \quad (1)$$

where $\text{Attn}^{(l)}$ and $\text{FFN}^{(l)}$ denote the multi-head self-attention layer and the feed-forward network for the $l$-th Transformer block, respectively:

$$\text{Attn}^{(l)}(\boldsymbol{X}) = \sum_{h=1}^{H} \text{softmax}\left(\boldsymbol{X}\boldsymbol{W}_Q^{(l,h)}(\boldsymbol{X}\boldsymbol{W}_K^{(l,h)})^\top + \boldsymbol{M}\right)\boldsymbol{X}\boldsymbol{W}_V^{(l,h)}\boldsymbol{W}_O^{(l,h)}, \quad (2)$$

$$\text{FFN}^{(l)}(\boldsymbol{X}) = \sigma(\boldsymbol{X}\boldsymbol{W}_1^{(l)})\boldsymbol{W}_2^{(l)}. \quad (3)$$

---

[2]Throughout this paper, we use the term CoT to refer to the general framework of the step-by-step generation process rather than a specific prompting technique. In other words, this paper studies why an LLM equipped with CoT can succeed in math/reasoning tasks rather than which prompt can trigger this process.

Here, we focus on the standard setting adopted in Vaswani et al. [58], namely, an $H$-head softmax attention followed by a two-layer pointwise FFN, both with residual connections. The size of the Transformer is determined by three key quantities: its depth $L$, width $d$, and the number of heads $H$. The parameters $\boldsymbol{W}_Q^{(l,h)}, \boldsymbol{W}_K^{(l,h)}, \boldsymbol{W}_V^{(l,h)}, \boldsymbol{W}_O^{(l,h)}$ are query, key, value, output matrices of the $h$-th head, respectively; and $\boldsymbol{W}_1^{(l)}, \boldsymbol{W}_2^{(l)}$ are two weight matrices in the FFN. The activation $\sigma$ is chosen as GeLU [28], following [51, 21]. The matrix $\boldsymbol{M} \in \{-\infty, 0\}^{n \times n}$ is a causal mask defined as $M_{ij} = -\infty$ iff $i < j$. This ensures that each position $i$ can only attend to preceding positions $j \leq i$ and is the core design for autoregressive generation.

After obtaining $\boldsymbol{X}^{(L)} \in \mathbb{R}^{n \times d}$, its last entry $\boldsymbol{X}_{n,:}^{(L)} \in \mathbb{R}^d$ will be used to predict the next token $s_{n+1}$ (e.g., via a softmax classifier). By concatenating $s_{n+1}$ to the end of the input sequence $\boldsymbol{s}$, the above process can be repeated to generate the subsequent token $s_{n+2}$. The process continues iteratively until a designated End-of-Sentence token is generated, signifying the completion of the process.

**Chain-of-Thought prompting**. Autoregressive Transformers possess the ability to tackle a wide range of tasks by encoding the task description into a partial sentence, with the answer being derived by complementing the subsequent sentence [9]. However, for some challenging tasks involving math or general reasoning, a direct generation often struggles to yield a correct answer. To address this shortcoming, researchers proposed the CoT prompting that induces LLMs to generate intermediate reasoning steps before reaching the answer [61, 34, 56, 44, 70, 11]. In this paper, our primary focus lies in understanding the mechanism behind CoT, while disregarding the aspect of how prompting facilitates its triggering. Specifically, we examine CoT from an *expressivity* perspective: for both mathematical problems and general decision-making tasks studied in Sections 3 and 4, we will investigate whether autoregressive Transformers are expressive for (i) directly generating the answer, and (ii) generating a CoT solution for the tasks.

## 3  CoT is the Key to Solving Mathematical Problems

Previous studies have observed that Transformer-based LLMs exhibit surprising math abilities in various aspects [46, 10]. In this section, we begin to explore this intriguing phenomenon via two well-chosen tasks: arithmetic and equation. We will give concrete evidence that LLMs are capable of solving both tasks when equipped with CoT, while LLMs without CoT are provably incapable.

### 3.1  Problem formulation

**Arithmetic.** The first task focuses on evaluating arithmetic expressions. As shown in Figure 1 (left), the input of this task is a sequence consisting of numbers, addition ($+$), subtraction ($-$), multiplication ($\times$), division ($\div$), and brackets, followed by an equal sign. The goal is to calculate the arithmetic expression and generate the correct result. This task has a natural CoT solution, where each step performs an intermediate computation, gradually reducing one atomic operation at a time while copying down other unrelated items. Figure 1 (left) gives an illustration, and the formal definition of the CoT format is deferred to Appendix B.

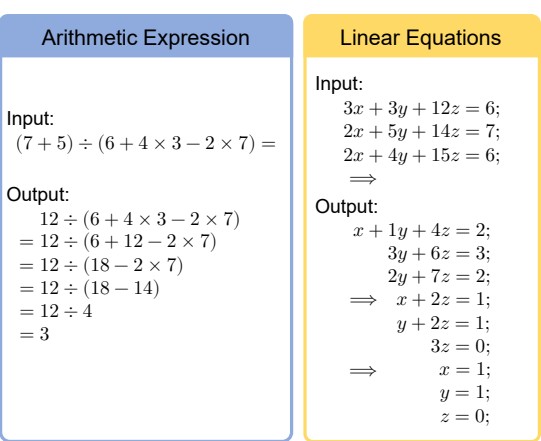

Figure 1: Illustrations of CoT on two math tasks.

**Equation.** The second task considers solving linear equations. As shown in Figure 1 (right), the input of this task is a sequence consisting of $m$ linear equations, each of which involves $m$ variables. The input ends with a special symbol $\Longrightarrow$. The goal is to output the value of these variables that satisfies the set of equations (assuming the answer exists and is unique). A natural CoT solution is the Gaussian elimination algorithm: at each step, it eliminates a certain variable in all but one equations. After $m - 1$ steps, all equations will have only one variable and the problem is solved. Figure 1 (right) gives an illustration, and we defer the formal definition of the CoT format to Appendix B.

**Number field**. Ideally, for both tasks, the input sequences involve not only symbol tokens but also floating-point numbers. This complicates the definitions of the model's input/output format

and further entails intricate precision considerations when dealing with floating-point divisions. To simplify our subsequent analysis, here we turn to a more convenient setting by transitioning to the *finite field* generated by integers modulo $p$ for a prime number $p$. Importantly, the finite field contains only $p$ numbers (ranging from 0 to $p-1$) and thus can be uniformly treated as tokens in a pre-defined dictionary (like other operators or brackets), making the problem setting much cleaner. Moreover, arithmetic operations $(+, -, \times, \div)$ are well-defined and parallel the real number field (see Appendix A.1 for details). Therefore, this setting does not lose generalities.

In subsequent sections, we denote by $\mathsf{Arithmetic}(n, p)$ the arithmetic evaluation task defined on the finite field modulo $p$, where the input length does not exceed $n$. Similarly, we denote by $\mathsf{Equation}(m, p)$ the linear equation task defined on the finite field modulo $p$ with no more than $m$ variables.

## 3.2 Theoretical results

We begin by investigating whether Transformers can directly produce answers to the aforementioned problems. This corresponds to generating, for instance, the number "3" or the solution "$x = 1; y = 1; z = 0$" in Figure 1 immediately after the input sequence (without outputting intermediate steps). This question can be examined via different theoretical perspectives. One natural approach is to employ the classic representation theory, which states that multi-layer perceptrons with sufficient size (e.g., the depth or width approaches infinity) are already universal function approximators [18, 35, 40]. Recently, such results have been well extended to Transformer models [67]: it is not hard to show that a constant-depth Transformer with sufficient size can solve the above tasks[3]. However, the above results become elusive when taking the representation *efficiency* into account, since it says nothing about the required model size for any specific task. Below, we would like to give a more fine-grained analysis of how large the network needs to be by leveraging the tool of complexity theory.

We focus on a *realistic* setting called the **log-precision Transformer** [42, 37]: it refers to a Transformer whose internal neurons can only store floating-point numbers within a finite $O(\log n)$ bit precision where $n$ is the maximal length of the input sequence (see Appendix A.3 for a formal definition). Such an assumption well-resembles practical situations, in which the machine precision (e.g., 16 or 32 bits) is typically much smaller than the input length (e.g., 2048 in GPT), avoiding the unrealistic (but crucial) assumption of infinite precision made in several prior works [49, 20]. Furthermore, log-precision implies that the number of values each neuron can take is *polynomial* in the input length, which is a *necessary* condition for representing important quantities like positional embedding. Equipped with the concept of log-precision, we are ready to present a central impossibility result, showing that the required network size must be prohibitively large for both math problems:

**Theorem 3.1.** *Assume* $\mathsf{TC}^0 \neq \mathsf{NC}^1$. *For any prime number $p$, any integer $L$, and any polynomial $Q$, there exists a problem size $n$ such that no log-precision autoregressive Transformer defined in Section 2 with depth $L$ and hidden dimension $d \leq Q(n)$ can solve the problem* $\mathsf{Arithmetic}(n, p)$.

**Theorem 3.2.** *Assume* $\mathsf{TC}^0 \neq \mathsf{NC}^1$. *For any prime number $p$, any integer $L$, and any polynomial $Q$, there exists a problem size $m$ such that no log-precision autoregressive Transformer defined in Section 2 with depth $L$ and hidden dimension $d \leq Q(m)$ can solve the problem* $\mathsf{Equation}(m, p)$.

**Why does this happen?** As presented in Appendices D.2 and E.2, the crux of our proof lies in applying circuit complexity theory [2]. By framing the finite-precision Transformer as a computation model, one can precisely delineate its expressivity limitation through an analysis of its circuit complexity. Here, bounded-depth log-precision Transformers of polynomial size represent a class of *shallow* circuits with complexity upper bounded by $\mathsf{TC}^0$ [42]. On the other hand, we prove that the complexity of both math problems above are lower bounded by $\mathsf{NC}^1$ by applying *reduction* from $\mathsf{NC}^1$-complete problems. Consequently, they are intrinsically hard to be solved by a well-parallelized Transformer unless the two complexity classes collapse (i.e., $\mathsf{TC}^0 = \mathsf{NC}^1$), a scenario widely regarded as impossible [65].

**How about generating a CoT solution?** We next turn to the setting of generating CoT solutions for these problems. From an expressivity perspective, one might intuitively perceive this problem as more challenging as the model is required to express the entire problem solving process, potentially necessitating a larger model size. However, we show this is not the case: a *constant-size* autoregressive Transformer already suffices to generate solutions for both math problems.

---

[3]For example, given an input sequence of length $n$, an $n$-head self-attention layer with hidden dimension $O(n)$ can extract the information of the entire sequence into the representation of the last position. Then, an MLP operating on the last position can universally approximate any (continuous) function over the input sequence.

**Theorem 3.3.** *Fix any prime $p$. For any integer $n > 0$, there exists an autoregressive Transformer defined in Section 2 with constant hidden size $d$ (independent of $n$), depth $L = 5$, and 5 heads in each layer that can generate the CoT solution defined in Appendix B for all inputs in* Arithmetic$(n, p)$. *Moreover, all parameter values in the Transformer are bounded by $O(\text{poly}(n))$.*

**Theorem 3.4.** *Fix any prime $p$. For any integer $m > 0$, there exists an autoregressive Transformer defined in Section 2 with constant hidden size $d$ (independent of $m$), depth $L = 4$, and 5 heads in each layer that can generate the CoT solution defined in Appendix B for all inputs in* Equation$(m, p)$. *Moreover, all parameter values in the Transformer are bounded by $O(\text{poly}(m))$.*

**Remark 3.5.** The polynomial upper bound for parameters in Theorems 3.3 and 3.4 readily implies that these Transformers can be implemented using log-precision without loss of accuracy. See Appendix A.3 for a detailed discussion on how this can be achieved.

*Proof sketch.* The proofs of Theorems 3.3 and 3.4 are based on construction. We begin by building a set of fundamental operations in Appendix C that can be implemented by Transformer layers. Specifically, the softmax attention head can perform two types of operations called the (conditional) COPY and MEAN (Lemmas C.7 and C.8). Here, conditional COPY extracts the content of the unique previous position that satisfies certain conditions, while Conditional MEAN averages the values of a set of previous positions that satisfy certain conditions. These two operations can be seen as a form of "*gather/scatter*" operator in parallel computing. On the other hand, the FFN in a Transformer layer can perform basic computations within each position, such as multiplication (Lemma C.1), conditional selection (Lemma C.4), and lookup tables (Lemma C.5). With these basic operations as "instructions" and by treating autoregressive generation as a loop, it is possible to write "programs" that can solve fairly complex tasks. As detailed in Appendices D.1 and E.1, we construct parallel algorithms that can generate CoT sequences for both math problems, thus concluding the proof. □

Several discussions are made as follows. *Firstly*, the constructions in our proof reveal the significance of several key components in the Transformer design, such as softmax attention, multi-head, feed-forward networks, and residual connection. Our proofs offer deep insights into the inner workings of Transformer models when dealing with complex tasks, significantly advancing prior understandings such as the "induction head" mechanism [45]. Moreover, our results identify an inherent advantage of Transformers compared to other sequence models like RNNs: indeed, as shown in Appendix F.2, constant-size RNNs *cannot* solve any of the above math tasks using the same CoT format. *Secondly*, we highlight that in our setting, the CoT derivations of both math problems are purely written in a *readable* math language format, largely resembling how humans write solutions. In a broad sense, our findings justify that LLMs have the potential to convey meaningful human thoughts through *grammatically precise* sentences. *Finally*, one may ask how LLMs equipped with CoT can bypass the impossibility results outlined in Theorems 3.1 and 3.2. Actually, this can be understood via the *effective depth* of the Transformer circuit. By employing CoT, the effective depth is no longer $L$ since the generated outputs are repeatedly looped back to the input. The dependency between output tokens leads to a significantly deeper circuit with depth proportional to the length of the CoT solution. Note that even if the recursive procedure is repeated within a fixed Transformer (or circuit), the expressivity can still be far beyond $\mathsf{TC}^0$: as will be shown in Section 4, with a sufficient number of CoT steps, autoregressive Transformers can even solve P-complete problems.

## 4 CoT is the Key to Solving General Decision-Making Problems

The previous section has delineated the critical role of CoT in solving math problems. In this section, we will switch our attention to a more general setting beyond mathematics. Remarkably, we find that LLMs with CoT are theoretically capable of emulating a powerful decision-making framework called *Dynamic Programming* [6], thus strongly justifying the ability of CoT in solving complex tasks.

### 4.1 Dynamic Programming

Dynamic programming (DP) is widely regarded as a core technique to solve decision-making problems [55]. The basic idea of DP lies in breaking down a complex problem into a series of small subproblems that can be tackled in a sequential manner. Here, the decomposition ensures that there is a significant interconnection (overlap) among various subproblems, so that each subproblem can be efficiently solved by utilizing the answers (or other relevant information) obtained from previous ones.

Formally, a general DP algorithm can be characterized via three key ingredients: state space $\mathcal{I}$, transition function $T$, and aggregation function $A$. Given a DP problem with $N$ input sequences $\boldsymbol{s}^{(1)}, \cdots, \boldsymbol{s}^{(N)}$, denote the problem size to be the vector $\boldsymbol{n} = (|\boldsymbol{s}^{(1)}|, \cdots, |\boldsymbol{s}^{(N)}|)$. Fixing the problem size $\boldsymbol{n}$, there is an associated **state space** $\mathcal{I}_{\boldsymbol{n}} \subset \mathcal{I}$ representing the finite set of decomposed subproblems, where each state $i \in \mathcal{I}_{\boldsymbol{n}}$ is an index signifying a specific subproblem. The size of the state space $\mathcal{I}_{\boldsymbol{n}}$ grows with the problem size $\boldsymbol{n}$. We denote by $\mathsf{dp}(i)$ the answer of subproblem $i$ (as well as other information stored in the DP process). Furthermore, there is a *partial order* relation between different states: we say state $j$ precedes state $i$ (denoted as $j \prec i$) if subproblem $j$ should be solved before subproblem $i$, i.e., the value of $\mathsf{dp}(i)$ depends on $\mathsf{dp}(j)$. This partial order creates a directed acyclic graph (DAG) within the state space, thereby establishing a reasoning chain where subproblems are resolved in accordance with the topological ordering of the DAG.

The **transition function** $T$ characterizes the interconnection among subproblems and defines how a subproblem can be solved based on the results of previous subproblems. It can be generally written as

$$\mathsf{dp}(i) = T\left(\boldsymbol{n}, \boldsymbol{s}, i, \{(j, \mathsf{dp}(j)) : j \prec i\}\right), \tag{4}$$

where $\boldsymbol{s}$ is the concatenation of all input sequences $\boldsymbol{s}^{(1)}, \cdots, \boldsymbol{s}^{(N)}$. In this paper, we focus on a restricted setting where each state $i$ only depends on (i) a finite number of tokens in the input sequence $\boldsymbol{s}$ and (ii) a finite number of previous states. Under this assumption, we can rewrite (4) into a more concrete form:

$$\begin{aligned}
\mathsf{dp}(i) &= f\left(\boldsymbol{n}, i, s_{\boldsymbol{g}(\boldsymbol{n},i)}, \mathsf{dp}(\boldsymbol{h}(\boldsymbol{n}, i))\right) \\
&= f\left(\boldsymbol{n}, i, s_{g_1(\boldsymbol{n},i)}, \cdots, s_{g_J(\boldsymbol{n},i)}, \mathsf{dp}(h_1(\boldsymbol{n}, i)), \cdots, \mathsf{dp}(h_K(\boldsymbol{n}, i))\right),
\end{aligned} \tag{5}$$

where functions $f, \boldsymbol{g}, \boldsymbol{h}$ fully determine the transition function $T$ and have the following form $f : \mathbb{N}^N \times \mathcal{I} \times \mathcal{X}^J \times \mathcal{Y}^K \to \mathcal{Y}, \boldsymbol{g} : \mathbb{N}^N \times \mathcal{I} \to (\mathbb{N} \cup \{\emptyset\})^J, \boldsymbol{h} : \mathbb{N}^N \times \mathcal{I} \to (\mathcal{I} \cup \{\emptyset\})^K$. Here, the state space $\mathcal{I}$, input space $\mathcal{X}$, and DP output space $\mathcal{Y}$ can be arbitrary domains, and $J, K$ are constant integers. If state $i$ depends on less than $J$ input tokens or less than $K$ previous states, we use the special symbol $\emptyset$ to denote a placeholder, such that all terms $s_\emptyset$ and $\mathsf{dp}(\emptyset)$ are unused in function $f$.

After solving all subproblems, the **aggregation function** $A$ is used to combine all results and obtain the final answer. We consider a general class of aggregation functions with the following form:

$$A\left(\{(i, \mathsf{dp}(i)) : i \in \mathcal{I}_{\boldsymbol{n}}\}\right) = u\left(\square_{i \in \mathcal{A}_{\boldsymbol{n}}} \mathsf{dp}(i)\right), \tag{6}$$

where $\mathcal{A}_{\boldsymbol{n}} \subset \mathcal{I}_{\boldsymbol{n}}$ is a set of states that need to be aggregated, $\square$ is an aggregation function such as $\min, \max$, or $\sum$, and $u : \mathcal{Y} \to \mathcal{Z}$ is any function where $\mathcal{Z}$ denotes the space of possible answers.

A variety of popular DP problems fits the above framework. As examples, the longest increasing subsequence (LIS) and edit distance (ED) are two well-known DP problems presented in the "Introduction to Algorithms" book [17] (see Appendix G.1 for problem descriptions and DP solutions). We list the state space, transition function, and aggregation function of the two problems in the table below.

| Problem | Longest increasing subsequence | Edit distance |
|---|---|---|
| Input | A string $\boldsymbol{s}$ of length $n$ | Two strings $\boldsymbol{s}^{(1)}, \boldsymbol{s}^{(2)}$ of length $n_1 = |\boldsymbol{s}^{(1)}|$ and $n_2 = |\boldsymbol{s}^{(2)}|$, concatenated together |
| State space | $\{(j, k) : j \in [n], k \in \{0, \cdots, j-1\}\}$ | $\{0, \cdots, n_1\} \times \{0, \cdots, n_2\}$ |
| Transition function | $\mathsf{dp}(j, k) = \begin{cases} 1 & \text{if } k = 0 \\ \max(\mathsf{dp}(j, k-1), & \\ \qquad \mathsf{dp}(k, k-1) \times & \text{if } k > 0 \\ \qquad \mathbb{I}[s_j > s_k] + 1) & \end{cases}$ | $\mathsf{dp}(j, k) = \begin{cases} ak & \text{if } j = 0 \\ bj & \text{if } k = 0 \\ \min(\mathsf{dp}(j, k-1) + a, & \\ \qquad \mathsf{dp}(j-1, k) + b, & \\ \qquad \mathsf{dp}(j-1, k-1) & \text{otherwise} \\ \qquad + c\mathbb{I}[s_j^{(1)} \neq s_k^{(2)}]) & \end{cases}$ |
| Aggregation function | $\max_{i \in [n]} \mathsf{dp}(i, i-1)$ | $\mathsf{dp}(n_1, n_2)$ |

## 4.2 Theoretical results

We begin by investigating whether LLMs with CoT can solve the general DP problems defined above. We consider a natural CoT generation process, where the generated sequence has the following form:

$$\boldsymbol{s}^{(1)} \quad | \quad \cdots \quad | \quad \boldsymbol{s}^{(N)} \quad | \quad (i_1, \mathsf{dp}(i_1)) \quad \ldots \quad (i_{|\mathcal{I}_{\boldsymbol{n}}|}, \mathsf{dp}(i_{|\mathcal{I}_{\boldsymbol{n}}|})) \quad \text{final answer}$$

Here, the input sequence consists of $N$ strings separated by special symbols, and $(i_1, \cdots, i_{|\mathcal{I}_n|})$ is a feasible topological ordering of the state space $\mathcal{I}_n$. We assume that all domains $\mathcal{I}, \mathcal{X}, \mathcal{Y}, \mathcal{Z}$ belong to the real vector space so that their elements can be effectively represented and handled by a neural network. Each $(i, \mathsf{dp}(i)) \in \mathcal{I} \times \mathcal{Y}$ above will be represented as a *single* vector and generated jointly in the CoT output. We further assume that $\mathcal{I}, \mathcal{X}, \mathcal{Y}, \mathcal{Z}$ are *discrete* spaces (e.g., integers) so that the elements can be precisely represented using finite precision. To simplify our analysis, we consider a *regression* setting where each element in the CoT output directly corresponds to the output of the last Transformer layer (without using a softmax layer for tokenization as in Section 3). Instead, the Transformer output is simply projected to the nearest element in the corresponding discrete space (e.g., $\mathcal{I} \times \mathcal{Y}$ or $\mathcal{Z}$). Likewise, each generated output is directly looped back to the Transformer input without using an embedding layer. This regression setting is convenient for manipulating numerical values and has been extensively adopted in prior works [24, 1].

Before presenting our main result, we make the following assumptions:

**Definition 4.1** (Polynomially-efficient approximation). Given neural network $P_\theta$ and target function $f : \mathcal{X}^{\text{in}} \to \mathcal{X}^{\text{out}}$ where $\mathcal{X}^{\text{in}} \subset \mathbb{R}^{d_{\text{in}}}$ and $\mathcal{X}^{\text{out}} \subset \mathbb{R}^{d_{\text{out}}}$, we say $f$ can be approximated by $P_\theta$ with polynomial efficiency if there exist $\rho > 0$, $\lambda > 0$ such that for any error $\epsilon > 0$ and radius $R > 0$, there exists parameter $\theta$ satisfying that (i) $\|f(x) - P_\theta(x + \delta)\|_\infty < \epsilon + \lambda \|\delta\|_\infty$ for all $x \in \mathcal{X}^{\text{in}}$, $\|x\|_\infty \leq R$ and all $\|\delta\|_\infty < \rho$; (ii) all elements of parameter $\theta$ are bounded by $O(\text{poly}(R, 1/\epsilon))$.

**Assumption 4.2.** The size of the state space can be polynomially upper bounded by the problem size $n$, i.e., $|\mathcal{I}_n| = O(\text{poly}(|s|))$. Similarly, all input elements, DP values, and answers are polynomially upper bounded by the problem size $n$.

**Assumption 4.3.** Each function $f$, $g$, $h$ and $u$ in (5) and (6) can be approximated with polynomial efficiency by a perceptron of constant size (with GeLU activation).

**Assumption 4.4.** The function $F : \mathbb{N}^N \times \mathcal{I} \to \mathcal{I}$ defined as $F(n, i_k) = i_{k+1}$ for $n \in \mathbb{N}^N$, $k \in [|\mathcal{I}_n| - 1]$ can be approximated with polynomial efficiency by a perceptron of constant size (with GeLU activation), where $(i_1, \cdots, i_{|\mathcal{I}_n|})$ is a feasible topological ordering of the state space $\mathcal{I}_n$.

**Assumption 4.5.** The function $F : \mathbb{N}^N \times \mathcal{I} \to \{0, 1\}$ defined as $F(n, i) = \mathbb{I}[i \in \mathcal{A}_n]$ (see (6)) can be approximated with polynomial efficiency by a perceptron of constant size (with GeLU activation).

**Remark 4.6.** All assumptions above are mild. Assumption 4.2 is necessary to ensure that the state vectors, inputs, and DP values can be represented using log-precision, and Assumptions 4.3 to 4.5 guarantee that all basic functions that determine the DP process can be well-approximated by a composition of finite log-precision Transformer layers of constant size. In Appendix G.1, we show these assumptions are satisfied for LIS and ED problems described above as well as the CFG Membership Testing problem in Theorem 4.8.

We are now ready to present our main result, which shows that LLMs with CoT can solve all DP problems satisfying the above assumptions. We give a proof in Appendix G.2.

**Theorem 4.7.** *Consider any DP problem satisfying Assumptions 4.2 to 4.5. For any integer $n \in \mathbb{N}$, there exists an autoregressive Transformer with constant depth $L$, hidden dimension $d$ and attention heads $H$ (independent of $n$), such that the answer generated by the Transformer is correct for all input sequences of length no more than $n$. Moreover, all parameter values are bounded by $O(\text{poly}(n))$.*

To complete the analysis, we next explore whether Transformers can directly predict the answer of DP problems without generating intermediate CoT sequences. We show generally the answer is no: many DP problems are intrinsically hard to be solved by a bounded-depth Transformer without CoT. One celebrated example is the Context-Free Grammar (CFG) Membership Testing, which take a CFG $G$ and a string $v$ as input and tests whether $v$ belongs to the context-free language generated by $G$. A formal definition of this problem and a standard DP solution are given in Appendix G.1. We have the following impossibility result (see Appendix G.3 for a proof):

**Theorem 4.8.** *Assume $\mathsf{TC}^0 \neq \mathsf{NC}^1$. There exists a context-free language such that for any depth $L$ and any polynomial $Q$, there exists a sequence length $n \in \mathbb{N}$ where no log-precision autoregressive transformer with depth $L$ and hidden dimension $d \leq Q(n)$ can generate the correct answer for the CFG Membership Testing problem for all input strings of length $n$.*

In contrast, enabling CoT substantially improves the expressivity of Transformers: as proved in Jones & Laaser [33], the universal CFG Membership Testing is a celebrated P-complete problem and is intrinsically hard to be solved by a well-parallelized computation model. Combined with these results, we conclude that CoT plays a critical role in tackling tasks that are inherently difficult.

# 5 Experiments

In previous sections, we proved by construction that LLMs exhibit sufficient expressive power to solve mathematical and decision-making tasks. On the other hand, it is still essential to check whether a Transformer model can *learn* such ability directly from training data. Below, we will complement our theoretical results with experimental evidence, showing that the model can easily learn underlying task solutions when equipped with CoT training demonstrations.

## 5.1 Experimental Design

**Tasks and datasets.** We choose four tasks for evaluation: Arithmetic, Equation, LIS, and ED. The first two tasks (Arithmetic and Equation) as well as their input/CoT formats have been illustrated in Figure 1. For the LIS task, the goal is to find the length of the longest increasing subsequence of a given integer sequence. For the ED task, the goal is to calculate the minimum cost required (called edit distance) to convert one sequence to another using three basic edit operations: insert, delete and replace. All input sequences, CoT demonstrations, and answers in LIS and ED are bounded-range integers and can therefore be tokenized (similar to the first two tasks). We consider two settings: (i) CoT datasets, which consist of <problem, CoT steps, answer> samples; (ii) Direct datasets, which are used to train models that directly predict the answer without CoT steps. These datasets are constructed by removing all intermediate derivations from the CoT datasets.

For each task, we construct three datasets with increasing difficulty. For Arithmetic, we build datasets with different numbers of operators ranging from $\{4, 5, 6\}$. For Equation, we build datasets with different numbers of variables ranging from $\{3, 4, 5\}$. For LIS, we build datasets with different input sequence lengths ranging from $\{50, 80, 100\}$. For ED, we build datasets with different string lengths, where the average length of the two strings ranges from $\{12, 16, 20\}$. Creating these datasets allows us to investigate how model performance varies with the increase of problem size. We generate 1M samples for each training dataset and 0.1M for testing while ensuring that duplicate samples between training and testing are removed. More details about the dataset construction can be found in Appendix H.

**Model training and inference.** For all experiments, we use standard Transformer models with hidden dimension $d = 256$, heads $H = 4$, and different model depths $L$. We adopt the AdamW optimizer [39] with $\beta_1 = 0.9, \beta_2 = 0.999, \mathrm{lr} = 10^{-4}$, and weight decay $= 0.01$ in all experiments. We use a fixed dropout ratio of 0.1 for all experiments to improve generalization. For CoT datasets, we optimize the negative log-likelihood loss on all tokens in the CoT steps and answers. This is similar to the so-called process supervision proposed in a concurrent work [36]. For direct datasets, we optimize the negative log-likelihood loss on answer tokens. All models are trained on 4 V100 GPUs for 100 epochs. During inference, models trained on the direct datasets are required to output the answer directly, and models trained on CoT datasets will generate the whole CoT process token-by-token (using greedy search) until generating the `End-of-Sentence` token, where the output in the final step is regarded as the answer. We report the accuracy as the evaluation metric. Please refer to Appendix H for more training configuration details.

## 5.2 Experimental Results

**Main results**. All results are shown in Figure 2, where each subfigure corresponds to a task with x-axis representing the difficulty level and y-axis representing the test accuracy (%). We repeat each experiment five times and report the error bars. In each subfigure, the purple bar and blue bars indicate the performance of the model trained on the CoT and direct datasets, respectively. The model depths are specified in the legend. From these results, one can easily see that 3-layer Transformers with CoT already achieve near-perfect performance on all tasks for all difficulty levels, which is consistent to Theorems 3.3, 3.4 and 4.7 and may further imply that they can learn better solutions with fewer model depth than our constructions. Moreover, it is worth noting that, for some difficult datasets such as Equation of 5 variables, a perfect accuracy would imply that the model can precisely generate a correct CoT with a very long length of roughly 500 for all inputs. In contrast, models trained on direct datasets perform much worse even when using larger depths (particularly on the Equation task). While increasing the depth usually helps the performance of direct prediction (which is consistent with our theory), the performance drops significantly when the length of the input sequence grows. All these empirical findings verify our theoretical results and clearly demonstrate the benefit of CoT in autoregressive generation.

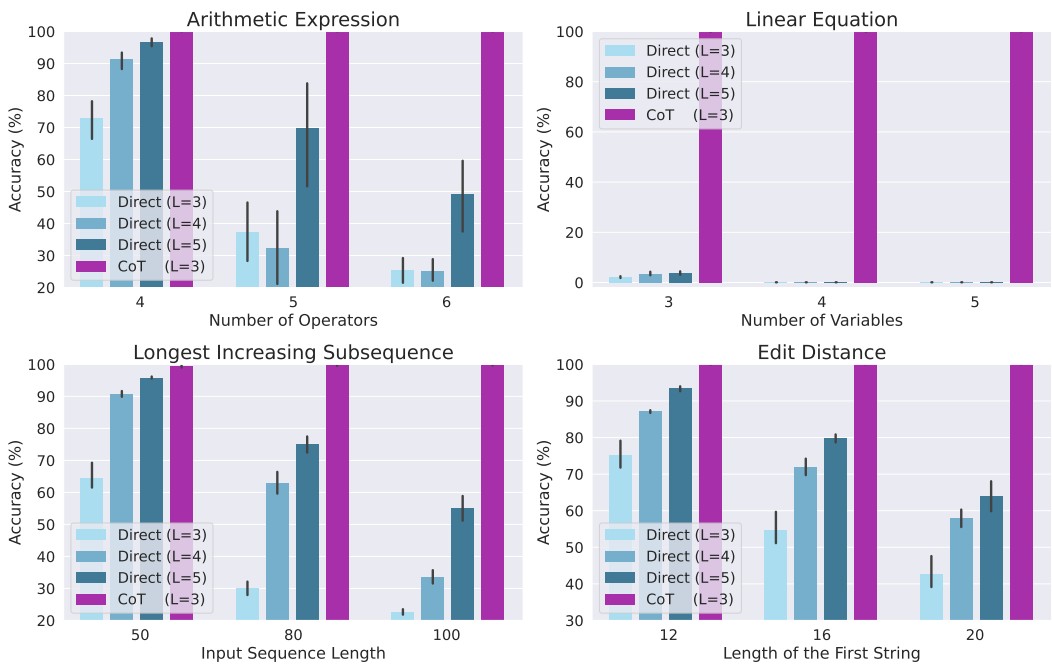

Figure 2: Model performance on different tasks. For all tasks and various difficulty levels, autoregressive Transformers with CoT consistently outperform Transformers trained on direct datasets. In particular, 3-layer Transformers already succeed in these tasks with almost perfect accuracy, while deeper Transformers ($L = 3, 4, 5$) trained on the direct datasets typically fail.

**Robustness to data quality**. Unlike the synthetic datasets constructed above, real-world training datasets are not perfect and often involve corruption or miss intermediate steps. This calls into question whether the model can still perform well when training on low-quality datasets. To investigate this question, we construct corrupted datasets for the arithmetic task in Appendix I. Surprisingly, our results show that the 3-layer Transformer model can still achieve more than 95% accuracy even with 30% of the data missing an intermediate CoT step and involving a single-token corruption. This clearly demonstrates the robustness of CoT training on low-quality datasets.

**Length extrapolation.** We finally study whether the learned autoregressive models can further extrapolate to data with longer lengths. We construct a CoT training dataset for the arithmetic task with the number of operators ranging from 1 to 15, and test the model on expressions with the number of operators in $\{16, 17, 18\}$. As shown in Figure 3, our three-layer Transformer model still performs well on longer sequences, suggesting that the model indeed learns the solution to some extent (instead of memorizing data distributions). Potentially, we believe models trained on more data with varying lengths can eventually reveal the complete arithmetic rules.

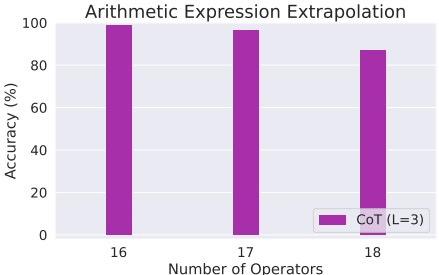

Figure 3: Performance of length extrapolation experiment, tested on sequences that are longer than those in training.

## 6 Related Work

Owing to the tremendous success of Transformers and Large Language Models across diverse domains, there has been a substantial body of works dedicated to theoretically comprehending their capabilities and limitations. Initially, researchers primarily focused on exploring the expressive power of Transformers in the context of function approximation. Yun et al. [67] proved that Transformers with sufficient size can universally approximate arbitrary continuous sequence-to-sequence functions on a compact domain. Recently, universality results have been extended to model variants such as Sparse Transformers [68] and Transformers with relative positional encodings (RPE) [41].

More relevant to this paper, another line of works investigated the power of Transformers from a computation perspective. Early results have shown that both standard encoder-decoder Transformers [58] and looped Transformer encoders are Turing-complete [49, 47, 20, 8]. However, these results depend on the unreasonable assumption of *infinite* precision, yielding a quite unrealistic construction that does not match practical scenarios. Recently, Giannou [25] demonstrated that a constant-depth looped Transformer encoder can simulate practical computer programs. Wei et al. [60] showed that finite-precision encoder-decoder Transformers can *approximately* simulate Turing machines with bounded computation time. Liu et al. [37] considered a restricted setting of learning automata, for which a shallow non-recursive Transformer provably suffices. Yao et al. [66] demonstrated that Transformers can recognize or generate bounded-depth Dyck language [30], a specific type of context-free language. Besides affirmative results, other works characterized the expressivity limitation of Transformers via the perspective of modeling formal languages [26, 7, 62, 66, 14] or simulating circuits [27, 43, 42]. However, none of these works (except [66]) explored the setting of autoregressive Transformers typically adopted in LLMs, which we study in this paper. Moreover, we consider a more practical setting that targets the emergent ability of LLMs in solving basic reasoning problems via a *readable* CoT output, which aligns well with real-world scenarios.

Recently, the power of Transformers has regained attention due to the exceptional in-context learnability exhibited by LLMs [9]. Garg et al. [24] demonstrated that autoregressive Transformers can in-context learn basic function classes (e.g., linear functions, MLPs, and decision trees) via input sample sequences. Subsequent works further revealed that Transformers can implement learning algorithms such as linear regression [1], gradient descent [1, 59, 19], and Bayesian inference [64], and a broad class of machine learning algorithms [3]. The works of [23, 45] studied in-context learning via the concept of "induction heads". All the above works investigated the power of (autoregressive) Transformer models from an expressivity perspective, which shares similarities to this paper. Here, we focus on the reasoning capability of Transformers and underscore the key role of CoT in improving the power of LLMs.

## 7 Limitations and Future Directions

In this work, from a model-capacity perspective, we theoretically analyze why Chain-of-Thought prompting is essential in solving mathematical and decision-making problems. Focusing on two basic mathematical problems as well as Dynamic Programming, we show that a bounded-depth Transformer without CoT struggles with these tasks unless its size grows prohibitively large. In contrast to our negative results, we prove by construction that when equipped with CoT, constant-size Transformers are sufficiently capable of addressing these tasks by generating intermediate derivations sequentially. Extensive experiments show that models trained on CoT datasets can indeed learn solutions almost perfectly, while direct prediction always fails. We further demonstrate that CoT has the potential to generalize to unseen data with longer lengths.

Several foundational questions remain to be answered. Firstly, while this paper investigates why CoT enhances the expressivity of LLMs, we do not yet answer how the CoT generation process is triggered by specific prompts. Revealing the relation between prompts and outputs is valuable for better harnessing LLMs. Secondly, it has been empirically observed that scaling the model size significantly improves the CoT ability [61]. Theoretically understanding how model size plays a role in CoT would be an interesting research problem. Thirdly, this paper mainly studies the expressivity of LLMs in generating CoT solutions, without theoretically thinking about their *generalization* ability. Given our experimental results, we believe it is an important future direction for theoretically studying how LLMs can generalize from CoT demonstrations (even in the out-of-distribution setting, e.g., length extrapolation (Figure 3)) [60, 13]. Finally, from a practical perspective, it is interesting to investigate how models can learn CoT solutions when there are only limited CoT demonstrations in training (or even purely from direct datasets). We would like to leave these questions as future work, which we believe are beneficial to better reveal the power and limitations of LLMs.

**Acknowledgement.** This work is supported by National Key R&D Program of China (2022ZD0114900) and National Science Foundation of China (NSFC62276005), and is partially funded by Microsoft Research Asia Collaborative Research Project. The authors are grateful to David Chiang, who pointed out a mistake regarding the P-completeness of CFG Membership Testing in the early version of this paper. The authors also thank all reviewers for their valuable suggestions.

## Author Contributions

**Guhao Feng** proposed to analyze the expressivity of Transformers using circuit complexity and proved all impossibility results in this paper, including Theorems 3.1, 3.2 and 4.8. He came up with the initial idea of Dynamic Programming. He and Bohang Zhang formalized the DP framework and proved all positive results in this paper, including Theorems 3.3, 3.4 and 4.7. He contributed to the paper writing of Appendices C to G.

**Bohang Zhang** supervised all undergraduate students. He raised the the problem of linear equation. He and Guhao Feng formalized the DP framework and proved all positive results in this paper, including Theorems 3.3, 3.4 and 4.7. In the experimental part, he helped generate datasets with Yuntian Gu, conducted hyper-parameter tuning, and finalized all experiments in Section 5. He was responsible for writing the majority of this paper and checking/correcting all proofs.

**Yuntian Gu** was responsible for the experimental part. He wrote the entire code, including the model details, training pipeline, and evaluation. He created the datasets for all four tasks in Section 5 and conducted extensive experimental exploration during this project (with the help of Bohang Zhang).

**Haotian Ye** participated in regular discussions, raised ideas, and helped check the code in experiments.

**Di He** initiated the problem of studying the capability of Transformers in basic tasks like evaluating arithmetic expressions. **Di He** and **Liwei Wang** led and supervised the research, suggested ideas and experiments, and assisted in writing the paper.

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

# Appendix

The Appendix is organized as follows. Appendix A introduces additional mathematical background and useful notations. Appendix B presents formal definitions and CoT solutions of the arithmetic expression task and the linear equation task. Appendix C gives several technical lemmas, which will be frequently used in our subsequent proofs. The formal proofs for arithmetic expression, linear equation, and dynamic programming tasks are given in Appendices D, E and G, respectively. Discussions on other architectures such as encoder-decoder Transformers and RNNs are presented in Appendix F. Finally, we provide experimental details in Appendix H.

## A    Additional Background and Notation

### A.1    Finite field

Intuitively, a *field* is a set $\mathcal{F}$ on which addition, subtraction, multiplication, and division are defined and behave as the corresponding operations on rational and real numbers do. Formally, the two most basic binary operations in a field is the addition ($+$) and multiplication ($\times$), which satisfy the following properties:

- Associativity: for any $a, b, c \in \mathcal{F}$, $(a + b) + c = a + (b + c)$ and $(a \times b) \times c = a \times (b \times c)$;
- Commutativity: for any $a, b \in \mathcal{F}$, $a + b = b + a$ and $a \times b = b \times a$;
- Identity: there exist two different elements $0, 1 \in \mathcal{F}$ such that $a + 0 = a$ and $a \times 1 = a$ for all $a \in \mathcal{F}$;
- Additive inverses: for any $a \in \mathcal{F}$, there exists an element in $\mathcal{F}$, denoted as $-a$, such that $a + (-a) = 0$;
- Multiplicative inverses: for any $a \in \mathcal{F}$ and $a \neq 0$, there exists an element in $\mathcal{F}$, denoted as $a^{-1}$, such that $a \times a^{-1} = 1$;
- Distributivity of multiplication over addition: for any $a, b, c \in \mathcal{F}$, $a \times (b+c) = (a \times b) + (a \times c)$.

Then, subtraction ($-$) is defined by $a - b = a + (-b)$ for all $a, b \in \mathcal{F}$; division ($\div$) is defined by $a \div b = a \times b^{-1}$ for all $a, b \in \mathcal{F}, b \neq 0$.

Two most widely-used fileds are the rational number field $\mathbb{Q}$ and the real number field $\mathbb{R}$, both of which satisfy the above properties. However, both fields contain an infinite number of elements. In this paper, we consider a class of fields called finite fields, which contain a *finite* number of elements. Given a prime number $p$, the finite field $\mathbb{Z}_p$ is the field consisting of $p$ elements, which can be denoted as $0, 1, \cdots, p - 1$. In $\mathbb{Z}_p$, both addition and multiplication are defined by simply adding/multiplying two input integers and then taking the remainder modulo $p$. It can be easily checked that the two operations satisfy the six properties described above. Thus, subtraction and division can be defined accordingly. Remarkably, a key result in abstract algebra shows that all finite fields with $p$ elements are *isomorphic*, which means that the above definitions of addition, subtraction, multiplication, and division are unique (up to isomorphism).

As an example, consider the finite field $\mathbb{Z}_5$. We have that $2 + 3$ equals $0$, since $(2 + 3) \bmod 5 = 0$. Similarly, $2 \times 3$ equals $1$; $2 - 3$ equals $4$; and $2 \div 3$ equals $4$.

In Section 3, we utilize the field $\mathbb{Z}_p$ to address the issue of infinite tokens. Both tasks of evaluating arithmetic expressions and solving linear equations (Section 3.1) are well-defined in this field.

### A.2    Circuit complexity

In circuit complexity theory, there are several fundamental complexity classes that capture different levels of computation power. Below, we provide a brief overview of these classes; however, for a comprehensive introduction, we recommend readers refer to Arora & Barak [2].

The basic complexity classes we will discuss in this subsection are $\mathsf{NC}^0$, $\mathsf{AC}^0$, $\mathsf{TC}^0$, $\mathsf{NC}^1$, and $\mathsf{P}$. These classes represent increasing levels of computation complexity. The relationships between these

classes can be summarized as follows:

$$NC^0 \subsetneq AC^0 \subsetneq TC^0 \subset NC^1 \subset P$$

Moreover, in the field of computational theory, it is widely conjectured that all subset relations in the hierarchy are *proper* subset relations. This means that each class is believed to capture a strictly larger set of computational problems than its predecessor in the hierarchy. However, proving some of these subset relations to be proper remains a critical open question in computational complexity theory. For example, $NC^1 = P$ will imply $P = NP$, which is widely regarded as impossible but is still a celebrated open question in computer science.

To formally define these classes, we first introduce the concept of Boolean circuits. A Boolean circuit with $n$ input bits is a directed acyclic graph (DAG), in which every node is either an input bit or an internal node representing one bit (also called a gate). The value of each internal node depends on its direct predecessors. Furthermore, several internal nodes are designated as output nodes, representing the output of the Boolean circuit. The in-degree of a node is called its *fan-in* number, and the input nodes have zero fan-in.

A Boolean circuit can only simulate a computation problem of a fixed number of input bits. When the input length varies, a series of distinct Boolean circuits will be required, each designed to process a specific length. In this case, circuit complexity studies how the circuit size (e.g., depth, fan-in number, width) increases with respect to the input length for a given computation problem. We now describe each complexity class as follows:

- $NC^0$ is the class of constant-depth, constant-fan-in, polynomial-sized circuits consisting of AND, OR, and NOT gates. $NC^0$ circuits is the weakest class in the above hierarchy with limited expressive power because they cannot express functions that depend on a growing number of inputs as the input size increases. For example, the basic logical-AND function with an arbitrary number of input bits is not in $NC^0$. In [22], the authors considered a restricted version of the Transformer model with constant depth and a *constant-degree* sparse selection construction, which can be characterized by this complexity class.
- $AC^0$ is the class of constant-depth, unbounded-fan-in, polynomial-sized circuits consisting of AND, OR, and NOT gates, with NOT gates allowed only at the inputs. It is strictly more powerful than $NC^0$ mainly because the fan-in number can (polynomially) depend on the input length. However, there are still several fundamental Boolean functions that are not in this complexity class, such as the parity function or the majority function (see below).
- $TC^0$ is an extension of $AC^0$ that introduces an additional gate called MAJ (i.e., the majority). The MAJ gate takes an arbitrary number of input bits and evaluates to false when half or more of the input bits are false, and true otherwise. Previous work [42, 43] showed that the log-precision Transformer is in this class.
- $NC^1$ is a complexity class that consists of constant-fan-in, polynomial-sized circuits with a logarithmic depth of $O(\log n)$, where $n$ is the input length. Similar to $NC^0$, the basic logical gates are AND, OR, and NOT. Allowing the number of layers to depend on the input length significantly increases the expressiveness of the circuit. On the other hand, the logarithmic dependency still enables a descent parallelizability. Indeed, $NC^1$ is widely recognized as an important complexity class that captures efficiently parallelizable algorithms.
- $P$ is the complexity class that contains problems that can be solved by a Turing machine in polynomial time. It contains a set of problems that do not have an efficient parallel algorithm. For example, the Context-Free-Grammar Membership Testing is in this class and is proved to be P-complete [33].

### A.3 Log-precision

In this work, we focus on Transformers whose neuron values are restricted to be floating-point numbers of finite precision, and all computations operated on floating-point numbers will be finally truncated, similar to how a computer processes real numbers. In practice, the two most common formats to store real numbers are the fixed-point format and floating-point format (e.g., the IEEE-754 standard [31]). Likewise, there are several popular truncation approaches (also called *rounding*), such as round-to-the-nearest, round-to-zero, round-up, and round-down. Our results in this paper hold for both formats and all these truncation approaches.

Specifically, the log-precision assumption means that we can use $O(\log(n))$ bits to represent a real number, where the length of the input sequence is bounded by $n$. For any floating-point format described above with $O(\log(n))$ bits, an important property is that it can represent all real numbers of magnitude $O(\text{poly}(n))$ within $O(\text{poly}(1/n))$ truncation error. We next analyze how the truncation error will propagate and magnify in a log-precision Transformer from the input to the output layer. Note that since the functions represented by Transformers are continuous, the approximation error in a hidden neuron will *smoothly* influence the approximation error of subsequent neurons in deeper layers. This impact can be bounded by the Lipschitz constant of the Transformer, which depends on its basic layers. In particular, the softmax function (in attention) is 1-Lipschitz, the GeLU activation is 2-Lipschitz, and the Lipschitz constant of a linear layer depends on the scale of its weight parameters. Combining these together leads to the following result: given a bounded-depth log-precision Transformer of polynomial size, when all parameter values of the Transformer are further bounded by $O(\text{poly}(n))$, all neuron values only yield an $O(\text{poly}(1/n))$ approximation error compared to the infinite-precision counterpart. Therefore, if a problem can be solved by a bounded-depth polynomial-size infinite-precision Transformer with *polynomially-bounded* parameters, it can also be solved by a log-precision Transformer of the same size. This finding is helpful for understanding Theorems 3.3, 3.4 and 4.7.

Finally, we point out that a key property of log-precision Transformer is that each neuron can only hold $O(\log(n))$-bit information and thus cannot store the full information of the entire input sequence. Therefore, the log-precision assumption captures the idea that the computation must be somehow distributed on each token, which well-resembles practical situations and the way Transformers work.

# B  Formal Definitions of CoT in Section 3

In this section, we will formally define the CoT derivation formats for the two math problems described in Section 3.

**Arithmetic expression**. In an arithmetic expression that contains operators, there exists at least one pair of neighboring numbers connected by an operator that can be calculated, which we refer to as a *handle*. More precisely, one can represent an arithmetic expression into a (binary) syntax tree where each number is a leaf node and each operator is an internal node that has two children. In this case, a pair of neighboring numbers is a handle if they share the same parent in the syntax tree. For instance, consider the arithmetic formula $7 \times (6 + 5 + 4 \times 5)$. Then, $6 + 5$ and $4 \times 5$ are two handles.

An important observation is that we can determine whether a pair of numbers $a$ and $b$ can form a handle with the operator op by examining the token before $a$ and the token after $b$, where these tokens are either operators, brackets, or empty (i.e., approaching the beginning/ending of the sequence, including the equal sign '='). Specifically, given subsequence $s_1 \; a \; \text{op} \; b \; s_2$, we have that $a \; \text{op} \; b$ forms a handle iff one of the following conditions holds:

- op $\in \{+, -\}$ and $s_1 \in \{ \, ( \, , \texttt{empty}\}$, $s_2 \notin \{\times, \div\}$;
- op $\in \{\times, \div\}$ and $s_1 \notin \{\times, \div\}$.

In the proposed chain of thought (CoT), an autoregressive Transformer calculates *one* handle at each step. If there are multiple handles, the leftmost handle is selected. The subsequence $a \; \text{op} \; b$ is then replaced by the calculated result. For the case of $s_1 = ($ and $s_2 = )$, there will be a pair of redundant brackets and thus the two tokens are removed. It is easy to see that the resulting sequence is still a valid arithmetic expression. By following this process, each CoT step reduces one operator and the formula is gradually simplified until there is only one number left, yielding the final answer.

**System of linear equations**. Assume that we have a system of $m$ linear equations with variables $x_1, x_2, \ldots, x_m$. The $i$-th equation in the input sequence is grammatically written as $a_{i1}x_1 + a_{i2}x_2 + \cdots + a_{im}x_m = b_i$, where $a_{ij} \in \{0, \cdots, p-1\}$ and $b_i \in \{0, \cdots, p-1\}$. For simplicity, we do not omit the token $a_{ij}$ or $a_{ij}x_j$ in the input sequence when $a_{ij} \in \{1, 0\}$.

We can construct the following CoT to solve the equations by using the Gaussian elimination algorithm. At each step $i$, we select an equation $k$ satisfying the following two conditions:

- The coefficient of $x_i$ is nonzero.
- The coefficients of $x_1, \cdots, x_{i-1}$ are all zero.

Such an equation must exist, otherwise the solution is not unique or does not exist. If there are multiple equations satisfying the above conditions, we choose the $k$-th equation with the smallest index $k$. We then swap it with equation $i$, so that the $i$-th equation now satisfy the above conditions.

We then eliminate the variable $x_i$ in all other equations by leveraging equation $i$. Formally, denote the $i$-th equation at the $i$-th step as

$$a_{ii}^{(i)} x_i + a_{i,i+1}^{(i)} x_{i+1} + \cdots + a_{im}^{(i)} x_m = b_i, \tag{7}$$

and denote the coefficient of $x_i$ in the $j$-th equation ($j \neq i$) as $a_{ji}^{(i)}$. We can multiply (7) by $-(a_{ii}^{(i)})^{-1} a_{ji}^{(i)}$ and add the resulting equation to the $j$-th equation. This will eliminate the term $x_i$ in the $j$-th equation. We further normalize equation $i$ so that the coefficient $a_{ii}^{(i)}$ becomes 1. Depending on whether $j \leq i$ or $j > i$, the resulting equation in the CoT output will have the following grammatical form:

- If $j \leq i$, the $j$-th equation will be written as $x_j + \tilde{a}_{j,i+1} x_{i+1} + \cdots + \tilde{a}_{jm} x_m = \tilde{b}_j$;
- If $j > i$, the $j$-th equation will be written as $\tilde{a}_{j,i+1} x_{i+1} + \cdots + \tilde{a}_{jm} x_m = \tilde{b}_j$.

Note that we remove all zero terms $\tilde{a}_{jk} x_k$ for $k \leq i, k \neq j$ in the CoT output and also remove the coefficient 1 in $\tilde{a}_{kk} x_k$ for $k \leq i$, similar to how human write solutions (see Figure 1 for an illustration). However, to simplify our proof, we reserve the coefficient 0 or 1 (i.e., outputting $0 x_k$ or $1 x_k$) when $k > i$ since it cannot be determined easily before computing the coefficient. The above process is repeated for $m - 1$ steps, and after the final step we obtain the solution.

## C  Technical Lemmas

### C.1  Technical lemmas for MLP

In this subsection, we will demonstrate the representation efficiency of two-layer MLPs in performing several basic operations, such as multiplication, linear transformation, conditional selection, and look-up table. These operations will serve as building blocks in performing complex tasks.

We first show that a two-layer MLP with GeLU activation can efficiently approximate the scalar multiplication, with all weights bounded by $O(\text{poly}(1/\epsilon))$ where $\epsilon$ is the approximation error.

**Lemma C.1.** *Let $f : \mathbb{R}^2 \to \mathbb{R}$ be a two-layer MLP with GeLU activation, and the hidden dimension is 4. Then, for any $\epsilon > 0$ and $M > 0$, there exist MLP parameters with $\ell_\infty$ norm upper bounded by $O(\text{poly}(M, 1/\epsilon))$ such that $|f(a,b) - ab| \leq \epsilon$ holds for all $a, b \in [-M, M]$.*

*Proof.* Denote the input vector to the MLP as $(a, b) \in \mathbb{R}^2$. After the first linear layer, it is easy to construct a weight matrix such that the hidden vector is $\frac{1}{\lambda}(a + b, -a - b, a - b, -a + b) \in \mathbb{R}^4$, where $\lambda$ is an arbitrary scaling factor. Let $\sigma$ be the GeLU activation. We can similarly construct a weight vector such that the final output of the MLP is

$$f(a,b) = \frac{\sqrt{2\pi} \lambda^2}{8} \left( \sigma\left(\frac{a+b}{\lambda}\right) + \sigma\left(\frac{-a-b}{\lambda}\right) - \sigma\left(\frac{a-b}{\lambda}\right) - \sigma\left(\frac{-a+b}{\lambda}\right) \right).$$

We will prove that the above MLP satisfies the theorem by picking an appropriate $\lambda$. By definition of GeLU activation, $\sigma(x) = x\Phi(x)$ where $\Phi(x)$ is the standard Gaussian cumulative distribution function. We thus have $\sigma'(0) = 0.5$ and $\sigma''(0) = \sqrt{2/\pi}$. Applying Taylor's formula and assuming $\lambda > 2M$, we have

$$\left| \sigma\left(\frac{a+b}{\lambda}\right) + \sigma\left(\frac{-a-b}{\lambda}\right) - \sigma\left(\frac{a-b}{\lambda}\right) - \sigma\left(\frac{-a+b}{\lambda}\right) - \frac{8ab}{\sqrt{2\pi}\lambda^2} \right|$$

$$\leq \left| \frac{1}{2}\sqrt{\frac{2}{\pi}} \left( \left(\frac{a+b}{\lambda}\right)^2 + \left(\frac{-a-b}{\lambda}\right)^2 - \left(\frac{a-b}{\lambda}\right)^2 - \left(\frac{-a+b}{\lambda}\right)^2 \right) - \frac{8ab}{\sqrt{2\pi}\lambda^2} \right|$$

$$\quad + \frac{4}{3!} \frac{(2M)^3}{\lambda^3} \left| \max_{x \in [-1,1]} \sigma^{(3)}(x) \right|$$

$$= \frac{16M^3}{3\lambda^3} \max_{x \in [-1,1]} \frac{1}{\sqrt{2\pi}} (x^3 - 4x) \exp(-\frac{x^2}{2}) < \frac{80M^3}{3\sqrt{2\pi}\lambda^3}.$$

Therefore, $|f(a,b) - ab| < \frac{10M^3}{3\lambda}$. Set $\lambda \geq \frac{10M^3}{3\epsilon}$, and then we can obtain $|f(a,b) - ab| < \epsilon$. Moreover, each weight element in the MLP is upper bounded by $O(\lambda^2)$, which is clearly $O(\text{poly}(M, 1/\epsilon))$. □

Next, we will demonstrate that a two-layer MLP with GeLU activation can efficiently approximate a two-layer MLP with ReLU activation, with all weights upper bounded by $O(\text{poly}(1/\epsilon))$. This result is useful in proving subsequent lemmas.

**Lemma C.2.** *Let $\boldsymbol{g} : \mathbb{R}^{d_1} \to \mathbb{R}^{d_2}$ be a two-layer MLP with* ReLU *activation, and all parameter values are upper bounded by $M$. Then, for any $\epsilon > 0$, there exists a two-layer MLP $\boldsymbol{f}$ of the same size with* GeLU *activation and parameters upper bounded by $O(\text{poly}(M, 1/\epsilon))$ in the $\ell_\infty$ norm, such that for all $\boldsymbol{x} \in \mathbb{R}^{d_1}$, we have $\|\boldsymbol{f}(\boldsymbol{x}) - \boldsymbol{g}(\boldsymbol{x})\|_\infty \leq \epsilon$.*

*Proof.* Let $\boldsymbol{g}(\boldsymbol{x}) = \boldsymbol{W}_2 \cdot \text{ReLU}(\boldsymbol{W}_1 \boldsymbol{x})$. We construct $\boldsymbol{f}(x) = \frac{1}{\lambda} \boldsymbol{W}_2 \cdot \text{GeLU}(\lambda \boldsymbol{W}_1 \boldsymbol{x})$ where $\lambda > 0$ is a sufficiently large constant. To prove that $\|\boldsymbol{f}(\boldsymbol{x}) - \boldsymbol{g}(\boldsymbol{x})\|_\infty \leq \epsilon$ for all $\boldsymbol{x} \in \mathbb{R}^{d_1}$, it suffices to prove that $\|\boldsymbol{W}_2(\text{ReLU}(\boldsymbol{z}) - \frac{1}{\lambda}\text{GeLU}(\lambda \boldsymbol{z}))\|_\infty \leq \epsilon$ for all $\boldsymbol{z} \in \mathbb{R}^d$ where $d$ is the hidden size. Since

$$\left\| \boldsymbol{W}_2 \left( \text{ReLU}(\boldsymbol{z}) - \frac{1}{\lambda}\text{GeLU}(\lambda \boldsymbol{z}) \right) \right\|_\infty \leq \|\boldsymbol{W}_2\|_\infty \left\| \text{ReLU}(\boldsymbol{z}) - \frac{1}{\lambda}\text{GeLU}(\lambda \boldsymbol{z})) \right\|_\infty$$

$$\leq Md \left\| \text{ReLU}(\boldsymbol{z}) - \frac{1}{\lambda}\text{GeLU}(\lambda \boldsymbol{z})) \right\|_\infty,$$

it suffices to consider the scalar setting and prove that $|\frac{1}{\lambda}\text{GeLU}(\lambda y) - \text{ReLU}(y)| \leq \epsilon/Md$ for all $y \in \mathbb{R}$. By definition of ReLU and GeLU, we have

$$\left| \frac{1}{\lambda}\text{GeLU}(\lambda y) - \text{ReLU}(y) \right| = \frac{1}{\lambda} \left| \text{ReLU}(\lambda y) - \int_{-\infty}^{\lambda y} \frac{\lambda y}{\sqrt{2\pi}} \exp\left(-\frac{t^2}{2}\right) dt \right|. \tag{8}$$

When $y \geq 0$, (8) becomes

$$\frac{1}{\lambda} \left| \int_{-\infty}^{+\infty} \frac{\lambda y}{\sqrt{2\pi}} \exp\left(-\frac{t^2}{2}\right) dt - \int_{-\infty}^{\lambda y} \frac{\lambda y}{\sqrt{2\pi}} \exp\left(-\frac{t^2}{2}\right) dt \right| = \frac{1}{\lambda} \int_{\lambda y}^{+\infty} \frac{\lambda y}{\sqrt{2\pi}} \exp\left(-\frac{t^2}{2}\right) dt.$$

Combined with the case of $y < 0$, (8) can be consistently written as

$$\left| \frac{1}{\lambda}\text{GeLU}(\lambda y) - \text{ReLU}(y) \right| = \frac{1}{\lambda} \int_{\lambda|y|}^{+\infty} \frac{\lambda|y|}{\sqrt{2\pi}} \exp\left(-\frac{t^2}{2}\right) dt$$

$$\leq \frac{1}{\sqrt{2\pi}\lambda} \int_{\lambda|y|}^{+\infty} t \exp\left(-\frac{t^2}{2}\right) dt = \frac{1}{\sqrt{2\pi}\lambda} \exp\left(-\frac{\lambda^2 y^2}{2}\right)$$

$$\leq \frac{1}{\sqrt{2\pi}\lambda}.$$

Picking $\lambda = \frac{Md}{\sqrt{2\pi}\epsilon}$ yields the desired result and completes the proof. □

Equipped with the above result, we now prove that a two-layer MLP with GeLU activation can perform linear transformation and conditional selection.

**Proposition C.3.** *Let $\boldsymbol{f} : \mathbb{R}^{d_1} \to \mathbb{R}^{d_2}$ be a two-layer MLP with* GeLU *activation, and the hidden dimension is $2d_2$. Let $\boldsymbol{W} \in \mathbb{R}^{d_2 \times d_1}$ be any matrix and denote $M = \max_{ij} |W_{ij}|$. Then, for any $\epsilon > 0$, there exist MLP parameters with $\ell_\infty$ norm bounded by $O(\text{poly}(M, 1/\epsilon))$, such that for any $\boldsymbol{x} \in \mathbb{R}^{d_1}$, we have $\|\boldsymbol{f}(\boldsymbol{x}) - \boldsymbol{W}\boldsymbol{x}\|_\infty \leq \epsilon$.*

*Proof.* We can use a two-layer MLP with ReLU activation to implement $\boldsymbol{g}(\boldsymbol{x}) = \boldsymbol{W}\boldsymbol{x}$ by the following construction:

$$\boldsymbol{W}\boldsymbol{x} = \text{ReLU}(\boldsymbol{W}\boldsymbol{x}) + \text{ReLU}(-\boldsymbol{W}\boldsymbol{x})$$

Combined with Lemma C.2, we can also implement $\boldsymbol{g}(\boldsymbol{x})$ by a two-layer MLP with GeLU activation. □

**Lemma C.4.** *Define the selection function* $\boldsymbol{g} : \mathbb{R}^d \times \mathbb{R}^d \times \mathbb{R} \to \mathbb{R}^d$ *as follows:*

$$\boldsymbol{g}(\boldsymbol{x}, \boldsymbol{y}, t) = \begin{cases} \boldsymbol{x} & \text{if } t \geq 0, \\ \boldsymbol{y} & \text{if } t < 0. \end{cases} \tag{9}$$

*Let* $\boldsymbol{f} : \mathbb{R}^d \times \mathbb{R}^d \times \mathbb{R} \to \mathbb{R}^d$ *be a two-layer MLP with GeLU activation, and the hidden dimension is* $2d + 2$. *Then, for any* $\epsilon > 0$, $\alpha > 0$, *and* $M > 0$, *there exist MLP parameters with* $\ell_\infty$ *norm bounded by* $O(\text{poly}(M, 1/\alpha, 1/\epsilon))$, *such that for all* $\boldsymbol{x} \in [-M, M]^d$, $\boldsymbol{y} \in [-M, M]^d$, *and* $t \in [-\infty, -\alpha] \cup [\alpha, +\infty]$, *we have* $\|\boldsymbol{f}(\boldsymbol{x}, \boldsymbol{y}, t) - \boldsymbol{g}(\boldsymbol{x}, \boldsymbol{y}, t)\|_\infty \leq \epsilon$.

*Proof.* We can simply use a two-layer MLP with ReLU activation to implement $\boldsymbol{g}$ by the following construction:

$$\boldsymbol{h}(\boldsymbol{x}, \boldsymbol{y}, t) = (\boldsymbol{h}_1, \boldsymbol{h}_2, h_3, h_4) := (\boldsymbol{x} + \alpha^{-1}Mt\mathbf{1}_d, \boldsymbol{y} - \alpha^{-1}Mt\mathbf{1}_d, \alpha^{-1}Mt, -\alpha^{-1}Mt) \in \mathbb{R}^{2d+2},$$
$$\boldsymbol{f}(\boldsymbol{x}, \boldsymbol{y}, t) = \text{ReLU}(\boldsymbol{h}_1) - \text{ReLU}(h_3)\mathbf{1}_d + \text{ReLU}(\boldsymbol{h}_2) - \text{ReLU}(h_4)\mathbf{1}_d,$$

where $\mathbf{1}_d$ is the all-one vector of $d$ dimension. It is easy to check that, for all $\boldsymbol{x} \in [-M, M]^{d_1}$, $\boldsymbol{y} \in [-M, M]^{d_2}$, and $t \in [-\infty, -\alpha] \cup [\alpha, +\infty]$, we have $\boldsymbol{f}(\boldsymbol{x}, \boldsymbol{y}, t) = \boldsymbol{g}(\boldsymbol{x}, \boldsymbol{y}, t)$. Moreover, all parameters are bounded by $O(M/\alpha)$. Therefore, by using Lemma C.2, we can also implement $\boldsymbol{g}(\boldsymbol{x})$ by a two-layer MLP with GeLU activation and all parameters are bounded by $O(\text{poly}(M, 1/\alpha, 1/\epsilon))$. $\quad\square$

We final show that a two-layer MLP can efficiently represent a look-up table. Consider a $k$-dimensional table of size $d^k$, where each element in the table is an integer ranging from 1 to $d$. Denote the set $\mathcal{D} = \{\boldsymbol{e}_i : i \in [d]\}$, where $\boldsymbol{e}_i$ is a $d$-dimensional one-hot vector with the $i$-th element being 1. The above look-up table can thus be represented as a discrete function $\boldsymbol{g} : \mathcal{D}^k \to \mathcal{D}$. The following lemma shows that $\boldsymbol{g}$ can be implemented by a two-layer MLP with GeLU activation.

**Lemma C.5.** *Let* $\boldsymbol{g} : \mathcal{D}^k \to \mathcal{D}$ *be any function defined above, and let* $\boldsymbol{f} : \mathbb{R}^{k \times d} \to \mathbb{R}^d$ *be a two-layer MLP with* GeLU *activation and bias, and the hidden dimension is* $d^k$. *Then, for any* $\epsilon > 0$, *there exist MLP parameters with* $\ell_\infty$ *norm bounded by* $O(\text{poly}(k, 1/\epsilon))$, *such that for all* $\boldsymbol{x} \in \mathcal{D}^k \subset \mathbb{R}^{k \times d}$ *and all perturbation* $\boldsymbol{\delta} \in [-1/2k, 1/2k]^{k \times d}$, *we have* $\|\boldsymbol{f}(\boldsymbol{x} + \boldsymbol{\delta}) - \boldsymbol{g}(\boldsymbol{x})\|_\infty \leq \epsilon + 2k\|\boldsymbol{\delta}\|_\infty$, *where* $\|\boldsymbol{\delta}\|_\infty$ *is the vector* $\ell_\infty$*-norm applied to the flattended matrix* $\boldsymbol{\delta}$.

*Proof.* We can simply use a two-layer MLP with ReLU activation to implement $\boldsymbol{g}$ by the following construction. Denote the index of the MLP hidden layer as $(i_1, \cdots, i_k) \in [d]^k$. We can construct the weights of the first MLP layer such that

$$h_{(i_1, \cdots, i_k)}(\boldsymbol{x}) = 2(x_{i_1} + x_{d+i_2} \cdots + x_{(k-1)d+i_k}) - 2k + 1.$$

We can then construct the weights of the second layer such that the final output of the MLP is

$$f_j(\boldsymbol{x}) = \sum_{g_j(\boldsymbol{e}_{i_1}, \cdots, \boldsymbol{e}_{i_k})=1} \text{ReLU}(h_{(i_1, \cdots, i_k)}(\boldsymbol{x})).$$

One can check that $\boldsymbol{f}(\boldsymbol{x}) = \boldsymbol{g}(\boldsymbol{x})$ holds for all $\boldsymbol{x} \in \mathcal{D}^k \subset \mathbb{R}^{d \times k}$. Furthermore, we have
$\|\boldsymbol{f}(\boldsymbol{x} + \boldsymbol{\delta}) - \boldsymbol{g}(\boldsymbol{x})\|_\infty = \|\boldsymbol{f}(\boldsymbol{x} + \boldsymbol{\delta}) - \boldsymbol{f}(\boldsymbol{x})\|_\infty$

$$= \max_{j \in [d]} \left| \sum_{g_j(\boldsymbol{e}_{i_1}, \cdots, \boldsymbol{e}_{i_k})=1} \left( \text{ReLU}(h_{(i_1, \cdots, i_k)}(\boldsymbol{x} + \boldsymbol{\delta})) - \text{ReLU}(h_{(i_1, \cdots, i_k)}(\boldsymbol{x})) \right) \right|$$

$$\leq \max_{(i_1, \cdots, i_k) \in [d]^k} \left| h_{(i_1, \cdots, i_k)}(\boldsymbol{x} + \boldsymbol{\delta}) - h_{(i_1, \cdots, i_k)}(\boldsymbol{x}) \right| \leq 2k\|\boldsymbol{\delta}\|_\infty$$

for all perturbations $\boldsymbol{\delta} \in [-1/2k, 1/2k]^{k \times d}$. Thus by using Lemma C.2, we can also implement $\boldsymbol{g}(\boldsymbol{x})$ by a two-layer MLP with GeLU activation and all parameters are bounded by $O(\text{poly}(k, 1/\epsilon))$. $\quad\square$

## C.2 Technical lemmas for the attention layer

In this subsection, we will introduce two special operations that can be performed by the attention layer (with causal mask). Below, let $n \in \mathbb{N}$ be an integer and let $\boldsymbol{x}_1, \boldsymbol{x}_2, \cdots, \boldsymbol{x}_n$ be a sequence of vectors where $\boldsymbol{x}_i = (\tilde{\boldsymbol{x}}_i, r_i, 1) \in [-M, M]^{d+2}$, $\tilde{\boldsymbol{x}}_i \in \mathbb{R}^d$, $r_i \in \mathbb{R}$, and $M$ is a large constant. Let $\boldsymbol{K}, \boldsymbol{Q}, \boldsymbol{V} \in \mathbb{R}^{d' \times (d+2)}$ be any matrices with $\|\boldsymbol{V}\|_\infty \leq 1$, and let $0 < \rho, \delta < M$ be any real numbers. Denote $\boldsymbol{q}_i = \boldsymbol{Q}\boldsymbol{x}_i$, $\boldsymbol{k}_j = \boldsymbol{K}\boldsymbol{x}_j$, $\boldsymbol{v}_j = \boldsymbol{V}\boldsymbol{x}_j$, and define the *matching set* $\mathcal{S}_i = \{j \leq i : |\boldsymbol{q}_i \cdot \boldsymbol{k}_j| \leq \rho\}$. Equipped with these notations, we define two basic operations as follows:

- COPY: The output is a sequence of vectors $\boldsymbol{u}_1, \cdots, \boldsymbol{u}_n$ with $\boldsymbol{u}_i = \boldsymbol{v}_{\mathrm{pos}(i)}$, where $\mathrm{pos}(i) = \mathrm{argmax}_{j \in \mathcal{S}_i}\, r_j$.
- MEAN: The output is a sequence of vectors $\boldsymbol{u}_1, \cdots, \boldsymbol{u}_n$ with $\boldsymbol{u}_i = \mathrm{mean}_{j \in \mathcal{S}_i}\, \boldsymbol{v}_j$.

The output $\boldsymbol{u}_i$ is undefined when $\mathcal{S}_i = \emptyset$. We next make the following regularity assumption:

**Assumption C.6.** The matrices $\boldsymbol{Q}, \boldsymbol{K}, \boldsymbol{V}$ and scalars $\rho, \delta$ satisfy that for all considered sequences $\boldsymbol{x}_1, \boldsymbol{x}_2, \cdots, \boldsymbol{x}_n$, the following hold:

- For any $i, j \in [n]$, either $|\boldsymbol{q}_i \cdot \boldsymbol{k}_j| \leq \rho$ or $\boldsymbol{q}_i \cdot \boldsymbol{k}_j \leq -\delta$.

- For any $i, j \in [n]$, either $i = j$ or $|r_i - r_j| \geq \delta$.

Assumption C.6 says that there are sufficient gaps between the attended position (e.g., $\mathrm{pos}(i)$) and other positions. The two lemmas below show that the attention layer with casual mask can implement both COPY operation and MEAN operation efficiently.

**Lemma C.7.** *Assume Assumption C.6 holds with $\rho \leq \frac{\delta^2}{8M}$. For any $\epsilon > 0$, there exists an attention layer with embedding size $O(d)$ and one causal attention head that can approximate the COPY operation defined above. Formally, for any considered sequence of vectors $\boldsymbol{x}_1, \boldsymbol{x}_2, \ldots, \boldsymbol{x}_n$, denote the corresponding attention output as $\boldsymbol{o}_1, \boldsymbol{o}_2, \ldots, \boldsymbol{o}_n$. Then, we have $\|\boldsymbol{o}_i - \boldsymbol{u}_i\|_\infty \leq \epsilon$ for all $i \in [n]$ with $\mathcal{S}_i \neq \emptyset$. Moreover, the $\ell_\infty$ norm of attention parameters is bounded by $O(\mathrm{poly}(M, 1/\delta, \log(n), \log(1/\epsilon)))$.*

*Proof.* The purpose of the attention head is to focus only on the vector that needs to be copied. To achieve this, we construct the key, query, and value vectors as follows (by assigning suitable key, query, and value weight matrices in the attention head):

- Query: $(\lambda \boldsymbol{q}_i, \mu) \in \mathbb{R}^{d+1}$

- Key: $(\boldsymbol{k}_i, r_i) \in \mathbb{R}^{d+1}$

- Value: $\boldsymbol{v}_i \in \mathbb{R}^d$

where $\lambda$ and $\mu$ are constants that will be defined later. Denote $a_{ij}$ as the attention score, then

$$a_{i,j} = \frac{\exp(\lambda(\boldsymbol{q}_i \cdot \boldsymbol{k}_j) + \mu r_j)}{\sum_{j'} \exp(\lambda(\boldsymbol{q}_i \cdot \boldsymbol{k}_{j'}) + \mu r_{j'})} = \frac{\exp(\lambda(\boldsymbol{q}_i \cdot \boldsymbol{k}_j))}{\sum_{j'} \exp(\lambda(\boldsymbol{q}_i \cdot \boldsymbol{k}_{j'}) + \mu(r_{j'} - r_j))}.$$

Since $\rho \leq \frac{\delta^2}{8M}$ and $M \geq \delta$, we have $\delta - \rho \geq \frac{7}{8}\delta$. By setting $\lambda = \frac{8M \ln(\frac{2nM}{\epsilon})}{\delta^2}$ and $\mu = \frac{3 \ln(\frac{2nM}{\epsilon})}{\delta}$ (which are bounded by $O(\mathrm{poly}(M, 1/\delta, \log(n), \log(1/\epsilon)))$), we have

$$a_{i,\mathrm{pos}(i)} \geq \frac{\exp(-\lambda\rho)}{\exp(-\lambda\rho) + (n-1)\exp(\max(-\lambda\delta + 2M\mu, \lambda\rho - \mu\delta))} \tag{10}$$

$$= \frac{1}{1 + (n-1)\exp(\max(-\lambda(\delta - \rho) + 2M\mu, 2\lambda\rho - \mu\delta))}$$

$$\geq 1 - n\exp(\max(-\lambda(\delta - \rho) + 2M\mu, 2\lambda\rho - \mu\delta)) \tag{11}$$

$$\geq 1 - n\exp\left(\max\left(-\frac{M}{\delta}\ln\left(\frac{2nM}{\epsilon}\right), -\ln\left(\frac{2nM}{\epsilon}\right)\right)\right)$$

$$\geq 1 - n\exp\left(-\ln\left(\frac{2nM}{\epsilon}\right)\right) \tag{12}$$

$$= 1 - \frac{\epsilon}{2M},$$

where in (10) we use Assumption C.6, which implies that whenever $j' \neq \mathrm{pos}(i)$, either $\boldsymbol{q}_i \cdot \boldsymbol{k}_{j'} \leq -\delta$ or $(\boldsymbol{q}_i \cdot \boldsymbol{k}_{j'} \leq \rho$ and $r_{j'} - r_j \leq -\delta)$; in (11) we use the inequality $\frac{1}{1+x} \geq 1 - x$ for all $x \geq 0$; in (12) we use the fact that $M \geq \delta$. We thus have

$$\|\boldsymbol{o}_i - \boldsymbol{u}_i\|_\infty = \left\|\sum_j a_{ij}\boldsymbol{v}_j - \boldsymbol{v}_{\mathrm{pos}(i)}\right\|_\infty \leq M\|\boldsymbol{V}\|_\infty \cdot \left(1 - a_{i,\mathrm{pos}(i)} + \sum_{j \neq \mathrm{pos}(i)} a_{i,j}\right)$$

$$= M\|\boldsymbol{V}\|_\infty (2 - 2a_{i,\mathrm{pos}(i)}) \leq \epsilon,$$

which concludes the proof. $\qquad\square$

**Lemma C.8.** *Assume Assumption C.6 holds with* $\rho \leq \frac{\delta\epsilon}{16M \ln(\frac{4Mn}{\epsilon})}$. *For any* $0 < \epsilon \leq M$, *there exists an attention layer with embedding size* $O(d)$ *and one causal attention head that can approximate the MEAN operation defined above. Formally, for any considered sequence of vectors* $\boldsymbol{x}_1, \boldsymbol{x}_2, \ldots, \boldsymbol{x}_n$, *denote the attention output as* $\boldsymbol{o}_1, \boldsymbol{o}_2, \ldots, \boldsymbol{o}_n$. *Then, we have* $\|\boldsymbol{o}_i - \boldsymbol{u}_i\|_\infty \leq \epsilon$ *for all* $i \in [n]$ *with* $\mathcal{S}_i \neq \emptyset$. *Moreover, the* $\ell_\infty$ *norm of attention parameters is bounded by* $O(\text{poly}(M, 1/\delta, \log(n), \log(1/\epsilon)))$.

*Proof.* The purpose of the attention head is to average across all tokens that satisfy the condition $\boldsymbol{q}_i \cdot \boldsymbol{k}_j \approx 0$. To achieve this, we construct the key, query, and value vectors as follows:

- Query: $\lambda \boldsymbol{q}_i \in \mathbb{R}^d$

- Key: $\boldsymbol{k}_i \in \mathbb{R}^d$

- Value: $\boldsymbol{v}_i \in \mathbb{R}^d$

where $\lambda$ is a constant which will be defined later. Denote $a_{ij}$ as the attention score, then

$$a_{i,j} = \frac{\exp(\lambda \boldsymbol{q}_i \cdot \boldsymbol{k}_j)}{\sum_{j'} \exp(\lambda \boldsymbol{q}_i \cdot \boldsymbol{k}_{j'})}.$$

By setting $\lambda = \frac{1}{\delta} \ln\left(\frac{4Mn}{\epsilon}\right)$ (which is bounded by $O(\text{poly}(M, 1/\delta, \log(n), \log(1/\epsilon)))$), we have:

$$\sum_{j \notin \mathcal{S}_i} a_{ij} \leq \frac{(n - |\mathcal{S}_i|) \exp(-\lambda\delta)}{(n - |\mathcal{S}_i|) \exp(-\lambda\delta) + |\mathcal{S}_i| \exp(-\lambda\rho)} \tag{13}$$

$$= \frac{1}{1 + \frac{|\mathcal{S}_i|}{n - |\mathcal{S}_i|} \exp(-\lambda(\rho - \delta))}$$

$$< n \exp(\lambda\rho) \exp(-\lambda\delta)) \tag{14}$$

$$\leq n \exp\left(\frac{\epsilon}{16M}\right) \exp\left(-\ln\left(\frac{4Mn}{\epsilon}\right)\right)$$

$$< \frac{\epsilon}{3M}, \tag{15}$$

where in (13) we use Assumption C.6, which implies that $\boldsymbol{q}_i \cdot \boldsymbol{k}_j \leq -\delta$ for all $j \notin \mathcal{S}_i$ and $\boldsymbol{q}_i \cdot \boldsymbol{k}_j \geq -\rho$ for all $j \in \mathcal{S}_i$; in (14) we use the inequality $\frac{1}{1+x} < \frac{1}{x}$ for all $x > 0$; in (15) we use that the assumption that $\epsilon \leq M$ and the fact that $\exp(1/16) < 4/3$.

Similarly, for any $j \in \mathcal{S}_i$, we have

$$\left| a_{ij} - \frac{1}{|\mathcal{S}_i|} \right| \leq \max\left( \frac{1}{|\mathcal{S}_i|} - \frac{\exp(-\rho\lambda)}{|\mathcal{S}_i| \exp(\rho\lambda) + (n - |\mathcal{S}_i|) \exp(-\lambda\delta)}, \frac{\exp(\rho\lambda)}{|\mathcal{S}_i| \exp(-\rho\lambda)} - \frac{1}{|\mathcal{S}_i|} \right) \tag{16}$$

$$\leq \frac{1}{|\mathcal{S}_i|} \max\left( 1 - \frac{1}{\exp(2\rho\lambda) + n \exp(-\lambda(\delta - \rho))}, \exp(2\rho\lambda) - 1 \right)$$

$$\leq \frac{1}{|\mathcal{S}_i|} \max\left( \exp(2\rho\lambda) - 1 + n \exp(-\lambda(\delta - \rho)), \exp(2\rho\lambda) - 1 \right) \tag{17}$$

$$= \frac{1}{|\mathcal{S}_i|} \left( \exp(2\rho\lambda) - 1 + n \exp(-\lambda(\delta - \rho)) \right)$$

$$\leq \frac{1}{|\mathcal{S}_i|} \left( \exp\left(\frac{\epsilon}{8M}\right) - 1 + \frac{\epsilon}{3M} \right) \tag{18}$$

$$\leq \frac{2\epsilon}{3M|\mathcal{S}_i|} \tag{19}$$

where in (16) we use Assumption C.6 similarly as before; in (17) we use the inequality $1 - \frac{1}{x} \leq x - 1$ for all $x > 0$; in (18) we use the inequality previously derived in (15); in (19) we use the inequality $\exp(x) \leq 1 + 2x$ for all $0 \leq x \leq 1$. We thus obtain

$$\|\boldsymbol{o}_i - \boldsymbol{u}_i\|_\infty = \left\| \sum_j a_{ij} \boldsymbol{v}_j - \frac{1}{|\mathcal{S}_i|} \sum_{j \in \mathcal{S}_i} \boldsymbol{v}_j \right\|_\infty \leq M \|\boldsymbol{V}\|_\infty \cdot \left( \sum_{j \notin \mathcal{S}_i} a_{ij} + \sum_{j \in \mathcal{S}_i} \left| a_{ij} - \frac{1}{|\mathcal{S}_i|} \right| \right) \leq \epsilon,$$

which concludes the proof. $\square$

# D  Arithmetic Formula

In this section, we prove that the autoregressive Transformer can evaluate arithmetic expressions when equipped with CoT, whereas the it cannot solve this task without CoT.

## D.1  Proof of Theorem 3.3

Before proving this theorem, there is one point that needs to be clarified: all residual connections in the attention/MLP layers can be replaced by concatenation, in the sense that both architectures have the same expressive power. Formally, consider an MLP (or an attention layer) denoted as $\boldsymbol{f} : \mathbb{R}^d \to \mathbb{R}^d$, and let $\boldsymbol{y} = \boldsymbol{f}(\boldsymbol{x})$. It is easy to see that we can construct another MLP (or attention layer) denoted as $\boldsymbol{g} : \mathbb{R}^{2d} \to \mathbb{R}^{2d}$ such that $\boldsymbol{g}(\boldsymbol{x}, \boldsymbol{0}) + (\boldsymbol{x}, \boldsymbol{0}) = (\boldsymbol{0}, \boldsymbol{y}) + (\boldsymbol{x}, \boldsymbol{0}) = (\boldsymbol{x}, \boldsymbol{y})$, namely, the residual connection can implement concatenation. Conversely, concatenation can implement residual connection by using a linear projection. Based on the equivalence, we can use the concatenation operation instead of residual connection in all subsequent proofs presented in Appendices D, E and G. Similarly, the output of multi-head attention can be replaced by the concatenation of the output of each head (instead of performing aggregation via matrices $\boldsymbol{W}_O^{(l,h)}$ defined in (2)). For clarity, we further omit the unnecessary parts in the concatenated outputs and only retain the outputs that are used in subsequent layers.

We now present the proof of Theorem 3.3. For ease of reading, we restate Theorem 3.3 below:

**Theorem D.1.** *For any prime $p$ and integer $n > 0$, there exists an autoregressive Transformer defined in Section 2 with hidden size $d = O(\mathrm{poly}(p))$ (independent of $n$), depth $L = 5$, and 5 heads in each layer that can generate the CoT solution defined in Appendix B for all inputs in* $\mathsf{Arithmetic}(n, p)$. *Moreover, all parameter values in the Transformer are bounded by $O(\mathrm{poly}(n))$.*

*Proof sketch.* The intuition behind our construction is that when the CoT output proceeds to a certain position, the Transformer can read the context related to this position and determine whether it should copy a token or perform a calculation. Remarkably, the context only contains a *fixed* number of tokens (as discussed in Appendix B). Based on the key observation, we can construct our five-layer transformer as follows. The first layer collects important positional information. The second and third layers determine whether to perform a calculation by examining the context related to the current token, which contains five tokens. The fourth layer and the fifth layers are used to generate the output via three cases: before/at/after the position that performs a calculation. For the first and the last cases, the output simply copies a previous token with position computed by the two layers. For the middle case, the outcome is computed via a look-up table that stores the arithemtic rules (+, -, ×, ÷).

*Proof.* We construct each layer as follows.

**Token Embeddings**. Assume that we have a sequence of tokens $s_1, \ldots, s_i$ and we want to generate the next token $s_{i+1}$. For any $j \in [n]$, let $\mathrm{id}(s_j)$ be the index of token $s_j$ in the embedding dictionary, with values ranging from 1 to the number of tokens. We can embed the token $s_j$ by $\boldsymbol{x}_j^{(0)} = (\boldsymbol{e}_{\mathrm{id}(s_j)}, j, 1) \in \mathbb{R}^{\mathrm{num\_tokens}+2}$, where $\boldsymbol{e}_j$ is a one-hot vector with the $j$-th element being 1, $j \in \mathbb{N}_+$ is the positional embedding, and the constant embedding 1 is used as a bias term.

**Layer 1**. The first layer of the autoregressive Transformer uses two attention heads to perform the following tasks:

1. Count the number of equal signs ('=') in previous tokens, denoted as $n_i^=$, i.e., $n_i^= = |\{j \le i : s_j = \text{'='}\}|$.

2. Copy the position of the last equal sign, denoted as $p_i^=$, i.e., $p_i^= = \max\{j : j \le i, s_j = \text{'='}\}$. If the set $\{j : j \le i, s_j = \text{'='}\}$ is empty, define $p_i^= = 0$.

3. Compute $i^2$.

Based on Appendix C.2 (Lemma C.8), we can use the first attention head to perform the MEAN operation that counts the percentage of equal signs in the preceding sentences (i.e., $n_i^= / i$). This can be achieved by setting $\boldsymbol{Q} = \boldsymbol{0}$, $\boldsymbol{K} = \boldsymbol{0}$, $\boldsymbol{V} = (\boldsymbol{e}_{\mathrm{id}(\text{'='})}, 0, 0)^\top$ (defined in Appendix C.2), so that

$$\boldsymbol{q}_i = \boldsymbol{0}, \quad \boldsymbol{k}_j = \boldsymbol{0}, \quad v_j = \boldsymbol{e}_{\mathrm{id}(\text{'='})} \cdot \boldsymbol{e}_{\mathrm{id}(s_j)} = \mathbb{I}[s_j = \text{'='}], \quad \mathcal{S}_i = [i].$$

Similarly, we can use the second attention head to perform a COPY operation that copies the position index of the last equal sign (by Lemma C.7). This can be achieved by setting $\boldsymbol{Q} = (0,0,1)^\top$, $\boldsymbol{K} = (e_{\text{id}(\text{‘=’})}, 0, -1)^\top$, $\boldsymbol{V} = (0,1,0)^\top$, $r_j = j$ (defined in Appendix C.2), so that

$$q_i = 1, \quad k_j = \mathbb{I}[s_j = \text{‘=’}] - 1, \quad v_j = j, \quad \mathcal{S}_i = \{j \le i : s_j = \text{‘=’}\}.$$

It is easy to check that the above construction outputs $u_i = \max\{j : j \le i, s_j = \text{‘=’}\}$ when $\mathcal{S}_i \ne \emptyset$. Note that $u_i$ may not equal to $p_i^=$ when $\mathcal{S}_i = \emptyset$.

Using the residual connection to perform concatenation, the output of the attention layer has the form $(e_{\text{id}(s_i)}, i, 1, n_i^=/i, \max\{j : j \le i, s_j = \text{‘=’}\})$. We can then use an MLP to multiply $n_i^=/i$ and $i$ to obtain $n_i^=$ and use another MLP to compute $i^2$ according to Lemma C.1; Simultaneously, we can compute the value $p_i^=$ using the following way:

$$p_i^= = \begin{cases} \max\{j : j \le i, s_j = \text{‘=’}\} & \text{if } n_i^=/i \ge 1/n, \\ 0 & \text{if } n_i^=/i = 0, \end{cases}$$

which is a conditional selection operation and can be implemented by an MLP (Lemma C.4). Also note that the gap in Lemma C.4 is $\alpha = 1/2n$, which can be implemented within log-precision. The final output of the first layer has the form $\boldsymbol{x}_i^{(1)} = (e_{\text{id}(s_i)}, i, i^2, n_i^=, p_i^=, 1)$.

**Layer 2**. The second layer of the Transformer does some tricky preparation work for the next layer.

1. Compute the distance to the *nearest* and the *last* equal sign, denoted as $d_i^=$ and $\hat{d}_i^=$, respectively. Formally, $d_i^= = i - \max\{j : j \le i, s_j = \text{‘=’}\}$, $\hat{d}_i^= = i - \max\{j : j < i, s_j = \text{‘=’}\}$. If the nearest/last equal sign does not exist, define $d_i^= = i$ or $\hat{d}_i^= = i$. The relation between $d_i^=, \hat{d}_i^=$, and $p_i^=$ can be expressed as $d_i^= = i - p_i^=$, $\hat{d}_i^= = i - p_{i-1}^=$.

2. Count the number of equal signs in *strictly* previous tokens, denoted as $\hat{n}_i^=$, i.e., $\hat{n}_i^= = |\{j < i : s_j = \text{‘=’}\}|$.

3. Compute $(n_i^=)^2$, $(\hat{n}_i^=)^2$, $(d_i^=)^2$, and $(\hat{d}_i^=)^2$.

The first and the second tasks can be done using the COPY operation by setting

$$\boldsymbol{q}_i = \boldsymbol{Q}\boldsymbol{x}_i^{(1)} = ((i-1)^2, i-1, 1), \quad \boldsymbol{k}_j = \boldsymbol{K}\boldsymbol{x}_j^{(1)} = (-1, 2j, -j^2), \quad r_j = 0, \quad \boldsymbol{v}_j = (n_j, j - p_j^= + 1).$$

Under the above construction, we have $\boldsymbol{q}_i \cdot \boldsymbol{k}_j = -(i-j-1)^2$, and thus $\mathcal{S}_i = \{i-1\}$, namely, the output is $(\hat{n}_i, \hat{d}_i^=)$. We then use an MLP to calculate $(d_i^=)^2$, $(\hat{d}_i^=)^2$, $(n_i^=)^2$, and $(\hat{n}_i^=)^2$ by using Lemma C.1. The output of the second layer is

$$\boldsymbol{x}_i^{(2)} = (e_{\text{id}(s_i)}, i, n_i^=, \hat{n}_i^=, d_i^=, \hat{d}_i^=, (n_i^=)^2, (\hat{n}_i^=)^2, (d_i^=)^2, (\hat{d}_i^=)^2, 1).$$

**Layer 3**. The third Transformer layer judges whether the calculation should be performed at the current position and computes the result when needed. Based on the CoT format given in Appendix B, we need to extract five previous tokens related to this position. Formally, we need five attention heads to perform the following tasks:

1. Copy the embedding $e_{\text{id}(s_j)}$ located at position $j$ such that $\hat{n}_j^= = n_i^= - 1$ and $\hat{d}_j^= = d_i^= + t$ for $t \in \{1, 2, 3, 4, 5\}$, as shown in Figure 4.

2. Check if the copied expression can be evaluated at the current position according to the rule given in Appendix B. If it can be evaluated, compute the result and determine how much sentence length will be reduced after this calculation (see Appendix B for details on how the reduced sentence length depends on brackets); otherwise, keep the token $e_{\text{id}(s_j)}$ with $\hat{n}_j^= = n_i^= - 1$ and $\hat{d}_j^= = d_i^= + 1$.

We can use the multi-head attention to perform the COPY operation five times in parallel. For each $t$, we construct the matrices $\boldsymbol{Q}, \boldsymbol{K}, \boldsymbol{V}$ of the COPY operation such that

$$\boldsymbol{q}_i = \boldsymbol{Q}\boldsymbol{x}_i^{(2)} = [(n_i^=)^2 - 2n_i^= + 1, \quad 1, \quad n_i^= - 1, \quad (d_i^=)^2 + 2td_i^= + t^2, \quad 1, \quad d_i^= - t]^\top,$$
$$\boldsymbol{k}_j = \boldsymbol{K}\boldsymbol{x}_j^{(2)} = [\quad -1, \quad -(\hat{n}_j^=)^2, \quad 2\hat{n}_j^=, \quad -1, \quad -(\hat{d}_j^=)^2, \quad 2\hat{d}_j^= \,]^\top,$$
$$\boldsymbol{v}_j = e_{\text{id}(s_j)},$$

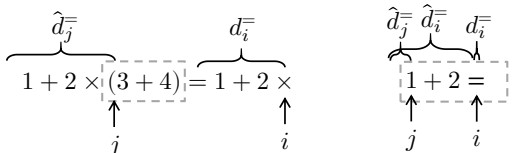

Figure 4: Illustration of the proof of Theorem 3.3.

and

$$\boldsymbol{K}\boldsymbol{x}_i^{(2)} \cdot \boldsymbol{Q}\boldsymbol{x}_j^{(2)} = -(n_i^= - \hat{n}_j^= - 1)^2 - (d_i^= - \hat{d}_j^= + t)^2.$$

Therefore, $\boldsymbol{K}\boldsymbol{x}_i^{(2)} \cdot \boldsymbol{Q}\boldsymbol{x}_j^{(2)} = 0$ only when $\hat{n}_j^= = n_i^= - 1$ and $\hat{d}_j^= = d_i^= + t$, and $\boldsymbol{K}\boldsymbol{x}_i^{(2)} \cdot \boldsymbol{Q}\boldsymbol{x}_j^{(2)} \le -1$ otherwise. It is easy to see that:

- Whenever $n_i^= > 0$, for any $t$, the number of indices $j$ satisfying $\boldsymbol{q}_i \cdot \boldsymbol{k}_j = 0$ is at most one (i.e., unique).

- Whenever $n_i^= > 0$, for any $t$, the index $j$ satisfying $\boldsymbol{q}_i \cdot \boldsymbol{k}_j = 0$ exists, unless there is a $t' < t$ such that the copied token at $t'$ is an equal sign ('=').

In other words, based on Lemma C.7, the above property guarantees that we can copy the desired tokens until reaching an equal sign, after which the copied tokens are invalid as illustrated in Figure 4(right). The output of the attention layer can be written as

$$(\boldsymbol{e}_{\mathrm{id}(s_i)}, \boldsymbol{e}_{j_1}, \boldsymbol{e}_{j_2}, \boldsymbol{e}_{j_3}, \boldsymbol{e}_{j_4}, \boldsymbol{e}_{j_5}, i, n_i^=, (n_i^=)^2, d_i^=, \hat{n}_i^=, (\hat{n}_i^=)^2, \hat{d}_i^=, (\hat{d}_i^=)^2, 1),$$

where we slightly abuse notation and use $\boldsymbol{e}_{j_t}$ to denote the embedding we copied by the $t$-th attention heads.

We can then use an MLP to perform the second task. Note that whether the current position can be calculated or not depends on the following six tokens $(\boldsymbol{e}_{\mathrm{id}(s_i)}, \boldsymbol{e}_{j_1}, \boldsymbol{e}_{j_2}, \boldsymbol{e}_{j_3}, \boldsymbol{e}_{j_4}, \boldsymbol{e}_{j_5})$. Concretely, there are several cases:

- $\boldsymbol{e}_{\mathrm{id}(s_i)}$ corresponds to the embedding of a number or a right bracket. In this case, the current position should simply output $\boldsymbol{e}_{j_1}$ (which is an operator or a right bracket).

- $\boldsymbol{e}_{\mathrm{id}(s_i)}$ corresponds to the embedding of a left bracket, an operator, or the equal sign '='. In this case, $\boldsymbol{e}_{j_1}$ corresponds to the embedding of a number or a left bracket. There are two subcases:

    - $\boldsymbol{e}_{j_1}$ corresponds to the embedding of a number. In this case, whether the current position can be evaluated depends on $(\boldsymbol{e}_{\mathrm{id}(s_i)}, \boldsymbol{e}_{j_1}, \boldsymbol{e}_{j_2}, \boldsymbol{e}_{j_3}, \boldsymbol{e}_{j_4})$ according to Appendix B.
    - $\boldsymbol{e}_{j_1}$ corresponds to the embedding of a left bracket. In this case, whether the current position can be evaluated simply depends on whether $\boldsymbol{e}_{j_5}$ corresponds to the embedding of a right bracket.

When all embeddings $\boldsymbol{e}_{j_t}$ are one-hot vectors, whether the expression at the current position can be calculated or not forms a look-up table. Therefore, it can be implemented by a two-layer MLP with hidden dimension $O(p^6)$ according to Lemma C.5. Similarly, the computed result and how much sentence length will be reduced after this calculation can also be implemented as look-up tables. However, some of the embeddings $\boldsymbol{e}_{j_t}$ may not be one-hot vectors when reaching an equal sign (as discussed above). In this case, we can similarly implement extra look-up tables that take a fewer number of inputs, and the result of which lookup table will be used depends on the position of the equal sign. This corresponds to a multivariate conditional selection operation with multiple Boolean conditions $\mathbb{I}[\boldsymbol{e}_{j_t} \cdot \boldsymbol{e}_{\mathrm{id}(\text{'='})} = 1]$ (for $t \in \{1, 2, 3, 4, 5\}$), which can be similarly implemented by an MLP by extending Lemma C.4. Moreover, we note that the composition of look-up tables and the multivariate conditional selection operation can be merged in just one MLP by following the construction in Lemmas C.4 and C.5 (we omit the details for clarity).

The final output of the third layer is represented by

$$\boldsymbol{x}_i^{(3)} = (\boldsymbol{e}_{j_1}, n_i^=, (n_i^=)^2, d_i^=, \hat{n}_i^=, (\hat{n}_i^=)^2, \hat{d}_i^=, (\hat{d}_i^=)^2, f_i, \boldsymbol{e}_i^{\mathrm{outcome}}, n_i^{\mathrm{reduce}}).$$

Here, $f_i$ is a Boolean value recording whether the next output at position $i$ is a computed value, $\boldsymbol{e}_i^{\mathrm{outcome}}$ is the one-hot embedding of the outcome when $f_i$ is true, and $n_i^{\mathrm{reduce}}$ records the reduced length after calculation when $f_i$ is true. When $f_i$ is false, $\boldsymbol{e}_i^{\mathrm{outcome}}$ and $n_i^{\mathrm{reduce}}$ are undefined.

**Layer 4**. Note that in an arithmetic expression there can be multiple expressions that can be calculated (or *handles* defined in Appendix B), all of which are processed in the last layer. Therefore, the fourth layer of the Transformer should keep only the leftmost calculation and discard other calculations (according to Appendix B). Meanwhile, for subsequent positions $i$ after the position that has been calculated, this layer finds the related token that should be copied for position $i$ based on the reduced setence length. Formally, we need two attention heads to perform the following tasks:

1. Check whether there is an index $j \leq i$ such that $n_j^= = n_i^=$ and $f_j$ is true. Denote the answer as $\hat{f}_i = \sum_{j=i-d_i^=}^{i} f_j$, where $\hat{f}_i \geq 1$ means the answer is yes, and $\hat{f}_i = 0$ otherwise.

2. If the answer is yes ($\hat{f}_i \geq 1$), copy the value $n_j^{\text{reduce}}$ at the leftmost position $j$ satisfying $f_j$ is true and $n_j^= = n_i^=$. Denote the result as $\hat{n}_i^{\text{reduce}} := n_j^{\text{reduce}}$. If $\hat{f}_i = 0$, $\hat{n}_i^{\text{reduce}}$ is undefined.

3. Filter the outcome: if $f_i$ is true, then maintain $e_i^{\text{outcome}}$, otherwise set $e_i^{\text{outcome}}$ to $e_{j_1}$.

Similar to the construction of the third layer, we can construct the matrices $Q$, $K$ and $V$ as follows. For the first task, we leverage the MEAN operation with

$$q_i = Q x_i^{(3)} = (1, (n_i^=)^2, 2n_i^=), \quad k_j = K x_j^{(3)} = (-(n_j^=)^2, -1, n_j^=), \quad v_j = f_j.$$

We have $q_i \cdot k_j = -(n_i^= - n_j^=)^2$. Therefore, $q_i \cdot k_j = 0$ iff $n_j^= = n_i^=$, and $q_i \cdot k_j \leq -1$ otherwise. This attention head thus outputs $\frac{1}{d_i+1} \sum_{j=i-d_i^=}^{i} f_j$. For the second task, we leverage the COPY operation with

$$q_i = Q x_i^{(3)} = (1, (n_i^=)^2, 2n_i^=, 1, 1), \quad k_j = K x_j^{(3)} = (-(n_j^=)^2, -1, n_j^=, f_j, -1), \quad v_j = n_j^{\text{reduce}}.$$

We have $q_i \cdot k_j = -(n_i^= - n_j^=)^2 + f_j - 1$. Therefore, $q_i \cdot k_j = 0$ iff $n_j^= = n_i^=$ and $f_j = 1$, and $q_i \cdot k_j \leq -1$ otherwise. Moreover, we set $r_j = -j$ in the COPY operation, by which the attention head copies $n_j^{\text{reduce}}$ where $j = \min\{j : n_j^= = n_i^=, f_j = 1\}$, as desired. The output of the attention layer has the form

$$\left( e_i^{\text{outcome}}, e_{j_1}, n_i^=, (n_i^=)^2, \hat{n}_i^=, (\hat{n}_i^=)^2, d_i^=, \hat{d}_i^=, (\hat{d}_i^=)^2, f_i, \frac{\hat{f}_i}{d_i^= + 1}, \hat{n}_i^{\text{reduce}} \right).$$

We next use an MLP to perform the third task, which is a conditional selection operation and can be done according to Lemma C.4. We can simultaneously obtain $\hat{f}_i$ by multiplying $\frac{\hat{f}_i}{d_i^= + 1}$ with $(d_i^= + 1)$. We also compute $(d_i^= + \hat{n}_i^{\text{reduce}})^2$, which will be used in the next layer. The final output of the fourth layer is represented by

$$x_i^{(4)} = (\tilde{e}_i^{\text{outcome}}, f_i, n_i^=, (n_i^=)^2, \hat{n}_i^=, (\hat{n}_i^=)^2, \hat{d}_i^=, (\hat{d}_i^=)^2, \hat{f}_i, \hat{n}_i^{\text{reduce}}, d_i^=, (d_i^= + \hat{n}_i^{\text{reduce}})^2),$$

where $\tilde{e}_i^{\text{outcome}}$ is either $e_i^{\text{outcome}}$ or $e_{j_1}$.

**Layer 5**. The final layer of the Transformer uses one attention head to copy the corresponding token for generating the output when $\hat{f}_i \geq 1$ and $f_i$ is false. Similar to previous layers, we can copy the embedding $e_{\text{id}(s_j)}$ located at position $j$ such that $\hat{n}_j^= = n_i^= - 1$ and $\hat{d}_j^= = d_i^= + \hat{n}_i^{\text{reduce}}$. The output of the attention layer is $(\tilde{e}_i^{\text{outcome}}, e_{\text{id}(s_j)}, f_i, \hat{f}_i)$. We then use an MLP to obtain the output: if $\hat{f}_i - f_i \geq 1$, then output $e_{\text{id}(s_j)}$; otherwise output $\tilde{e}_i^{\text{outcome}}$. This corresponds to a conditional selection operation and can be implemented by an MLP according to Lemma C.4. Finally, we pass the output through a softmax layer to generate the next token $s_{i+1}$.

Now it remains to conduct an error analysis and determine the scale of parameters. Note that we can tolerate $O(1)$ error of the final layer output in the sense that the generated token $s_{i+1}$ is still correct. Based on Lemmas C.1, C.4, C.5, C.7 and C.8, we can guarantee that when all parameters of the Transformer are bounded by $O(\text{poly}(n, 1/\epsilon))$, all intermediate neurons will only induce an error below $\epsilon$. (Also note that Assumption C.6 in Lemmas C.7 and C.8 is satisfied when $\epsilon$ is small enough.) Therefore, by picking a fixed small $\epsilon = \Theta(1)$, all parameter values in the Transformer are bounded by $O(\text{poly}(n))$. $\qquad\square$

## D.2 Proof of Theorem 3.1

We now prove that evaluating arithmetic expressions without CoT is extremely difficult for bounded-depth autoregressive Transformers. We will make the widely-believed assumption that $\mathsf{TC}^0 \neq \mathsf{NC}^1$ (see Appendix A.2 for definitions of these complexity classes). We further need the notion of *uniformity*: informally, this condition says that there exists an efficient algorithm to construct the circuits. For a rigorous definition, we refer readers to Arora & Barak [2].

We first present some lemmas on the expressive power of the $\mathsf{TC}^0$ circuits, based on which we will give a reduction proof of Theorem 3.1.

**Lemma D.2.** *For any $n$ $n$-bits binary integers $a_1, a_2, \cdots, a_n$, let $f : \{0,1\}^{n^2} \to \{0,1\}^{2n}$ be a boolean function such that $f(a_1, a_2, \cdots, a_n) = \sum_{i \in [n]} a_i$. Then, $f$ can be implemented by the uniform $\mathsf{TC}^0$ circuits.*

This lemma demonstrates that computing the sum of $n$ $n$-bit integers (known as iterated addition or simply summation) is in uniform $\mathsf{TC}^0$. The detailed proof of this lemma can be found in the previous works [29, 15].

**Lemma D.3.** *Consider any string $s = s_1 s_2 \cdots s_n$ of length $n$ containing brackets '(', ')', and other characters, and all brackets in $s$ are paired. Let $f$ be a boolean function taking $s$ as input and output $n$ pairs of integers defined as follows:*

$$f_i(s) = \begin{cases} (-1, j) & \text{if } s_i \text{ is a left bracket and } s_i, s_j \text{ are paired.} \\ (j, -1) & \text{if } s_i \text{ is a right bracket and } s_i, s_j \text{ are paired.} \\ (j, k) & \text{if } s_i \text{ is not a bracket, and } s_j, s_k \text{ is the nearest pair of matching brackets containing } s_i. \end{cases}$$

*Then $f$ can be implemented by the $\mathsf{TC}^0$ circuits.*

Note that the input characters and the output integers are all encoded in binary. This lemma demonstrates that the task of bracket matching is in $\mathsf{TC}^0$.

*Proof.* Given the input string $s$, we first get the index $i$ of each character $s_i$, which can be hard-coded into the circuits. Then for each character $s_i$, we calculate the following term:

$$r_i = \sum_{j < i} \mathbb{I}[s_j = \text{`('}] - \sum_{j < i} \mathbb{I}[s_j = \text{`)'}] + \mathbb{I}[s_i = \text{`('}].$$

Then, the output of $f_i(s)$ can be written as the following form:

$$f_i(s) = \begin{cases} (-1, \min\{j > i : s_j = \text{`)'}, r_j = r_i\}) & \text{if } s_i = \text{`(',} \\ (\max\{j < i : s_j = \text{`(',} r_j = r_i\}, -1) & \text{if } s_i = \text{`)',} \\ (\max\{j < i : s_j = \text{`(',} r_j = r_i\}, \min\{j > i : s_j = \text{`)'}, r_j = r_i\}) & \text{if } s_i \text{ is not a bracket.} \end{cases}$$

It is simple to verify that the construction can be implemented by $\mathsf{TC}^0$ circuits. $\square$

**Theorem D.4.** *Assume $\mathsf{TC}^0 \neq \text{uniform } \mathsf{NC}^1$. For any prime number $p$, any integer $L$, and any polynomial $Q$, there exists a problem size $n$ such that no log-precision autoregressive Transformer defined in Section 2 with depth $L$ and hidden dimension $d \leq Q(n)$ can solve the problem $\mathsf{Arithmetic}(n, p)$.*

*Proof.* Our proof is based on leveraging the $\mathsf{NC}^1$-completeness of a classic problem: Boolean Formula Evaluation. According to the Buss reduction [12], calculating whether a Boolean formula is true or false is complete for uniform $\mathsf{NC}^1$. Based on this theorem, it suffices to prove that the Boolean Formula Evaluation problem can be *reduced* to evaluating the arithmetic expression. This will yield the conclusion by using the result that bounded-depth log-precision Transformers with polynomial size are in $\mathsf{TC}^0$ [42][4] as well as the assumption that $\mathsf{TC}^0 \neq \text{uniform } \mathsf{NC}^1$.

Formally, let $\Sigma = \{0, 1, \wedge, \vee, \neg, (, )\}$ be the alphabet. A Boolean formula is a string defined on alphabet $\Sigma$ by the following recursive way:

---

[4]While the authors only proved that predicting a binary label using Transformer encoder can be implemented by $\mathsf{TC}^0$ circuits, it is straightforward to extend the result to autoregressive Transformers predicting a label in a finite vocabulary.

- 0 and 1 are Boolean formulae;

- If $\varphi$ is a Boolean formula, then $(\neg\varphi)$ is a Boolean formula;

- If $\varphi_1, \varphi_2$ are two Boolean formulae, then both $(\varphi_1 \wedge \varphi_2)$ and $(\varphi_1 \vee \varphi_2)$ are Boolean formulae.

The Boolean Formula Evaluation problem aims to compute whether a Boolean formula is true (1) or false (0). We now show that we can translate this problem into the problem of evaluating arithmetic expressions. Given a Boolean formula $s$, the translation function $f$ generates the corresponding arithmetic expression $f(s)$ that has the same result as $s$ under evaluation. The translation is recursively defined as follows:

- $f(0) = 0$ and $f(1) = 1$;

- For any Boolean formula $\varphi$, $f(\neg\varphi) = 1 - \varphi$ and $f((\varphi)) = (f(\varphi))$;

- For any Boolean formulae $\varphi_1, \varphi_2$, $f(\varphi_1 \wedge \varphi_2) = f(\varphi_1) \times f(\varphi_2)$;

- For any Boolean formulae $\varphi_1, \varphi_2$, $f(\varphi_1 \vee \varphi_2) = 1 - (1 - f(\varphi_1)) \times (1 - f(\varphi_2))$.

It is easy to see that for any Boolean formula $s$, the length of $f(s)$ is upper-bounded by $O(|s|)$. Moreover, the translation function can be simply implemented using the circuits within $\mathsf{TC}^0$ complexity. To do so, we first replace the symbols $\neg$, $\wedge$, and $\vee$ with $1-$, $\times$, and $\times$, respectively. Furthermore, for each operator $\vee$, we must insert '$1 - (1-$' after the nearest left bracket containing the operator $\vee$, insert a right bracket before $\times$, insert '$(1-$' after $\times$, and insert a right bracket before the nearest right bracket containing the operator $\vee$. According to Lemmas D.2 and D.3, all of these operations can be implemented by the $\mathsf{TC}^0$ circuits. Therefore, this translation function can be simply implemented by $\mathsf{TC}^0$ circuits. Also, note that the above construction does not depend on the modulus $p$. Therefore, by reduction, we obtain that the problem of evaluating arithmetic expressions is $\mathsf{NC}^1$-hard. □

# E   System of Linear Equations

In this section, we will prove that the autoregressive Transformer equipped with CoT can solve a system of linear equations, whereas the autoregressive Transformer without CoT cannot solve it.

## E.1   Proof of Theorem 3.4

For ease of reading, we restate Theorem 3.3 below:

**Theorem E.1.** *For any prime $p$ and integer $m > 0$, there exists an autoregressive Transformer defined in Section 2 with hidden size $d = O(\text{poly}(p))$ (independent of $m$), depth $L = 4$, and 5 heads in each layer that can generate the CoT solution defined in Appendix B for all inputs in $\mathsf{Equation}(m, p)$. Moreover, all parameter values in the Transformer are bounded by $O(\text{poly}(m))$.*

*Proof.* The proof technique is similar to that of Theorem 3.3. We recommend readers to read the proof Theorem 3.3 first as we will omit redundant details in the subsequent proof. Below, without abuse of notation, we use $\boldsymbol{x}_i^{(l)}$ to denote the output at position $i$ after the $l$-th Transformer layer, and use $x_i$ to denote the $i$-th variable in linear equations. We also note that $m$ is the *upper bound* on the number of variables, and we will construct Transformer parameters such that the Transformer can solve all linear equations with the number of variables *no more than* $m$.

**Token Embeddings**. Assume that we have a sequence of tokens $s_1, s_2, \ldots, s_t$ and we want to generate the next token $s_{t+1}$. We can embed the token $s_i$ using the format $\boldsymbol{x}_i^{(0)} = (\boldsymbol{e}_{\text{id}(s_i)}, \boldsymbol{l}_i, i, 1)$:

1. The vector $\boldsymbol{e}_i$ represents the one-hot vector with the $i$-th element being 1, and $\text{id}(s_i)$ is the index of token $s_i$ in the vocabulary. Since we hope the embedding dimension is a constant and does not depend on the number of variable tokens $m$, we consider representing them using a unified (single) encoding and distinguishing them via the term $\boldsymbol{l}_i$. This means that if $s_i$ and $s_j$ are two different variables, we have $\text{id}(s_i) = \text{id}(s_j)$ and $\boldsymbol{l}_i \neq \boldsymbol{l}_j$.

2. $\boldsymbol{l}_i \in \mathbb{R}^3$ is a vector used to distinguish between different variables. Its first element, denoted as $\text{var}(s_i)$, represents the index of the variable $s_i$. If the token $s_i$ is not a variable, then

$l_i = (0, 0, 0)$ and $\text{var}(s_i) = 0$. If it is the variable $x_j$ for some $j \in [m]$, then $\text{var}(s_i) = j$ and $l_i = \left(j, m^2 \sin(\frac{2j\pi}{m}), m^2 \cos(\frac{2j\pi}{m})\right)$.

3. $i$ is the positional embedding, representing the position of the token in the sequence.

4. The constant embedding 1 is used as a bias term.

**Layer 1**. The first layer of the Transformer uses three attention heads to record some basic information:

1. Count the number of ';' (i.e., equations) in previous tokens, denoted as $n_i^{\text{eq}} = |\{j \leq i : s_j = \text{';'}\}|$.

2. Count the number of '$\Longrightarrow$' in previous tokens, denoted as $n_i^{\text{cot}} = |\{j \leq i : s_j = \text{'}\Longrightarrow\text{'}\}|$. Namely, the current position belongs to the $n_i^{\text{cot}}$-th CoT step.

3. Determine the number of variables in the system of linear equations. This can be done by copying $\text{var}(s_j)$ for index $j$ such that $s_j$ is a variable and $\text{var}(s_j)$ is the largest. Denote the result as $n_i^{\text{var}}$. Note that according to the input format, $n_i^{\text{var}}$ is correct whenever $n_i^{\text{eq}} \geq 1$.

Similar to the proof of arithmetic expression, the first and the second tasks can be implemented by two attention heads, which perform the MEAN operation to obtain the fraction of ';' and '$\Longrightarrow$' tokens in all previous tokens. The last attention head perform the COPY operation with $\mathcal{S}_i = \{j : j \leq i : s_j \text{ is a variable}\}$, $r_j = \text{var}(s_j)$, and $v_j = \text{var}(s_j)$. Note that while $r_{j_1} = r_{j_2}$ may hold for different positions $j_1, j_2$, their values are the same (i.e., $v_{j_1} = v_{j_2}$), so the COPY operation still works and obtains $n_i^{\text{var}}$ (when $n_i^{\text{eq}} \geq 1$).

Then, we use MLPs in parallel to calculate $n_i^{\text{eq}} = (n_i^{\text{eq}}/i) \cdot i$ and $n_i^{\text{cot}} = (n_i^{\text{cot}}/i) \cdot i$ based on Lemma C.1. Besides, we use an MLP to compute the auxiliary term $i^2$ that will be used in the next layer. Therefore, the output of the first layer is

$$x_i^{(1)} = (e_{\text{id}(s_i)}, l_i, i, i^2, 1, n_i^{\text{var}}, n_i^{\text{eq}}, n_i^{\text{cot}}).$$

**Layer 2**. As described in Appendix B, each CoT step eliminates one variable, and thus at the current position we are eliminating variable $x_{n_i^{\text{cot}}}$. By the uniqueness of the solution, there must exist an equation with a nonzero coefficient for variable $x_{n_i^{\text{cot}}}$. In the second Transformer layer, we can determine which equation satisfies this condition. More precisely, we record whether the current equation will be used to eliminate the variable $x_{n_i^{\text{cot}}+1}$ in the next CoT step $n_i^{\text{cot}} + 1$. We also use additional attention heads to perform some auxiliary calculations that will be used in subsequent layers. Concretely, the second layer uses four attention heads to perform the following tasks:

1. Copy the value $n_j^{\text{eq}}$ with position $j$ corresponding to the nearest '$\Longrightarrow$' token $s_j$ ($j \leq i$). Clearly, the value is well-defined when $n_i^{\text{cot}} \geq 1$, and we define the value to be 0 if $n_i^{\text{cot}} = 0$.

2. Compute $d_i^{\text{eq}} = n_i^{\text{eq}} - n_j^{\text{eq}} + 1$, which corresponds to the index of the current equation in the current CoT step.

3. Copy the embedding $e_{\text{id}(s_j)}$ with the smallest $j$ satisfying $n_j^{\text{eq}} = n_i^{\text{eq}}$ and $s_j$ is a number. Note that $e_{\text{id}(s_j)}$ is well-defined when $s_i = \text{'='}$.

4. Compute a Boolean flag (denoted as $f_i$), which is true only when $e_{\text{id}(s_j)} \neq e_{\text{id}(0)}$, $d_i^{\text{eq}} > n_i^{\text{cot}}$, and $s_i = \text{'='}$. The definition of $f_i$ means that in the $n_i^{\text{cot}}$-th CoT step, we only focus on the $j$-th equation when $j > n_i^{\text{cot}}$ and check whether the first number in the equation is non-zero. If it is non-zero, we set the flag to true at the specific position corresponding to token '='.

5. Copy the embeddings $(e_{\text{id}(s_{i-1})}, l_{i-1})$ and $(e_{\text{id}(s_{i-2})}, l_{i-2})$ of the $(i-1)$-th and $(i-2)$-th token.

The first task can be implemented by an attention head via the COPY operation to obtain $n_j^{\text{eq}}$ when $n_i^{\text{cot}} \geq 1$. For the third task, we construct the matrices $Q, K, V$ of the COPY operation such that

$$q_i = Qx_i^{(1)} = (-n_i^{\text{eq}}, 1, 1, 1), \quad k_j = Kx_j^{(1)} = \left(1, n_j^{\text{eq}}, \sum_{a \in [p]} \mathbb{I}[s_j = a], -1\right),$$

$v_j = e_{\text{id}(s_j)}$, and $r_j = -j$. By construction, $q_i \cdot k_j = (n_j^{\text{eq}} - n_i^{\text{eq}}) + \sum_{a \in [p]} \mathbb{I}[s_j = a] - 1$, and thus $q_i \cdot k_j = 0$ only when $n_j^{\text{eq}} = n_i^{\text{eq}}$ and $s_j$ is a number, and $q_i \cdot k_j \leq -1$ otherwise. Furthermore, the

choice of $r_j$ guarantees that the leftmost position satisfies $\boldsymbol{q}_i \cdot \boldsymbol{k}_j = 0$ is copied. This exactly solves the third task. For the fifth task, we use two attention heads to perform the COPY operation. We only give the construction of the first head that copies $(\boldsymbol{e}_{\text{id}(s_{i-1})}, \boldsymbol{l}_{i-1})$. The matrices $\boldsymbol{Q}, \boldsymbol{K}, \boldsymbol{V}$ of the COPY operation is constructed such that

$$\boldsymbol{q}_i = \boldsymbol{Q}\boldsymbol{x}_i^{(1)} = ((i-1)^2, i-1, -1), \quad \boldsymbol{k}_j = \boldsymbol{K}\boldsymbol{x}_j^{(1)} = (-1, 2j, j^2), \quad \boldsymbol{v}_j = (\boldsymbol{e}_{\text{id}(s_j)}, \boldsymbol{l}_j),$$

and $\boldsymbol{q}_i \cdot \boldsymbol{k}_j = 0$ iff $j = i - 1$.

We next use an MLP to correct the value of $n_j^{\text{eq}}$ when $n_i^{\text{cot}} = 0$ and compute the second task, which is a linear operation. We also compute an auxiliary flag $\mathbb{I}[n_i^{\text{cot}} = d_i^{\text{eq}}]$ via an MLP. Regarding the fourth task, it is a multivariate conditional selection operation and can be similarly implemented by an MLP by extending Lemma C.4. Note that we can compute the second task and the fourth task *in parallel* using a two-layer MLP because both tasks correspond to (multivariate) conditional selection and can be merged. We finally use multiplication to compute the auxiliary terms $(n_i^{\text{cot}})^2$, $(\text{var}(s_i))^2$ and $(\text{var}(s_{i-1}))^2$. The output of the MLP is

$$\boldsymbol{x}_i^{(2)} = (\boldsymbol{e}_{\text{id}(s_i)}, \boldsymbol{l}_i, i, i^2, 1, n_i^{\text{var}}, n_i^{\text{cot}}, (n_i^{\text{cot}})^2, d_i^{\text{eq}}, f_i,$$
$$\boldsymbol{e}_{\text{id}(s_{i-1})}, \boldsymbol{l}_{i-1}, \boldsymbol{e}_{\text{id}(s_{i-2})}, \boldsymbol{l}_{i-2}, (\text{var}(s_{i-1}))^2, (\text{var}(s_i))^2, \mathbb{I}[n_i^{\text{cot}} = d_i^{\text{eq}}]).$$

**Layer 3**. The third layer of the Transformer uses two attention heads to perform the following tasks:

1. Copy the embedding $d_j^{\text{eq}}$ with the smallest $j$ satisfying $f_j = 1$ and $n_j^{\text{cot}} = n_i^{\text{cot}} - 1$. Denote the answer as $\hat{d}_i^{\text{eq}}$.

2. Determine whether the next token $s_{i+1}$ is a number. Denote the result as $f_i^{\text{num}}$.

3. Determine the output of the next token $s_{i+1}$ if $s_{i+1}$ is not a number. We denote its embedding as $\boldsymbol{e}_i^{\text{next}}$. Also, we need to determine the variable index $\text{var}(s_{i+1})$ of the next token if the next token is a variable.

4. Determine the token $s_{i+2}$ if the next token $s_{i+1}$ is a number. There are two cases: $s_{i+2}$ is a variable, and $s_{i+2}$ is the token ';'. Denote the result as $\boldsymbol{e}_i^{\text{next2}}$ and $\text{var}(s_{i+2})$ and compute $(\text{var}(s_{i+2}))^2$.

5. If the current token $s_i$ is a variable, copy the embedding $\boldsymbol{e}_{\text{id}(s_{j-1})}$ (which is a number) for index $j$ satisfying $n_j^{\text{cot}} = n_i^{\text{cot}}$, $n_j^{\text{cot}} = d_j^{\text{eq}}$, and $\text{var}(s_j) = \text{var}(s_i)$. Denote the answer as $\boldsymbol{e}_i^{\text{cot\_num}}$. When $s_i$ is not a variable or $d_i^{\text{eq}} \leq n_i^{\text{cot}}$, $\boldsymbol{e}_i^{\text{cot\_num}}$ is undefined.

We can use an attention head to perform the COPY operation that completes the first task. The construction is similar to the fourth layer in arithmetic expression and we omit it for clarity. The second attention head performs the fifth task, which can also be done via the COPY operation. Regarding the second task, whether the next token is a number can be purely determined by $d_i^{\text{eq}}$, $n_i^{\text{cot}}$, and the current token $s_i$. Specifically, $s_{i+1}$ is a number if $s_i = $ '+', or $s_i = $ '=', or ($s_i = $ ';' and $d_i^{\text{eq}} > n_i^{\text{cot}}$). Whether the output of the next token is a variable can also be purely determined by the previous tokens $s_{i-1}, s_i$ and also $d_i^{\text{eq}}$ and $n_i^{\text{cot}}$. Specifically, $s_{i+1}$ is a variable if $s_{i-1} = $ '+' and $s_i$ is a number, or $s_{i-1} = $ ';' and $s_i$ is a number, or ($s_i = $ ';' or $s_i = $ ' $\implies$ ') and $d_i^{\text{eq}} \leq n_i^{\text{cot}}$. The variable index can be determined by either $\text{var}(s_{i-2})$ or $d_i^{\text{eq}}$. When the next token is neither a variable nor a number (i.e., the symbols '+', '=', ';', or '$\implies$', we can similarly determine the token by checking $s_{i-1}, s_i, d_i^{\text{eq}}$, and $n_i^{\text{var}}$. When the next token is a number, $s_{i+2}$ can be determined by checking the variable $s_{i-1}$ via three cases: (i) if $s_{i-1}$ is a variable and $\text{var}(s_{i-1}) < n_i^{\text{var}}$, then $s_{i+2}$ is a variable and $\text{var}(s_{i+2}) = \text{var}(s_{i-1}) + 1$; (ii) if $s_{i-1}$ is a variable and $\text{var}(s_{i-1}) = n_i^{\text{var}}$, then $s_{i+2} = $ ';'; (iii) otherwise, $s_{i-1}$ is a number, then $s_{i+2}$ is a variable and $\text{var}(s_{i+2}) = n_i^{\text{cot}} + 1$.

All these tasks can implemented by MLPs that performs the conditional selection or look-up table based on Lemmas C.4 and C.5. Moreover, the composition of conditional selection and look-up table can be merged into a single two-layer MLP (as shown in the construction of the third layer in arithmetic expression). We next use multiplication to compute the auxiliary terms $(d_i^{\text{eq}})^2$, $(d_i^{\text{eq}} + \mathbb{I}[s_{i+2} = ';'])^2$, and $(\hat{d}_i^{\text{eq}})^2$. However, to compute $(\text{var}(s_{i+2}))^2$, we cannot use multiplication directly as the composition of multiplication and conditional selection will require a deeper MLP. Instead, note that $(\text{var}(s_{i+2}))^2$ linearly depends on $(\text{var}(s_{i-1}))^2$ and $\text{var}(s_{i-1})$, or linearly depends on $(n_i^{\text{cot}})^2$ and $n_i^{\text{cot}}$, all of which is already computed. Therefore, we can compute $(\text{var}(s_{i+2}))^2$

without multiplication. The output of this layer has the form

$$\boldsymbol{x}_i^{(3)} = (\boldsymbol{e}_{\mathrm{id}(s_i)}, \boldsymbol{l}_i, i, 1, n_i^{\mathrm{var}}, n_i^{\mathrm{cot}}, (n_i^{\mathrm{cot}})^2, d_i^{\mathrm{eq}}, (d_i^{\mathrm{eq}})^2, (d_i^{\mathrm{eq}} + \mathbb{I}[s_{i+2} = \text{`;'}])^2, \hat{d}_i^{\mathrm{eq}}, (\hat{d}_i^{\mathrm{eq}})^2, f_i^{\mathrm{num}},$$
$$\boldsymbol{e}_i^{\mathrm{next}}, \boldsymbol{e}_i^{\mathrm{next2}}, \mathrm{var}(s_i), \mathrm{var}(s_{i+1}), \mathrm{var}(s_{i+2}), (\mathrm{var}(s_i))^2, (\mathrm{var}(s_{i+2}))^2, \boldsymbol{e}_{\mathrm{id}(s_{i-1})}, \boldsymbol{e}_i^{\mathrm{cot\_num}}).$$

**Layer 4**. The fourth layer of the Transformer performs the the core calculation of equation coefficients when the next token is a number. There are two equations related to the calculation: the $d_i^{\mathrm{eq}}$-th equation in the last CoT step, and the $\hat{d}_i^{\mathrm{eq}}$-th equation in the last CoT step. There are also two variables related to the calculation: the variable $x_{\mathrm{var}(s_{i+2})}$ and $x_{n_i^{\mathrm{cot}}}$. Specifically, we need to copy four coefficients $a_{\hat{d}_i^{\mathrm{eq}}, n_i^{\mathrm{cot}}}, a_{\hat{d}_i^{\mathrm{eq}}, \mathrm{var}(s_{i+2})}, a_{d_i^{\mathrm{eq}}, n_i^{\mathrm{cot}}}, a_{d_i^{\mathrm{eq}}, \mathrm{var}(s_{i+2})}$ defined as follows:

The $\hat{d}_i^{\mathrm{eq}}$-th equation: $\quad \cdots + a_{\hat{d}_i^{\mathrm{eq}}, n_i^{\mathrm{cot}}} x_{n_i^{\mathrm{cot}}} + \cdots + a_{\hat{d}_i^{\mathrm{eq}}, \mathrm{var}(s_{i+2})} x_{\mathrm{var}(s_{i+2})} + \cdots = b_{\hat{d}_i^{\mathrm{eq}}}$
The $d_i^{\mathrm{eq}}$-th equation: $\quad \cdots + a_{d_i^{\mathrm{eq}}, n_i^{\mathrm{cot}}} x_{n_i^{\mathrm{cot}}} + \cdots + a_{d_i^{\mathrm{eq}}, \mathrm{var}(s_{i+2})} x_{\mathrm{var}(s_{i+2})} + \cdots = b_{d_i^{\mathrm{eq}}}$

For the case of $s_{i+2} = \text{`;'}$, we need to copy coefficients $b_{\hat{d}_i^{\mathrm{eq}}}$ and $b_{d_i^{\mathrm{eq}}}$. To unify the two cases, this Transformer layer uses four attention heads to perform the following tasks (note that we define $\mathrm{var}(s_j) = 0$ when $s_j$ is not a variable):

1. Copy the embedding $\boldsymbol{e}_{\mathrm{id}(s_{j-1})}$ for position $j$ satisfying $n_j^{\mathrm{cot}} = n_i^{\mathrm{cot}} - 1$, $d_j^{\mathrm{eq}} = \hat{d}_i^{\mathrm{eq}}$, $s_j$ is a variable, and $\mathrm{var}(s_j) = n_i^{\mathrm{cot}}$.

2. Copy the embedding $\boldsymbol{e}_{\mathrm{id}(s_{j-1})}$ for position $j$ satisfying $n_j^{\mathrm{cot}} = n_i^{\mathrm{cot}} - 1$, $d_j^{\mathrm{eq}} = \hat{d}_i^{\mathrm{eq}} + \mathbb{I}[s_{i+2} = \text{`;'}]$, $\boldsymbol{e}_{\mathrm{id}(s_j)} = \boldsymbol{e}_i^{\mathrm{next2}}$, and $\mathrm{var}(s_j) = \mathrm{var}(s_{i+2})$.

3. Copy the embeddings $\boldsymbol{e}_{\mathrm{id}(s_{j-1})}$ and $\boldsymbol{e}_j^{\mathrm{cot\_num}}$ for position $j$ satisfying $n_j^{\mathrm{cot}} = n_i^{\mathrm{cot}} - 1$, $d_j^{\mathrm{eq}} = d_i^{\mathrm{eq}}$, $s_j$ is a variable, and $\mathrm{var}(s_j) = n_i^{\mathrm{cot}}$.

4. Copy the embedding $\boldsymbol{e}_{\mathrm{id}(s_{j-1})}$ and $\boldsymbol{e}_j^{\mathrm{cot\_num}}$ for position $j$ satisfying $n_j^{\mathrm{cot}} = n_i^{\mathrm{cot}} - 1$, $d_j^{\mathrm{eq}} = d_i^{\mathrm{eq}} + \mathbb{I}[s_{i+2} = \text{`;'}]$, $\boldsymbol{e}_{\mathrm{id}(s_j)} = \boldsymbol{e}_i^{\mathrm{next2}}$, and $\mathrm{var}(s_j) = \mathrm{var}(s_{i+2})$.

Note that for each task, there is exactly one index $j$ satisfying the condition, and thus the copied embeddings contain the four coefficients defined above. Then, we can use an MLP to compute the desired output $a_{d_i^{\mathrm{eq}}, \mathrm{var}(s_{i+2})} - a_{\hat{d}_i^{\mathrm{eq}}, \mathrm{var}(s_{i+2})} / a_{\hat{d}_i^{\mathrm{eq}}, n_i^{\mathrm{cot}}} \cdot a_{d_i^{\mathrm{eq}}, n_i^{\mathrm{cot}}}$ (or $b_{d_i^{\mathrm{eq}}} - b_{\hat{d}_i^{\mathrm{eq}}} / a_{\hat{d}_i^{\mathrm{eq}}, n_i^{\mathrm{cot}}} \cdot a_{d_i^{\mathrm{eq}}, n_i^{\mathrm{cot}}}$), which can be implemented as a look-up table (according to Lemma C.5). However, there are several special cases we have to consider:

- $d_i^{\mathrm{eq}} = n_i^{\mathrm{cot}}$. In this case, the coefficient is simply computed by normalizing the $\hat{d}_i^{\mathrm{eq}}$-th equation, which can also be implemented via a look-up table.

- $d_i^{\mathrm{eq}} = \hat{d}_i^{\mathrm{eq}}$ and $\hat{d}_i^{\mathrm{eq}} \neq n_i^{\mathrm{cot}}$. In this case, the $\hat{d}_i^{\mathrm{eq}}$-th equation and the $n_i^{\mathrm{cot}}$-th equation are swapped according to Appendix B, and the coefficient should be instead computed by $a_{n_i^{\mathrm{cot}}, \mathrm{var}(s_{i+2})} - a_{\hat{d}_i^{\mathrm{eq}}, \mathrm{var}(s_{i+2})} / a_{\hat{d}_i^{\mathrm{eq}}, n_i^{\mathrm{cot}}} \cdot a_{n_i^{\mathrm{cot}}, n_i^{\mathrm{cot}}}$. Fortunately, the embeddings $\boldsymbol{e}_j^{\mathrm{cot\_num}}$ in the third and the fourth tasks contain exactly $a_{n_i^{\mathrm{cot}}, \mathrm{var}(s_{i+2})}$ and $a_{n_i^{\mathrm{cot}}, n_i^{\mathrm{cot}}}$.

Overall, the coefficient can be computed by a composition of look-up tables and (multivariate) conditional selection operations, which can be merged in a single two-layer MLP.

Now two more things remain to be done. The first is to obtain the 3-dimensional embedding $\boldsymbol{l}_{i+1}$ when $s_{i+1}$ is a variable, while currently we have only obtained $\mathrm{var}(s_{i+1})$. However, we cannot compute the remaining two dimensions $m^2 \sin(\frac{2\mathrm{var}(s_{i+1})\pi}{m})$ and $m^2 \cos(\frac{2\mathrm{var}(s_{i+1})\pi}{m})$ since we do not assume that the MLP can approximate sin and cos functions. Nevertheless, this can be done by directly copying the embedding $\boldsymbol{l}_j$ for any $j$ such that $s_j$ is the variable $x_{\mathrm{var}(s_{i+1})}$ by using an attention head. Finally, the output is conditioned on the flag $f_i^{\mathrm{num}}$: when $f_i^{\mathrm{num}}$ is true, this layer outputs the computed coefficient embedding; otherwise, it outputs $\boldsymbol{e}_i^{\mathrm{next}}$ and $\boldsymbol{l}_{i+1}$. We denote the output of this layer as $\boldsymbol{x}_i^{(4)} = (\boldsymbol{e}_i^{\mathrm{out}}, \boldsymbol{l}_i^{\mathrm{out}})$.

**Linear projection and softmax layer**. Finally, we pass it through a softmax layer to predict the next token $s_{i+1}$. Unlike the proof of arithmetic expression, here the embedding $\boldsymbol{l}_i^{\mathrm{out}}$ is not one-hot (which contains $\mathrm{var}(s_{i+1})$), so we need to additionally prove the following result: let the output logit corresponding to token $t$ (before softmax) be $z_t(\boldsymbol{e}, \boldsymbol{l}) = \boldsymbol{w}_t \cdot [\boldsymbol{e}^\top, \boldsymbol{l}^\top]^\top + b_t$, where $\boldsymbol{w}_t$ and $b_t$ are

parameters of the linear projection for logit $t$. Then, there exist parameters $\{\boldsymbol{w}_t, b_t\}_t$ such that for any two tokens $t$ and $\tilde{t}$ with $t \neq \tilde{t}$

$$
\begin{aligned}
\text{Gap} :=& z_t\left(\boldsymbol{e}_{\text{id}(t)}, \text{var}(t), m^2 \sin\left(\frac{2\text{var}(t)\pi}{m}\right), m^2 \cos\left(\frac{2\text{var}(t)\pi}{m}\right)\right) - \\
& z_t\left(\boldsymbol{e}_{\text{id}(\tilde{t})}, \text{var}(\tilde{t}), m^2 \sin\left(\frac{2\text{var}(\tilde{t})\pi}{m}\right), m^2 \cos\left(\frac{2\text{var}(\tilde{t})\pi}{m}\right)\right) \geq \Theta(1).
\end{aligned}
$$

To prove the above result, simply set $\boldsymbol{w}_t = \left(\boldsymbol{e}_{\text{id}(t)}, \text{var}(t), m^2 \sin\left(\frac{2\text{var}(t)\pi}{m}\right), m^2 \cos\left(\frac{2\text{var}(t)\pi}{m}\right)\right)$. We have

$$
\begin{aligned}
\text{Gap} =& 1 + (\text{var}(t))^2 + m^4 - \mathbb{I}[\text{id}(t) = \text{id}(\tilde{t})] - \text{var}(t)\text{var}(\tilde{t}) \\
& - m^4\left(\sin\left(\frac{2\text{var}(t)\pi}{m}\right)\sin\left(\frac{2\text{var}(\tilde{t})\pi}{m}\right) + \cos\left(\frac{2\text{var}(t)\pi}{m}\right)\cos\left(\frac{2\text{var}(\tilde{t})\pi}{m}\right)\right) \\
=& (1 - \mathbb{I}[\text{id}(t) = \text{id}(\tilde{t})]) + \text{var}(t)(\text{var}(t) - \text{var}(\tilde{t})) + m^4\left(1 - \cos\left(\frac{2(\text{var}(t) - \text{var}(\tilde{t}))\pi}{m}\right)\right)
\end{aligned}
$$

When $\text{var}(t) = \text{var}(\tilde{t})$, we have $\text{id}(t) \neq \text{id}(\tilde{t})$ and thus $\text{Gap} = 1$. Otherwise,

$$
\begin{aligned}
\text{Gap} &\geq 1 - m^2 + m^4\left(1 - \cos(2\pi/m)\right) \\
&= 1 - m^2 + m^4 \sin^2(\pi/m) \geq 1,
\end{aligned}
$$

where we use the fact that $\sin(x) \geq x/\pi$ whenever $0 < x \leq \pi/2$.

Now it remains to conduct an error analysis and determine the scale of parameters. Similar to the proof of arithemetic expression, we can prove that all parameter values in the Transformer are bounded by $O(\text{poly}(n))$. $\qquad\square$

## E.2    Proof of Theorem 3.2

We will now prove that solving a system of linear equations without CoT is extremely difficult for bounded-depth autoregressive Transformers.

**Theorem E.2.** *Assume* $\mathsf{TC}^0 \neq \mathsf{NC}^1$. *For any prime number $p$, any integer $L$, and any polynomial $Q$, there exists a problem size $m$ such that no log-precision autoregressive Transformer defined in Section 2 with depth $L$ and hidden dimension $d \leq Q(m)$ can solve the problem* $\mathsf{Equation}(m, p)$.

*Proof.* Our proof is based on leveraging the $\mathsf{NC}^1$-completeness of a classic problem: Unsolvable Automaton Membership Testing. According to Barrington's theorem [4, 5], given a fixed *unsolvable* automaton, judging whether the automaton accepts an input is complete in $\mathsf{NC}^1$. Below, we will prove that solving the system of linear equations is $\mathsf{NC}^1$-hard by demonstrating that the Unsolvable Automaton Membership Testing problem is $\mathsf{NC}^0$ reducible to the problem of solving a system of linear equations. This will yield the conclusion since bounded-depth log-precision Transformers with polynomial size are in $\mathsf{TC}^0$ [42].

Let $D = (\mathcal{Q}, \Sigma, \delta, \mathcal{F}, q_0)$ be any automaton, where $\mathcal{Q}$ is a set of states, $\Sigma$ is a set of symbols (alphabet), $\delta : \mathcal{Q} \times \Sigma \to \mathcal{Q}$ is the transition function, $\mathcal{F} \subset \mathcal{Q}$ is a set of accept states, and $q_0$ is the initial state. For any input string $\omega_1\omega_2\cdots\omega_n$, whether $D$ accepts the string can be reduced into solving a system of linear equations defined as follows. The system of linear equations has $(n+1)|\mathcal{Q}| + 1$ variables, which we denote as $x^*$ and $x_{i,q}$ ($i \in \{0, \cdots, n\}, q \in \mathcal{Q}$). The equations are defined as follows:

$$
\begin{cases}
x^* = \sum_{q \in \mathcal{F}} x_{n,q} \\
x_{0,q_0} = 1 \\
x_{0,q} = 0 & \text{for } q \in \mathcal{Q}\backslash\{q_0\} \\
x_{i,q} = \sum_{\delta(r,\omega_i)=q} x_{i-1,r} & \text{for } 0 < i \leq n, q \in \mathcal{Q}
\end{cases}
$$

It is easy to see that $x_{i,q} = 1$ iff the automaton arrives at state $q$ when taking the substring $\omega_1\omega_2\cdots\omega_i$ as input. Therefore, $x^* = 1$ iff the automaton accepts the input string. Note that the above solution does not depend on the modulus $p$, and the solution of these equations always exists and is unique.

Furthermore, the coefficient of each equation only depends on at most one input symbol. This implies that these equations can be efficiently constructed using a highly parallelizable algorithm within a complexity of $\mathsf{NC}^0$. Therefore, by reduction, we obtain that the problem of judging whether there exists a solution such that $x^* = 1$ is $\mathsf{NC}^1$-hard.

Now consider solving linear equations using a Transformer without CoT. While the output of the Transformer contains multiple tokens, we can arrange the order of variables such that the Transformer has to output the value of $x^*$ first. The parallel complexity of outputting the first token is bounded by $\mathsf{TC}^0$ according to [42]. Therefore, it cannot judge whether there exists a solution satisfying $x^* = 1$. $\qquad\square$

## F  Discussion on Other Architectures

### F.1  Encoder-Decoder Transformer

In this paper, we choose the autoregressive Transformer as it is simple and the de facto standard for LLMs (e.g., GPT). However, all of the theoretical results of this paper can be easily transferred to an encoder-decoder Transformer (such as T5) using the following arguments.

- Given a bounded-depth polynomial-size log-precision encoder-decoder Transformer, its computation complexity is still bounded by $\mathsf{TC}^0$, so our negative results hold.
- Any finite-depth autoregressive Transformer can be mimicked by a finite-depth encoder-decoder Transformer of roughly the same size, since causal masking for the input sequence can be mimicked through the use of positional encoding and joint attention can be mimicked by the integration of cross-attention and self-attention. Thus, all positive results in this paper are not exclusive to autoregressive Transformers.

### F.2  Recurrent Neural Network(RNN)

Although the theorems in this paper can be naturally extended to other popular Transformer architectures, such as the encoder-decoder Transformer (e.g., T5), RNNs do not have this property. Theorem F.1 shows that RNNs cannot generate the CoT sequence using the same format proposed in our paper for the arithmetic formula task and the linear equation task unless the hidden dimension of the RNN is at least $\Omega(\frac{n}{\log n})$, where $n$ is the length of the input sequence.

**Theorem F.1.** *For any integer $p$, any log-precision RNN with constant layers can neither generate the CoT solution for the problem* $\mathsf{Arithmetic}(n,p)$ *with hidden dimension of* $o(\frac{n}{\log n})$ *nor generate the CoT solution for the problem* $\mathsf{Eqution}(n,p)$ *with the hidden dimension of* $o(\frac{n^2}{\log n})$.

*Proof.* Given an RNN model, if the hidden embeddings in each layer of the last symbols of two different input sequences are the same, the two output sequences will also be exactly the same. When the RNN finishes processing the input sequence, the input sequence has been compressed into a hidden state of $O(D \log n)$ bits, where $D$ is the hidden dimension and each scalar is represented by $O(\log n)$ bits (by definition of log-precision). Therefore, an RNN with a hidden dimension of $D$ can only generate at most $n2^D$ different output sequences. However, for the problem $\mathsf{Arithmetic}(n,p)$, the first step of the CoT needs to output a sequence of length $O(n)$, which contains $O(n)$ numbers. Therefore, there are at least $O(2^{O(n)})$ different solution sequences for the problem $\mathsf{Arithmetic}(n,p)$. Similarly, there are at least $O(2^{O(n^2)})$ different solution sequences for the problem $\mathsf{Eqution}(n,p)$. Thus, by the Pigeon Hole Principle, to be able to generate all $2^{O(n)}$ different output sequences, we must have a hidden dimension of $D = \Omega(\frac{n}{\log n})$ for the problem $\mathsf{Arithmetic}(n,p)$ and a hidden dimension of $D = \Omega(\frac{n^2}{\log n})$ for the problem $\mathsf{Eqution}(n,p)$. $\qquad\square$

Due to the great difference in the model architecture between the RNN and the transformer, most of the theorems in this paper cannot be simply generalized to the RNN. Moreover, To the best of our knowledge, there is very little work to investigate the chain of thought in RNN model, either from the theoretical or the empirical aspect.

# G   Dynamic Programming

## G.1   Examples

**Longest Increasing Subsequence (LIS)**. The LIS problem aims to compute the length of the longest increasing subsequence given an input sequence $s \in \mathbb{N}^n$. Formally, $\tilde{s}$ is a subsequence of $s$ if there exists indices $1 \le i_1 \le i_2 \le \cdots \le i_{|\tilde{s}|} \le n$ such that $\tilde{s}_k = s_{i_k}$ holds for all $k \in [|\tilde{s}|]$. A sequence $\tilde{s}$ is called increasing if $\tilde{s}_1 < \tilde{s}_2 < \cdots < s_{|\tilde{s}|}$. The LIS problem aims to find an increasing subsequence of $s$ with maximal length. A standard DP solution is to compute the length of the longest increasing subsequence that ends at each position $i$, which we denote as $\mathsf{dp}(i)$. It is easy to write the transition function as follows:

$$\mathsf{dp}(i) = 1 + \max_{j < i, s_j < s_i} \mathsf{dp}(j). \tag{20}$$

The final answer will be $\max_{i \in [n]} \mathsf{dp}(i)$.

However, the above DP transition function does not match the form of (5), since $\mathsf{dp}(i)$ may depend on (an unbounded number of) all previous $\mathsf{dp}(j)$ $(j < i)$. Nevertheless, this issue can be easily addressed by using a different DP formulation. Let $\mathsf{dp}(j, k)$ be the longest increasing subsequence that ends at position $j$ and the second last position is no more than $k$ $(k < j)$. In this case, it is easy to write the transition function as follows:

$$\mathsf{dp}(j, k) = \begin{cases} 1 & \text{if } k = 0 \\ \max(\mathsf{dp}(j, k-1), \mathsf{dp}(k, k-1) \cdot \mathbb{I}[s_j > s_k] + 1) & \text{if } k > 0 \end{cases} \tag{21}$$

The final answer will be $\max_{i \in [n]} \mathsf{dp}(i, i-1)$. This DP formulation fits our framework (5).

**Edit Distance (ED)**. The ED problem aims to find the minimum operation cost required to convert a sequence $u \in \Sigma^{n_1}$ to another sequence $v \in \Sigma^{n_2}$. There are three types of operations: *inserting* a letter into any position, *deleting* a letter from any position, and *replacing* a letter at any position by a new one. The costs of insert, delete, and replace are $a$, $b$, and $c$, respectively. These operations are sequentially executed and the total operation cost is the summation of all costs of individual operations.

A standard DP solution is to compute the minimum operation cost to convert the substring $u_1 u_2 \cdots u_j$ to the substring $v_1 v_2 \cdots v_k$, which we denote as $\mathsf{dp}(j, k)$. It is easy to write the transition function as follows:

$$\mathsf{dp}(j, k) = \begin{cases} ak & \text{if } j = 0 \\ bj & \text{if } k = 0 \\ \min(\mathsf{dp}(j, k-1) + a, \mathsf{dp}(j-1, k) + b, \mathsf{dp}(j-1, k-1) + c\mathbb{I}[s_j^{(1)} \ne s_k^{(2)}]) & \text{otherwise} \end{cases} \tag{22}$$

The final answer will be $\mathsf{dp}(n_1, n_2)$. This DP formulation fits our framework (5).

**CFG Membership Testing**.   A context-free grammar (CFG) is a 4-tuple, denoted as $G = (\mathcal{V}, \Sigma, R, S)$, where

- $\mathcal{V}$ is a finite set of non-terminal symbols.
- $\Sigma$ is a finite set of terminal symbols, disjoint from $\mathcal{V}$.
- $R$ is a finite set of production rules, where each rule has the form $A \to \beta$, with $A \in \mathcal{V}$ and $\beta \in (V \cup \Sigma)^*$ (the asterisk represents the Kleene star operation).
- $S \in \mathcal{V}$ is the start symbol.

The non-terminal symbols in $\mathcal{V}$ represent (abstract) syntactic categories, while the terminal symbols in $\Sigma$ represent the actual words/tokens of the language. The production rules in $R$ specify how a non-terminal symbol can be replaced by sequences of terminal and non-terminal symbols concatenated together. Below, we focus on a canonical version of CFG [33], in which the term $\beta$ in each rule is either an empty string or contains exactly two symbols:

- $A \to \epsilon$, where $A \in \mathcal{V}$ and $\epsilon$ is the empty string;
- $A \to BC$, where $B, C \in \mathcal{V} \cup \Sigma$.

By introducing additional non-terminal symbols to split complex rules, any CFG can be easily translated into the canonical version expressing the same language.

The (universal) CFG Membership Testing problem is defined as follows: given a canonical CFG $G$ and a string $\boldsymbol{v}$, judge whether $\boldsymbol{v}$ can be generated from $G$. This problem is known to be P-complete [33]. The CYK algorithm [53], which is based on DP, is a classic algorithm to solve the CFG Membership Testing problem. Here, we consider a variant of the CYK algorithm that can also handle rules of the form $A \to \epsilon$. Denote $R = \{R_1, \cdots, R_m\}$ and let $n = |\boldsymbol{v}|$. The state space of the DP process is defined as

$$\mathcal{I}_{|\mathcal{V}|,m,n} = \{(t, i, j, k, A, r) : 0 \le t \le |\mathcal{V}|, 0 \le i \le k \le j \le n, A \in \mathcal{V}, 0 \le r \le m\},$$

where $\mathsf{dp}(t, i, j, k, A, r)$ stores whether the substring $v_{i+1} \cdots v_j$ can be generated by nonterminal $A$ by first using rule $R_{\tilde{r}} : A \to BC$ with $\tilde{r} \le r$, and there exists index $\tilde{k} \le k$ such that the substring $v_{i+1} \cdots v_{\tilde{k}}$ can be generated by $B$ and the substring $v_{\tilde{k}+1} \cdots v_j$ can be generated by $C$. The index $t$ represents the number of iterations (which will be detailed later). For the boundary setting when $i = j = k$, $\mathsf{dp}(0, i, i, i, A, r)$ simply stores whether nonterminal $A$ can generate an empty string by first using rule $R_{\tilde{r}}$ with $\tilde{r} \le r$. The transition function can be formally written as

$\mathsf{dp}(t, i, j, k, A, r) =$

$$\begin{cases} 0 & \text{if } i = j = k, t = 0, r = 0, \\ \mathbb{I}\left[\mathsf{dp}(t, i, j, k, A, r - 1) = 1 \text{ or } R_r : A \to \epsilon\right] & \text{if } i = j = k, t = 0, r \ge 1, \\ 0 & \text{if } i < j, t = 0, \\ \mathsf{dp}(t - 1, i, j, j, A, m) & \text{if } t > 0, r = 0, k = i \\ \mathsf{dp}(t, i, j, k - 1, A, m) & \text{if } t > 0, r = 0, k > i, \\ \mathbb{I}[\mathsf{dp}(t, i, j, k, A, r - 1) = 1 \vee (R_r : A \to BC \text{ for some } BC \text{ s.t.} \\ \quad (B \in \Sigma \wedge k = i + 1 \wedge v_k = B) \vee (B \in \mathcal{V} \wedge \mathsf{dp}(t, i, k, k, B, m) = 1), \\ \quad (C \in \Sigma \wedge j = k + 1 \wedge v_j = C) \vee (C \in \mathcal{V} \wedge \mathsf{dp}(t, k, j, j, C, m) = 1))] & \text{if } t > 0, r > 0. \end{cases}$$

The final answer will be $\mathsf{dp}(|\mathcal{V}|, 0, n, n, S, m)$. This DP formulation fits our framework [5].

**Regarding Remark 4.6.** It can be easily verified that the state spaces of the three problems mentioned above are of polynomial size, satisfying Assumption 4.2. Additionally, the MLP with the ReLU activation function can implement (the composition of) the following functions:

- $\max(a, b)$ and $\min(a, b)$, where $a, b \in \mathbb{R}$;
- $\mathbb{I}[a \ne b]$, $\mathbb{I}[a < b]$, $\mathbb{I}[a > b]$, where $a, b \in \mathbb{Z}$;
- $a \times b$, where $a \in \mathbb{R}, b \in \{0, 1\}$;
- linear transformation;
- conditional selection (Lemma C.4), for example,

$$f^{\text{select}}(x) = \begin{cases} f^>(x) & \text{if } x \ge 0, \\ f^<(x) & \text{if } x < 0, \end{cases}$$

  where $f^>$ and $f^<$ are functions that can be implemented by MLPs with ReLU activation, and $x \in \mathbb{Z}$.

This implies that the MLP with ReLU activation can approximate the functions $f, g, h$ in the transition function for the above three DP problems. According to Lemma C.2, these functions can be efficiently approximated by a perceptron of constant size with GeLU activation. Similarly, the topological ordering can also be efficiently implemented by MLPs with GeLU activation:

- LIS: $(j, k) \to \begin{cases} (j, k+1) & \text{if } k < j - 1 \\ (j+1, 0) & \text{if } k = j - 1 \end{cases}$

- ED: $(j, k) \to \begin{cases} (j, k+1) & \text{if } k < n_2 \\ (j+1, 0) & \text{if } k = n_2 \end{cases}$

---

[5]There is a subtle mismatch between the DP formulation and our framework (5), in that the terms $B$ and $C$ depend on the input (rather than purely determined by the state). Nevertheless, it is easy to extend Theorem 4.7 to this setting. Specifically, an autoregressive Transformer can first extract $B$ and $C$ using one layer and then extract $\mathsf{dp}(t, i, k, k, B, m)$ and $\mathsf{dp}(t, k, j, j, C, m))$ using an additional layer.

- CFG Membership Testing:

$$(t, i, j, k, A, r) \rightarrow \begin{cases} (t, i, j, k, A, r+1) & \text{if } r < m \\ (t, i, j, k, \mathsf{next}(A), 0) & \text{if } r = m, A \neq \mathsf{last}(\mathcal{V}) \\ (t, i, j, k+1, \mathsf{first}(\mathcal{V}), 0) & \text{if } r = m, A = \mathsf{last}(\mathcal{V}), k < j \\ (t, i+1, j+1, i+1, \mathsf{first}(\mathcal{V}), 0) & \text{if } r = m, A = \mathsf{last}(\mathcal{V}), k = j < n \\ (t+1, i+1, j+1, i+1, \mathsf{first}(\mathcal{V}), 0) & \text{if } r = m, A = \mathsf{last}(\mathcal{V}), k = j = n, t < |\mathcal{V}| \\ (0, 0, j-i+1, 0, \mathsf{first}(\mathcal{V}), 0) & \text{if } r = m, A = \mathsf{last}(\mathcal{V}), k = j = n, t = |\mathcal{V}| \end{cases}$$

where we order the set $\mathcal{V}$ and denote $\mathsf{first}(\mathcal{V})$ as the first element of $\mathcal{V}$, $\mathsf{last}(\mathcal{V})$ as the last element of $\mathcal{V}$, and denote $\mathsf{next}(A)$ the successor element of $A$.

Therefore, Assumptions 4.3 to 4.5 are satisfied and all three problems can be solved by autoregressive Transformers with CoT.

### G.2 Proof of Theorem 4.7

In this subsection, we will give proof of the Theorem 4.7.

**Theorem G.1.** *Consider any DP problem satisfying Assumptions 4.2 to 4.5. For any integer $n \in \mathbb{N}$, there exists an autoregressive Transformer with constant depth L, hidden dimension d and attention heads H (independent of $n$), such that the answer generated by the Transformer is correct for all input sequences $\boldsymbol{s}$ of length no more than $n$. Moreover, all parameter values are bounded by $O(\mathrm{poly}(n))$.*

*Proof.* **Input Format**. Assume that we have a sequence of tokens $s_1, \cdots, s_t$ and we want to generate the next token $s_{t+1}$. We embed the token $s_k$ by

$$\boldsymbol{x}_k^{(0)} = (\boldsymbol{e}_k^{\mathrm{input}}, \boldsymbol{e}_k^{\mathrm{state}}, \boldsymbol{e}_k^{\mathrm{dp}}, \boldsymbol{e}_k^{\mathrm{answer}}, \boldsymbol{e}_k^{\mathrm{sep}}, k, 1),$$

where each part of the embedding is defined as follows:

1. $\boldsymbol{e}_k^{\mathrm{input}}$ is the embedding of the input token $s_k \in \mathcal{X}$. If the current position does not represent an input token, then $\boldsymbol{e}_k^{\mathrm{input}} = \boldsymbol{0}$.

2. $\boldsymbol{e}_k^{\mathrm{state}}$ is the embedding of the DP state in $\mathcal{I}$ at position $k$. If the current position corresponds to an input token or the final answer, then $\boldsymbol{e}_k^{\mathrm{state}} = \boldsymbol{0}$. We also assume that for all $i \in \mathcal{I}$, the embedding of state $i$ is non-zero.

3. $\boldsymbol{e}_k^{\mathrm{dp}}$ is the embedding of the DP value in $\mathcal{Y}$ at position $k$. If the current position corresponds to an input token or the final answer, then $\boldsymbol{e}_k^{\mathrm{dp}} = \boldsymbol{0}$.

4. $\boldsymbol{e}_k^{\mathrm{answer}}$ is the embedding of the answer token in $\mathcal{Z}$, and $\boldsymbol{e}_k^{\mathrm{answer}} = \boldsymbol{0}$ if the current position corresponds to an input token or an intermediate DP position.

5. $\boldsymbol{e}_k^{\mathrm{sep}}$ is the embedding of the separator | separating different input sequences. We set $\boldsymbol{e}_k^{\mathrm{sep}} = \boldsymbol{e}_j$ if the current token $s_k$ is the $j$-th separator, where $\boldsymbol{e}_j$ is the one-hot vector with the $j$-th element begin 1.

6. The position embedding $k$ indicates the index of the token in the sequence.

**Block 1**. The first block of the autoregressive Transformer contains several layers. It first uses $N$ attention heads to perform the following task:

- Copy the positional embedding of the $N$ separators $p_k^{\mathrm{sep},1}, \cdots, p_k^{\mathrm{sep},N} \in \mathbb{N}$.

Similar to the previous proofs, this can be achieved via the COPY operation with $\mathcal{S}_k = \{j \leq k : \boldsymbol{e}_j^{\mathrm{sep}} = \boldsymbol{e}_t\}$ for $t \in [N]$ and $v_j = j$. Then, several MLPs follow, which perform the following tasks:

- Calculate the problem size $\boldsymbol{n}_k = (p_k^{\mathrm{sep},1} - 1, p_k^{\mathrm{sep},2} - p_k^{\mathrm{sep},1} - 1, \cdots, p_k^{\mathrm{sep},N} - p_k^{\mathrm{sep},N-1} - 1)$.

- Obtain the next state $\boldsymbol{e}_k^{\mathrm{next\_state}}$. If the current state is already the last state, set $\boldsymbol{e}_k^{\mathrm{next\_state}} = \boldsymbol{0}$.

The first task is a linear transformation, which can clearly be processed by an MLP Proposition C.3. According to Assumption 4.4, we can use an MLP to compute the embedding of the next state $\boldsymbol{e}_k^{\mathrm{next\_state}}$ based on the embedding of the current state $\boldsymbol{e}_k^{\mathrm{state}}$ and the problem size $\boldsymbol{n}$. When the required MLP in Assumption 4.4 has multiple layers (i.e., $\tilde{L}$ layers), we can use $\tilde{L} - 1$ Transformer layers

to implement a $\tilde{L}$-layer MLP. This can be achieved by just zero the weight matrices in the attention layers while maintaining the input using residual connections. The output of this block is

$$\boldsymbol{x}_k^{(1)} = (\boldsymbol{e}_k^{\text{input}}, \boldsymbol{e}_k^{\text{state}}, \boldsymbol{e}_k^{\text{next\_state}}, \boldsymbol{e}_k^{\text{dp}}, \boldsymbol{e}_k^{\text{sep}}, \boldsymbol{n}_k, k, 1).$$

**Block 2**. The second layer of the Transformer does not use attention heads. It only uses the MLP to perform the following tasks:

- Calculate $\boldsymbol{h}(\boldsymbol{n}_k, \boldsymbol{e}_k^{\text{next\_state}})$ and $\boldsymbol{g}(\boldsymbol{n}_k, \boldsymbol{e}_k^{\text{next\_state}})$. We assume that the embedding of $\emptyset$ is $\boldsymbol{0}$.
- Set the flag $f_k^{\text{state}}$ representing whether current state $\boldsymbol{e}_k^{\text{state}}$ is the last state.
- Set the flag $f_k^{\text{answer}}$ representing whether current state $\boldsymbol{e}_k^{\text{state}}$ is in the set $\mathcal{A}$, i.e., used in the aggregation function.

Similar to the first block, we stack several two-layer perceptrons to implement a multilayer perceptron. According to Assumptions 4.3 and 4.5, we can use an MLP to complete the first and the last tasks. The second task can be done by checking whether $\boldsymbol{e}_k^{\text{state}} \neq \boldsymbol{0}$ and $\boldsymbol{e}_k^{\text{next\_state}} = \boldsymbol{0}$. We also compute the auxiliary quantities $(\boldsymbol{h}(\boldsymbol{n}_k, \boldsymbol{e}_k^{\text{next\_state}}))^2$, $(\boldsymbol{g}(\boldsymbol{n}_k, \boldsymbol{e}_k^{\text{next\_state}}))^2$, $(\boldsymbol{e}_k^{\text{state}})^2$, and $k^2$, which are elementwise square operations and can be implemented by an MLP (Lemma C.1). The output of this block is

$$\boldsymbol{x}_k^{(2)} = (\boldsymbol{e}_k^{\text{input}}, \boldsymbol{e}_k^{\text{state}}, \boldsymbol{e}_k^{\text{next\_state}}, \boldsymbol{e}_k^{\text{dp}}, \boldsymbol{e}_k^{\text{sep}}, \boldsymbol{n}_k, \boldsymbol{h}(\boldsymbol{n}_k, \boldsymbol{e}_k^{\text{next\_state}}), \boldsymbol{g}(\boldsymbol{n}_k, \boldsymbol{e}_k^{\text{next\_state}}),$$
$$(\boldsymbol{h}(\boldsymbol{n}_k, \boldsymbol{e}_k^{\text{next\_state}}))^2, (\boldsymbol{g}(\boldsymbol{n}_k, \boldsymbol{e}_k^{\text{next\_state}}))^2, (\boldsymbol{e}_k^{\text{state}})^2, f_k^{\text{state}}, f_k^{\text{answer}}, k, k^2, 1).$$

**Block 3**. The third block of the Transformer uses $K + J$ heads to perform the following tasks (where $K$ and $J$ are defined in (5)):

- Copy the input token embeddings corresponding to $s_{g_1(i)}, \cdots, s_{g_J(i)}$ where $i$ corresponds to $\boldsymbol{e}_k^{\text{next\_state}}$. When $g_t(i) = \emptyset$, we set $s_{g_t(i)}$ to be a special token.
- Copy the DP value embeddings corresponding to $\mathsf{dp}(h_1(i)), \cdots, \mathsf{dp}(h_K(i))$ for $i$ corresponds to $\boldsymbol{e}_k^{\text{next\_state}}$. When $h_t(i) = \emptyset$, we set $\mathsf{dp}(h_t(i))$ to be a special value.
- Calculate the output $\mathsf{dp}(i)$ for $i$ corresponds to $\boldsymbol{e}_k^{\text{next\_state}}$, denoted as $\boldsymbol{e}_k^{\text{next\_dp}}$.

The first two tasks can be done via the COPY operation. To copy DP values, the attention head attends to positions $j$ with $\boldsymbol{e}_j^{\text{state}}$ matching $h_t(i)$ for $t \in [K]$. To copy input tokens, the attention head attends to positions $j = g_t(i)$ for $t \in [J]$. To handle the special token/value, it is simply a conditional selection operation and can be handled by an MLP (Lemma C.4). According to Assumption 4.3, we can calculate the function $f$ (defined in (5)) using an MLP. The output of this layer is

$$\boldsymbol{x}_k^{(3)} = (\boldsymbol{e}_k^{\text{next\_state}}, \boldsymbol{e}_k^{\text{dp}}, \boldsymbol{e}_k^{\text{next\_dp}}, \boldsymbol{n}_k, f_k^{\text{state}}, f_k^{\text{answer}}, k, 1).$$

**Block 4**. The fourth block of the autoregressive transformer contains one Transformer layer. Depending the aggregation function, it uses one attention head for the operation $\max$ or $\min$, or two attention heads for the operation $\sum$. This block performs the following tasks:

- Aggregate the DP values according to the aggregation function Equation (6).
- Generate the output based on the flag $f_k^{\text{answer}}$.

For the first task, if the aggregation function is $\max$ or $\min$, we use one attention head to simply copy the embedding $\boldsymbol{e}_j^{\text{dp}}$ for index $j$ such that $f_j^{\text{answer}} = 1$ and $\boldsymbol{e}_j^{\text{dp}}$ is the largest/smallest, according to Lemma C.7. If the aggregation function is $\sum$, we use two attention heads, where one attention head computes the mean of $\boldsymbol{e}_j^{\text{dp}}$ for index $j$ such that $f_j^{\text{answer}} = 1$, and the other attention head calculates the fraction of elements in the sequence such that $f_j^{\text{answer}} = 1$. Finally, the second task is a conditional selection operation and thus can be implemented by an MLP (Lemma C.4). □

### G.3 Proof of the Theorem 4.8

**Theorem G.2.** *Assume* $\mathsf{TC}^0 \neq \mathsf{NC}^1$. *There exists a context-free language such that for any depth $L$ and any polynomial $Q$, there exists a sequence length $n \in \mathbb{N}$ where no log-precision autoregressive transformer with depth $L$ and hidden dimension $d \leq Q(n)$ can generate the correct answer for the CFG Membership Testing problem for all input strings of length $n$.*

*Proof.* According to Barrington's theorem [4, 5], given a fixed unsolvable automaton, judging whether the automaton accepts an input is complete in $\mathsf{NC}^1$, which is a special case for the CFG Membership Testing problem. Therefore, the CFG Membership Testing problem is $\mathsf{NC}^1$-hard. With the assumption that $\mathsf{TC}^0 \neq \mathsf{NC}^1$, the CFG Membership Testing problem is out of the capacity of the log-precision autoregressive transformer. □

## H   Experimental Details

In this section, we present the experimental details.

### H.1   Datasets

We set the number field $p = 11$ in the math experiments. In the LIS experiment, we set the number of different input tokens to 150; in the ED experiment, we set the number of different input tokens to 26. The vocabulary is constructed by including all symbols. For all tasks and settings (direct v.s. CoT), the size of the training and testing dataset is 1M and 0.1M respectively. The constructions of different datasets are introduced below.

**Arithmetic Expression.**   All arithmetic expression problems are generated according to Algorithm 1. In Algorithm 1, we first create a number that serves as the answer to the problem. We then decompose the number using sampled operators sequentially, serving as the problem, until the maximum number of operators is met. The CoT procedure is precisely defined by reversing this problem generation process. For example, a sample in the direct dataset looks like

$$1 + 5 \times (1 - 2) = 7$$

while the corresponding sample in the CoT data looks like

$$1 + 5 \times (1 - 2) = 1 + 5 \times 10 = 1 + 6 = 7$$

---

**Algorithm 1:** Arithmetic Expression Problem Generation

**Input** : Number of Operators $n$
**Input** : Vocabulary of numbers $V = \{0, 1...10\}$ // `number field` $p = 11$
**Output :** Arithmetic expression $s$

1  Sample the first number $t$ uniformly from $V$;
2  $s = []$;
3  Append $t$ to $s$;
4  **for** $i \leftarrow 1$ **to** $n$ **do**
5      Sample $p$ uniformly from $\{0, 1, ..., len(s) - 1\}$, satisfying $s[p]$ is a number;
6      Sample $o$ uniformly from $\{+, -, \times, \div\}$;
7      Sample numbers $t_1, t_2$, satisfying the result of $o(t_1, t_2)$ equals $s[p]$;
8      **if** $s[p-1] = \div$ *or (* $o \in \{+, -\}$ *and* $s[p-1] \in \{-, \times\}$*) or (*$o \in \{+, -\}$ *and* $s[p+1] \in \{\times, \div\}$*)* **then**
9          pop $s[p]$;
10         insert $[(], [t_1], [o], [t_2], [)]$ sequentially into $s[p]$;
11     **else**
12         pop $s[p]$;
13         insert $[t_1], [o], [t_2]$ sequentially into $s[p]$;
14     **end**
15 **end**

---

**Linear Equation.**   All linear equation problems are generated according to Algorithm 2. In Algorithm 2, we consider the linear systems that only have a unique solution. Given a sampled linear system that satisfies this condition, we "translate" it to a sequence by concatenating all the equations (separated by commas), which serves as the problem. The answer to the problem is also a sequence consisting of variables and the corresponding values. The CoT solution of each problem is

the calculation process of the Gaussian elimination algorithm applied to each variable sequentially. For example, a sample in the direct dataset looks like

$$2x_1 + 3x_2 + 3x_3 = 8, 1x_1 + 7x_2 + 0x_3 = 0, 0x_1 + 2x_2 + 1x_3 = 1, \text{ [SEP] } x_1 = 4, x_2 = 1, x_3 = 10,$$

while the corresponding sample in the CoT dataset looks like

$$2x_1 + 3x_2 + 3x_3 = 8, 1x_1 + 7x_2 + 0x_3 = 0, 0x_1 + 2x_2 + 1x_3 = 1,$$
$$\text{[SEP] } x_1 + 7x_2 + 7x_3 = 4, 0x_2 + 4x_3 = 7, 2x_2 + 1x_3 = 1,$$
$$\text{[SEP] } x_1 + 9x_3 = 6, x_2 + 6x_3 = 6, 4x_3 = 7,$$
$$\text{[SEP] } x_1 = 4, x_2 = 1, x_3 = 10,$$

---

**Algorithm 2:** Linear Equation Data Generation

**Input** : Number of Variable $n$
**Input** : Vocabulary of numbers $V = \{0, 1...10\}$ // `number field` $p = 11$
**Output :** Linear Equation $s$
1 Sample $b$ uniformly from $V^{n \times 1}$;
2 **do**
3     Sample $A$ uniformly from $V^{n \times n}$;
4 **while** $A$ *is not invertible*;
5 $s \leftarrow$ "$A_{11}x_1 + ... + A_{1n}x_n = b_1, ..., A_{n1}x_1 + ... + A_{nn}x_n = b_n$"

---

**Longest Increasing Subsequence.** All input sequences (i.e., problems) are generated according to Algorithm 3. To make the task challenging enough, we first concatenate several increasing subsequences of given length, and then randomly insert numbers into the whole sequence. The inputs has 150 different tokens, ranging from 101 to 250 to avoid token overlap with DP array. The CoT solution to the problem is the DP array plus the final answer, which is defined in (20). Here, we consider the DP formulation (20) because the CoT output length is much shorter than the one corresponding to formulation (21). This allows us to consider more challenging input sequences with longer length. While this DP formulation does not precisely obey the theoretical assumption given in Assumption 4.3, we found that the Transformer can still learn it easily.

For example, a sample in the direct dataset looks like

$$103 \ 107 \ 109 \ 112 \ 101 \ 103 \ 105 \ 107 \ 115 \ 109 \ 111 \ 113 \ 102 \text{ [SEP] } 7$$

while the corresponding sample in the CoT dataset looks like

$$103 \ 107 \ 109 \ 112 \ 101 \ 103 \ 105 \ 107 \ 115 \ 109 \ 111 \ 113 \ 102$$
$$\text{[SEP] } 1 \ 2 \ 3 \ 4 \ 1 \ 2 \ 3 \ 4 \ 5 \ 5 \ 6 \ 7 \ 2$$
$$\text{[SEP] } 7$$

---

**Edit Distance.** All input sequences (i.e., problems) are generated according to Algorithm 4. In Algorithm 4, we generate the first string randomly. For the generation of the second string, we use two methods. In the first method, we generate the second string randomly, corresponding to a large edit distance. In the second method, we copy the first string with random corruption, corresponding to a small edit distance. The two strings are concatenated by "|", and the concatenation is used as the model input. For the calculation of edit distance, we assign different costs to different operators. The costs for the ADD and DELETE operators are set to 2, while the REPLACE operator is assigned a cost of 3, since REPLACE should be more costly than ADD/DELETE while less costly than their summation. The CoT procedure is also the DP array, defined in Section 4.1. For example, a sample in the direct dataset looks like

$$a \ s \ | \ p \ a \ s \ s \text{ [SEP] } 4$$

while the corresponding sample in the CoT dataset looks like

$$a \ s \ | \ p \ a \ s \ s$$
$$\text{[SEP] } 3 \ 2 \ 4 \ 6$$
$$\text{[SEP] } 5 \ 4 \ 2 \ 4$$
$$\text{[SEP] } 4$$

**Algorithm 3:** LIS Data Generation

**Input** : Sequence Length $n$
**Input** : Vocabulary of numbers $V = \{101, 101...250\}$
**Output** : Sequence $s$

1   Sample $l$ uniformly from $\{3, 4...n\}$ ;
2   Sample $t$ uniformly from $\{1, 2, 3\}$ ;
3   $a = []$ ;
4   push $0$ to $a$ ;
5   **if** $t = 2$ **then**
6      Sample $j$ uniformly from $\{1, 2...\lfloor l/2 \rfloor + 1\}$ ;
7      push $j$ to $a$ ;
8   **else if** $t = 3$ **then**
9      Sample $j$ uniformly from $\{1, 2...\lfloor l/3 \rfloor + 1\}$ ;
10      Sample $k$ uniformly from $\{1, 2...\lfloor (l - j)/2 \rfloor + 1\}$ ;
11      push $j$ to $a$ ;
12      push $j + k$ to $a$ ;
13   push $l$ to $a$;
14   $s \leftarrow$ Sample $l$ numbers from $V$;
15   **for** $i \leftarrow 1$ **to** $t$ **do**
16      Sort $s[a[i - 1] : a[i]]$;// This process makes sure the LIS of the generated sequence $s$ is at least $\lceil l/t \rceil$.
17   **end**
18   $r \leftarrow$ Sample $n - l$ numbers from $V$;
19   Randomly insert $r$ into $s$;

---

**Algorithm 4:** ED Data Generation

**Input** : Length of the First String $n$
**Input** : Alphabet $V = \{a, b...z\}$
**Output** : Sequence $s_1$, $s_2$

1   Sample $t$ uniformly from $\{3, 4...10\}$ ;
2   $T \leftarrow$ Sample $t$ letters from $V$ ;
3   $s_1 \leftarrow$ Sample $n$ letters uniformly from $T$ ;
4   Sample $p$ uniformly from $[0, 1]$ ;
5   **if** $p < 0.4$ **then**
6      Sample $l$ uniformly from $\{n - 3, n - 2, ..., n + 2\}$;
7      $s_2 \leftarrow$ Sample $l$ letters uniformly from $T$ ;
8   **else**
9      **do**
10         $s_2 \leftarrow s_1$ ;
11         **for** $i \leftarrow 1$ **to** $n$ **do**
12            Sample $p$ uniformly from $\{0, 1...len(s_2) - 1\}$;
13            Sample $l$ uniformly from $T$;
14            Randomly conduct one of the followings: pop $s_2[p]$, substitute $s_2[p]$ with $l$, insert $l$ into $s_2[p]$;
15         **end**
16      **while** $len(s_2)$ *not in* $[n - 3, n + 2]$;
17   **end**

## H.2 Model training

We use the minGPT implementation[6] for all the experiments, where the detailed Transformer layer are listed below.

$$\boldsymbol{X}^{(0)} = \mathrm{LayerNorm}\left([\boldsymbol{v}_1 + \boldsymbol{p}_1, \cdots, \boldsymbol{v}_n + \boldsymbol{p}_n]^\top\right) \tag{23}$$

$$\mathrm{Attn}^{(l)}(\boldsymbol{X}) = \sum_{h=1}^{H}\left(\mathrm{softmax}\left(\boldsymbol{X}\boldsymbol{W}_Q^{(l,h)}(\boldsymbol{X}\boldsymbol{W}_K^{(l,h)})^\top/\sqrt{d} + \boldsymbol{M}\right)\right)\boldsymbol{X}\boldsymbol{W}_V^{(l,h)}\boldsymbol{W}_O^{(l,h)} \tag{24}$$

$$\mathrm{FFN}^{(l)}(\boldsymbol{X}) = \sigma(\boldsymbol{X}\boldsymbol{W}_1^{(l)})\boldsymbol{W}_2^{(l)} \tag{25}$$

$$\boldsymbol{Y}^{(l-1)} = \boldsymbol{X}^{(l-1)} + \mathrm{Attn}^{(l)}(\mathrm{LayerNorm}(\boldsymbol{X}^{(l-1)})) \tag{26}$$

$$\boldsymbol{X}^{(l)} = \boldsymbol{Y}^{(l-1)} + \mathrm{FFN}^{(l)}(\mathrm{LayerNorm}(\boldsymbol{Y}^{(l-1)})) \tag{27}$$

We use sinusoidal positional embedding and use Xavier initialization for all the parameters. The activation function is chosen to be GeLU. The dimension of the embedding is set to 256, and the number of heads is set to 4. The hidden size in the FFN layer is set to 1024.

We use the same hyperparameter configuration for all experiments, i.e., the performance comparison between the models trained on the direct and CoT datasets of Arithmetic, Equation, LIS, and ED tasks, and the additional length extrapolation experiments (which we use relative positional encodings [48] instead of absolute positional encodings). In detail, we use AdamW optimizer with $\beta_1$, $\beta_2 = 0.9$, 0.999. The learning rate is set to 1e-4, and the weight decay is set to 0.01. We set the batch size to 512 during training with a linear learning rate decay scheduler. We use learning rate warm-up, and the warm-up stage is set to 5 epochs. The dropout rate is set to 0.1. The total number of training epochs is set to 100.

# I  Robustness in Training Data Quality

In the training of language models, the quality of data, especially intermediate steps, is crucial for optimal performance. However, real-world training datasets may contain corrupted or missing intermediate steps. This highlights the importance of evaluating model resilience under imperfect conditions. To investigate this, we conducted a series of experiments focused on arithmetic tasks, introducing varying rates of corruption and omission, encapsulated by the parameter $\gamma$. For instance, when $\gamma = 0.1$, it denotes that 10% of the intermediate steps are omitted and 10% of the remaining steps are corrupted through a single-token random replacement. Our motivation for introducing this metric is to simulate potential imperfections in real-world datasets, thus assessing the robustness of our model more thoroughly. The number of operators in the arithmetic task is set to 10.

Table 1: Experimental results of robustness.

| $\gamma$ | 0 | 0.1 | 0.2 | 0.3 |
|---|---|---|---|---|
| **Accuracy** | 100.0% | 98.5% | 97.6% | 95.8% |

Table 2: Model performance on Arithmetic task with a 3-layer Transformers.

Our experimental results are presented in Table 1. As the rate of corruption and omission increases, there is a slight decrease in accuracy. These results elucidate the inherent robustness of training CoT demonstrations. It is noteworthy that the accuracy of the model remains commendably high despite significant distortions present in the training data, highlighting its potential utility in real-world scenarios where data quality may be compromised.

---

[6] https://github.com/karpathy/minGPT

