# OpenReview forum: "Towards Revealing the Mystery behind Chain of Thought: A Theoretical Perspective"
_NeurIPS.cc/2023/Conference — NeurIPS 2023 oral_

### Official Review · Reviewer_7MZe · 2023-07-05

**Soundness:** 4 excellent
**Presentation:** 3 good
**Contribution:** 3 good
**Rating:** 7
**Confidence:** 4

**Summary:**

The paper mainly focuses on theoretically proving the effectiveness of the CoT in autoregressive Transformer models for solving fundamental mathematical and decision problems through generating intermediate steps. It demonstrates that any finite-depth Transformer model cannot directly output correct answers to these tasks unless the model size grows super-polynomially with the input length. The paper also includes experimental validation using a constructed dataset.

**Strengths:**

1. The paper provides a solid theoretical proof of the effectiveness of CoT, making a valuable contribution to further exploring the underlying mechanisms of CoT operation.

2. The theoretical proofs presented in the paper are comprehensive and well-organized.

3. The research problem addressed in the paper is clearly significant. CoT has shown strong empirical performance but lacks theoretical analysis. Thus, this study is timely and necessary.


**Weaknesses:**

1. The experimental section seems to be insufficient, for example, the length extrapolation experiment only provides results for one task.

2. The exploration of model architectures is lacking. (see my questions)

**Questions:**

1. Is there a specific reason for choosing autoregressive Transformer? Were other models, such as the encoder-decoder architecture like T5 model be considered? Would CoT be equally effective in those models?

2. In the experiments, a Transformer with three layers achieves near-perfect accuracy on each task, but the proofs utilize five layers. How much does the number of layers affect the performance of CoT, and are there necessary layer numbers for different tasks? Additionally, the difficulty of the datasets for now seem insufficient to explore this question.

3. In the Length Extrapolation section, only Arithmetic Expression Extrapolation is explored. Does CoT also possess the ability to learn the underlying mechanisms for other tasks? The conclusion seems to lack sufficient experimental verification and analysis.

4. Have you considered the impact of the quality of intermediate steps on the performance, such as inserting some invalid intermediate steps or omitting a certain number of intermediate steps?

**Limitations:**

Section 7 of the paper provides a thorough discussion of the limitations. Additionally, while CoT demonstrates excellent performance across various tasks, this paper mainly focuses on mathematical tasks and could further explore other types of tasks, such as logic reasoning tasks represented in natural language.

---

> ### Author Rebuttal · Authors · 2023-08-09
>
> We sincerely thank Reviewer 7MZe for the careful reading, thoughtful inquiries, and positive feedback. Below, we are happy to provide further elaboration on each of the points you raised:
>
> > Q1: Is there a specific reason for choosing autoregressive Transformer? Were other models, such as the encoder-decoder architecture like T5 model be considered? Would CoT be equally effective in those models?
>
> Thanks for the question. We choose autoregressive Transformer as it is the de facto standard for LLMs (e.g., GPT). It is also simpler than encoder-decoder architectures, which makes our analysis and proof cleaner. Yet, our theoretical results can be easily transferred to an encoder-decoder model (like T5) using the following argument. (a) On one hand, given a bounded-depth polynomial-size log-precision encoder-decoder Transformer, its parallel complexity is still bounded by $\mathsf{TC}^0$, so our negative results hold. (b) On the other hand, any finite-depth autoregressive Transformer can be mimicked by a finite-depth encoder-decoder Transformer of roughly the same size, since causal masking for the input sequence can be mimicked through the use of positional encoding and joint attention can be mimicked by an integration of cross-attention and self-attention. Thus, all results in this paper are not exclusive to autoregressive Transformers. We can add those discussions into the paper if you think they are helpful.
>
> > Q2: How much does the number of layers affect the performance of CoT, and are there necessary layer numbers for different tasks? Additionally, the difficulty of the datasets for now seem insufficient to explore this question.
>
>
> Thanks for the question. We study simple tasks in the paper and found that from both theoretical and practical perspectives, a shallow LLM with CoT already has enough capacity to solve them. We agree and believe deeper LLMs are required to solve more complicated tasks with CoT. For example, it can be easily imagined that a deeper LLM is needed to complete a task if it composes of solving linear equations and arithmetic with dynamic programming. We are working on more general and advanced reasoning tasks and investigating the dependency between task and LLM parameter complexity.
>
>
> > Q3: Does CoT also possess the ability to learn the underlying mechanisms for other tasks beyond Arithmetic Expression Extrapolation?
>
> Thanks for the question. During the tight discussion period, we have completed experiments for longest increasing subsequence task. The results of the LIS task are shown in the table below. Here, we train the autoregressive Transformer model with various input length ranging from 1 to 80, and test the model using longer input sequence lengths unseen during training. It can be seen that the model can still extrapolate well to longer sequences.
>
>    | **Length**   | 82 | 84 | 86 | 88 | 90 |
>    | ------------ | -- | -- | -- | -- | -- |
>    | **Accuracy** | 94.3% | 92.8% | 90.7% | 89.6% | 87.4% |
>
> We are currently running experiments on the linear equation task. But it cannot be finished before the rebuttal deadline given very limited GPU memory resources (the required CoT length is very long even for solving linear equations with 7 variables). We will report the performance when it is finished using more advanced GPUs.
>
> > Q4: Have you considered the impact of the quality of intermediate steps on the performance, such as inserting some invalid intermediate steps or omitting a certain number of intermediate steps?
>
> This is a good catch as real-world language model training data often involves corrupted or omitted intermediate steps. To assess this, we conducted experiments on arithmetic task with varying rates of corruption and omission, denoted as γ. Specifically, γ = 0.1 indicates that we skip 10% intermediate steps and corrupt 10% steps with a single-token random replacement. The table below presents the experimental results:
>
>    | γ | Accuracy |
>    | ---- | -------- |
>    | 0.1 | 98.5% |
>    | 0.2 | 97.6% |
>    | 0.3 | 95.8% |
>
> The results clearly demonstrate the robustness of training CoT demonstrations, showcasing its ability to maintain high accuracy even in the presence of imperfect intermediate steps in the training datasets. We will add those robustness evaluation into the final version of the paper.
>
> We hope these clarifications can address your questions satisfactorily and we are happy to delve further into any
> of these aspects.

---

### Official Review · Reviewer_P6n5 · 2023-07-06

**Soundness:** 4 excellent
**Presentation:** 4 excellent
**Contribution:** 4 excellent
**Rating:** 8
**Confidence:** 2

**Summary:**

The paper contributes to theoretical and empirical understanding of Chain-of-thought, i.e. intermediate process generation to assist desired output generation. In the theory part, authors show
- log-presicion Transformer with bound depth cannot solve simple math tasks (calculate, solve linear equations) unless the model size grows super-polynomially w.r.t. input length. the proof is based on circuit complexity and a bottleneck of parallel complexity (assuming $TC^0 \neq NC^1$). However, constant-size Transformer can solve both by generating common math intermediate steps.
- for dynamic programming (DP), Transformers with CoT can generate the correct answer intuitively. In contrast, it is proven that Context-Free Grammar Membership Testing cannot be solved with a bounded-depth Transformer with polynomial depth.

Experiments then validate these results using two math and two DP tasks, showing CoT > no CoT (with more layers even).

**Strengths:**

- CoT has been important in language processing, and its theoretical and empirical understanding is important and timely.

- As far as I can tell, both the theory and experiment parts of the paper is solid, and well-written.

- I like the connection to circuit complexity theory. Though the conclusion and "intuition" is not hard to grasp, the theory part is technical and non-trivial to establish.

**Weaknesses:**

- As noted by authors, it's still limited to expressivity (not learning with large corpora, large model).

- Some missing references around Dyck language recognition:

Self-attention networks can process bounded hierarchical languages. ACL 2021
RNNs can generate bounded hierarchical languages with optimal memory. EMNLP 2020


**Questions:**

- Seems "Self-attention networks can process bounded hierarchical languages" proved that (2-layer) Transformers can recognize Dyck, yet this papers proves CFG recognition is in general hard? Some discussion would be nice.

- Would results hold for RNN or other architectures? Some discussion would make the paper stronger.

**Limitations:**

Authors write about limitations fairly.

---

> ### Author Rebuttal · Authors · 2023-08-09
>
> We sincerely thank Reviewer P6n5 for the positive feedback, valuable suggestions, and two insightful questions regarding the related work and other architectures. Below, we would like to give detailed responses to each of your comments and questions.
>
> **Regarding related work**. Thanks for the valuable suggestions on the related work. We will follow your advice and include these two papers in our related work section. These two papers focus on the Dyck language, a special case of the CFG. The first one shows that the hard-attention transformers can recognize Dyck language with bounded depth. Moreover, a two-layer soft-attention transformer can generate Dyck language. The second paper tries to use RNN to generate bounded-depth Dyck language. This paper demonstrates that RNN can generate the Dyck language with hidden units of reasonable size. Furthermore, the paper proves that the size of the hidden units is optimal. The conclusions of these two papers complement our theorems on the general CFG.
>
> We also thank you for posing two insightful questions that deserve careful discussion.
>
> > **Question**: Seems "Self-attention networks can process bounded hierarchical languages" proved that (2-layer) Transformers can recognize Dyck, yet this papers proves CFG recognition is in general hard?
>
> Yes, the problem of general CFG recognition is much harder than the special case of the Dyck language. The paper “Self-attention networks can process bounded hierarchical languages” employs a Transformer encoder that is similar to the auto-regressive Transformer without CoT, except for the causal mask. The complexity of both Transformer encoder and autoregressive Transformer without CoT is upper bounded by the circuit complexity $\mathsf{TC}^0$. For the special case of Dyck language, it has been proved that the problem of recognizing Dyck language is actually in the complexity class $\mathsf{TC}^0$ [1]. Therefore, it is possible for a transformer model to recognize the Dyck language. However, general CFG recognition is $\mathsf{P}$-complete, which is intrinsically hard to be solved by a Transformer without CoT.
>
> [1] On the relative complexity of some languages in NC1. Barrington et al.
>
> > **Question**: Would results hold for RNN or other architectures?
>
> First, most of the theorems in this paper can be naturally extended to some popular settings with Transformer, such as the encoder-decoder architectures (e.g., T5). However, as for the RNN, it is actually not the case. We can show that RNNs cannot generate the CoT sequence using the same format proposed in our paper for the arithmetic formula task and the linear equation task unless the hidden dimension of the RNN is at least $O(\frac{n}{\log n})$, where $n$ is the input length. The reason is as follows. When the RNN generates the first equal sign, it has to compress the input sequence into a hidden state of $O(D\log n)$ bits, where $D$ is the hidden dimension and each element is represented by $O(\log n)$ bits (by definition of log-precision). On the other hand, the first step of the CoT needs to output a sequence of length $O(n)$, which contains $O(n)$ numbers. Therefore, there are at least $2^{O(n)}$ different output sequences, and thus by the Pigeon Hole Principle, to be able to generate all $2^{O(n)}$ different output sequences, we must have $D=\Omega(\frac{n}{\log n})$.

---

> > ### Comment · Reviewer_P6n5 · 2023-08-14
> > **Thanks**
> >
> > Thanks! Would be nice to see these incorporated into the revision.

---

> > > ### Author Response · Authors · 2023-08-16
> > > **Thanks for your feedback**
> > >
> > > Thank you! We will definitely incorporate them into the next version of this paper.

---

### Official Review · Reviewer_QoTB · 2023-07-06

**Soundness:** 3 good
**Presentation:** 3 good
**Contribution:** 3 good
**Rating:** 8
**Confidence:** 3

**Summary:**

This paper presents various separation results, showing that a Transformer with CoT can solve certain formally-defined reasoning tasks, but a Transformer *without* CoT cannot (assuming bounded depth). This sheds light on the power of CoT. The formal results are supplemented with empirical results that support the claims.

**Strengths:**

- Understanding CoT is an important and timely question. This paper is therefore tackling a significant question.
- To the best of my knowledge, this is the first paper to provide a theoretical explanation of the power of CoT (i.e., originality).
- In general, the paper is quite clear and high-quality, though I think the notation could be improved (see below).

**Weaknesses:**

- I think calling it CoT but focusing on "CoT generation" is perhaps misleading. Many people think of CoT as a prompting technique. In my opinion, the generation aspect that this paper focuses on is actually bigger/more important than CoT suggests, and I actually think that - although buzzwordy - CoT diminishes the general power of generation that this paper is getting at (i.e., this paper transcends CoT). I would suggest looking into alternative, more general titles.
- Section 4.1 has a lot of notation, including double subscripts. I think there's a significant lack of accessibility created by the heavy notation. I would really encourage the authors to think about whether the notation could be simplified. I think doing so could substantially increase the long-term impact of this paper.
- Relatedly, equations (4) and (5) would be clearer if their new objects were introduced and explained conceptually before jumping into equations (4) and (5).
- Overall, I think the paper could do a better job explaining the significance of the theoretical results. Although this is an important problem to study theoretically, and this paper does a good job of initiating that study, it's not entirely clear what can be gained conceptually from this analysis. I think many people already intuitively grasp that generating more tokens gives an LLM more power, and various methods that allow more tokens to be generated before arriving at a final answer are more powerful. It would be nice to understand whether there's a conceptual message here that goes any deeper than the aforementioned intuition.

**Questions:**

- See CoT naming comment above. Do you agree?
- See notation comment above. Is there a way to simplify the notation in Section 4.1? Is there a reason it has to be this complicated? If it seems necessary, maybe there's a simpler version that can be presented in the body of the paper, with the full version moved to the appendix?
- My main question is also discussed in the Weaknesses section. Is there any conceptual takeaway from this theoretical analysis beyond "generating more tokens is more powerful"? Even without that, I think this is a strong paper. However, I think this could be an especially powerful paper if the intuitions from the theoretical analysis could be further synthesized in this direction (and made accessible to readers).

**Limitations:**

The authors do a nice job of discussing some of the limitations of this work. There is no discussion of societal impact, but I do not think this is a problem for this particular paper.

---

> ### Author Rebuttal · Authors · 2023-08-09
>
> We sincerely thank Reviewer QoTB for the careful reading, positive feedback, valuable suggestions regarding presentations, and detailed comments. Below, we would like to give detailed responses to each of your comments and questions.
>
> **Regarding the scope of the paper**. Thanks for the suggestion. We connect our work to "CoT" because our theory suggests that to solve math/reasoning problems, "generating intermediate deviations in an autoregressive way" is easier and much more parameter-efficient. As this generation process is regarded as "Chain of Thought" by the community, we leverage the concept during writing. On the other hand, we fully agree that intrinsically, this work is more about the way of using LLMs rather than how to develop specific CoT prompts. We will make this clear in the introduction and are happy to illustrate more and try to come up with a more accurate and appropriate title.
>
> **Regarding notations in Section 4**. We appreciate the constructive suggestions and will revise our manuscript accordingly. In particular, we intend to simplify the notations in Section 4.1 by using a vectorized notation. For example, in equation (5), we will use $s_{\mathbf{g}(i)}$ to represent the vector $(s_{g_1(i)},\cdots,s_{g_J(i)})$. In this way, equation (5) can be rewritten in a more concise form, avoiding the problem of double subscript. The original equation and the updated equation are shown below:
> $$
> \mathsf{dp}(i)=f(i,s_{g_1(i)},\cdots,s_{g_n(i)},\mathsf{dp}(h_1(i)),\cdots,\mathsf{dp}(h_K(i)))
> $$
>
> $$
> \mathsf{dp}(i)=f(i,s_{\mathbf{g}(i)},\mathsf{dp}(\mathbf{h}(i)))
> $$
>
> > **Question**: Is there any conceptual takeaway from this theoretical analysis beyond "generating more tokens is more powerful"? ...... However, I think this could be an especially powerful paper if the intuitions from the theoretical analysis could be further synthesized in this direction (and made accessible to readers).
>
> Thanks for raising this good question. It makes us realize that there is still much room for us to improve the writing of the paper. Indeed, our theoretical analysis has a bunch of conceptual takeaways beyond the surface conclusion that "generating more tokens is more powerful". We detail these takeaways below:
>
> * **The key role of self-attention in CoT generation**. Our analysis points out two key components that enable CoT generation, which we call **COPY** and **MEAN**. Specifically, the COPY operation extracts the hidden information of a specific previous position that satisfies certain conditions, e.g., extracting the hidden embedding of the last equal sign (=), or the last non-number token. The MEAN operation averages the hidden embedding for a set of previous positions that satisfy certain conditions, e.g., the average hidden embedding of all *number* tokens between the last equal sign and the current token. We prove that **both COPY and MEAN can be realized by a self-attention layer**. We then use exactly the two operations to build entire **parallel algorithms** that can generate CoT sequences for all math and DP problems. Our result highlights the crucial role played by the self-attention and Transformer architecture, and may inspire future research on architectural design in Large Language Models (e.g., more efficient LLMs).
>
> * **Regarding the length of CoT generation**. Our theoretical analysis also gives insights into how many intermediate steps are needed in the CoT generation. In particular, when the Transformer model generates a CoT sequence, we have to ensure that the complexity of generating each step is within $\mathsf{TC}^0$. Using this argument, one can easily check whether the length of a specific CoT format is sufficient. For more complex problems, we need to decompose it into more subproblems (or more steps), so that each step is within the complexity of $\mathsf{TC}^0$. Moreover, we give a standard criterion to check whether the $\mathsf{TC}^0$ expressivity is satisfied: if each CoT step can be represented by a finite composition of **COPY** and **MEAN** operations, then each CoT step will be within the complexity of $\mathsf{TC}^0$.
>
> * **Regarding dynamic programming**. We theoretically show that CoT allows Transformers to solve general DP problems. Thus, as a direct consequence, Transformers are even capable of solving extremely hard problems that are $\mathsf{P}$-complete. This provides a deeper understanding of why popular large language models can be so powerful in reasoning. We believe our proposed dynamic programming framework may also be useful for studying and measuring the expressive power of other architectures in the future.
>
> We will incorporate these discussions into the next version of our paper.
>
> We hope our response can clarify your concerns and will improve the paper writing according to your suggestions and questions. We are happy to go into more detail regarding any of the above questions and we look forward to your reply.

---

> > ### Comment · Reviewer_QoTB · 2023-08-18
> >
> > Thanks for the very thoughtful response! I think incorporating these takeaways into the body of the paper and highlighting them as contributions significantly increases the impact of the paper. To what extent do you think this theoretical construction is capturing what's happening empirically? Do you have any evidence in either direction that you could provide in the body of the paper? I strongly support highlighting these takeaways in the body either way, but I also want to make sure the community doesn't latch onto them too prematurely (especially since the focus here is on what can be represented, not what's actually learned through gradient descent). Although negative evidence, to whatever extent it exists, seems like an important caveat, positive evidence would certainly be a very compelling addition to this paper.
> >
> > With the additions provided in the authors' response, I'm raising my score to a 7 (under the assumption that they'll be included in the final version).

---

> > > ### Author Response · Authors · 2023-08-20
> > > **Thanks for your feedback**
> > >
> > > Thank you for your positive feedback and additional comments, and for asking the good question. We are happy to provide additional responses to the insightful question you raised:
> > >
> > > > Question: To what extent do you think this theoretical construction is capturing what's happening empirically? Do you have any evidence in either direction that you could provide in the body of the paper?
> > >
> > > We believe our theoretical construction can to some extent capture what's happening empirically and is more meaningful than several prior works. This is due to the following reasons. First, we use log-precision Transformer instead of infinite precision, which can be precisely implemented in modern GPU architectures in practice. Moreover, the values of weight elements in our constructions are often not large and thus are likely to be learned by gradient descent. Second, the size of the Transformer architecture in our construction is reasonable. For example, we only use no more than 5 layers, 5 heads and a hidden dimension of $O(1)$. Third (and most importantly), prior work has pointed out that the COPY operation, which forms the basic building block in our construction, does appear empirically [1]. Specifically, the authors proposed the concept of "induction head", which is a module that can find the position in the sentence where the current token previously appeared and then extract the next token after that position. The authors visualized the attention score matrix for various tasks and found that the Transformer model is indeed performing the induction head. Therefore, based on the above points, we argue that the theoretical construction is capturing some intrinsic characteristics in practical scenarios.
> > >
> > > Nevertheless, the above evidence is still kind of "indirect" since we cannot prove that gradient-descent-based optimizers can learn the construction. But we believe adding the above discussion into the main paper, especially the connection to the "induction head" mechanism, can benefit our community and may further enhance the impact of our paper.
> > >
> > > For other suggestions, we will definitely incorporate these insights into the final version of our paper. Finally, we really appreciate your effort in making our paper better and we will be more than happy to discuss more if you have other questions or suggestions. Thank you!
> > >
> > > [1] In-context Learning and Induction Heads. Olsson et al.

---

> > > > ### Comment · Reviewer_QoTB · 2023-08-20
> > > >
> > > > Thanks for the reply! I think these points are compelling. Assuming all of this is incorporated into the main paper and is appropriately caveated (i.e., something along the lines of the "indirect" sentence above), I am raising my score to an 8. I think all of these additions significantly strengthen the paper's impact and elevate it from a paper that would primarily benefit the theory community to a paper that will really benefit anyone trying to understand Transformers.

---

> > > > > ### Author Response · Authors · 2023-08-21
> > > > > **Thank you**
> > > > >
> > > > > Thanks for your additional feedback! We will definitely incorporate them into the next version of this paper.

---

### Official Review · Reviewer_C8fV · 2023-07-09

**Soundness:** 4 excellent
**Presentation:** 4 excellent
**Contribution:** 4 excellent
**Rating:** 9
**Confidence:** 2

**Summary:**

This paper studies the theoretical power of the Chain-of-Thought (CoT) prompting. In particular, this paper mathematically confirms that two well-chosen tasks (e.g., arithmetic and equation) and the problem of Dynamic Programming are beyond bounded-depth Transformer models without CoT (unless their size grows prohibitively large); with CoT and generated intermediate derivations, those problems become solvable. The mathematical derivations are under mild assumptions, and empirical results confirms the mathematical study (on four representative tasks).

**Strengths:**

Given the prevalence of CoT, theoretical study of the limit of CoT becomes extremely valuable. This paper overcomes several limits in previous studies (e.g., assuming infinite precision) and focuses on the setting of autoregressive Transforms, which is close to the scenario of real-world LLMs. Moreover, the proposed empirical tasks are illustrative and easily reproducible.

**Weaknesses:**

I don't find any noticeable weakness in this paper.

**Questions:**

Line 245, the success of solving Dynamic Programming problems critically depends on the input sequences being laid out in a topological order. In the two DP experiments (Longest Increasing Subsequence and Edit Distance), the topological order appears to be learnable from the CoT dataset. Could there be any theoretical study about how hard it is to learn this order?

**Limitations:**

The authors have adequately addressed the limitations.

---

> ### Author Rebuttal · Authors · 2023-08-09
>
> We sincerely thank Reviewer C8fV for the positive feedback, appreciation for our work, and insightful questions.
>
> > **Question**: Line 245, the success of solving Dynamic Programming problems critically depends on the input sequences being laid out in topological order. In the two DP experiments (Longest Increasing Subsequence and Edit Distance), the topological order appears to be learnable from the CoT dataset. Could there be any theoretical study about how hard it is to learn this order?
>
> We thank the reviewer for the insightful question. In our paper, we mainly study CoT from an expressivity perspective, i.e., whether there exists a model that can generate the topological ordering of the DP states and find the correct solution, and we prove that the answer is yes. Your question relates to the generalization ability of CoT training, i,e., why the model behaves well on unseen data. For this question,  we can offer some intuitive insights into why generating the topological ordering is easy for an autoregressive Transformer. In our formulation, the topological ordering can be determined by a function $F$ that takes the current state $i_k$ and the problem scale $n$ as inputs and outputs the next state $i_{k+1}$, i.e., $F(i_k,n)=i_{k+1}$. Assumption 4.4 guarantees that $F$ can be efficiently approximated by a constant-sized perceptron with GeLU activation, by which **the topological order can be easily generated in an autoregressive way by this simple function $F$**. In the Appendix, we give an explicit expression of the function $F$ for each DP problem considered in this paper, all of which are simple:
> $$
> \text{LIS}: F((j,k),n)=\begin{cases}
> (j,k+1)\ \ \text{if}\ \ k<j-1\\\\
> (j+1,0)\ \ \text{if}\ \ k=j-1
> \end{cases}
> $$
> $$
> \text{ED}: F((j,k),(n_1,n_2))=\begin{cases}
> (j,k+1)\ \ \text{if}\ \ k<n_2\\\\
> (j+1,0)\ \ \text{if}\ \ k=n_2
> \end{cases}
> $$
> It is easy to see that $F$ can be represented by a small MLP with ReLU (or GeLU) activation. Therefore, it may not be difficult for the model to learn the topological order given sufficient training demonstrations.
>
> On the other hand, we believe theoretically understanding the generalization ability of CoT learning process is extremely important and we leave it as future work. Although there are some works trying to explain the generalization of modern deep learning models [1,2], they may not provide useful insights for the CoT setting. Studying the generalization ability of large language models from massive data around their reasoning ability should be a very fundamental and practical problem that deserves more attention in the future.
>
> [1] Li, Yingcong, et al. "Transformers as algorithms: Generalization and implicit model selection in in-context learning."
>
> [2] Xu, Keyulu, et al. "How neural networks extrapolate: From feedforward to graph neural networks."

---

> > ### Comment · Reviewer_C8fV · 2023-08-16
> > **Thanks**
> >
> > Thank you very much for the detailed and enlightening reply! I really appreciate the analysis.

---

### Author Rebuttal · Authors · 2023-08-09

We would like to express our sincere thanks to the reviewers and the area chair for taking the time to review our paper. We have responded to each reviewer's comments separately and will incorporate their suggestions into the next version of our paper. We hope that our response can adequately addresse the reviewers' concerns, and we're happy to provide more details about any questions they may have.

---

### Decision · Program_Chairs · 2023-09-21

**Decision:**

Accept (oral)

**Comment:**

This paper provides new theoretical understandings of the Chain-of-Thought (CoT) technique, by proving rigorous separation results between the CoT (autoregressive) mode and the direct mode for transformers to solve arithmetic and linear equation solving problems. The positive results for CoT are further extended to dynamical programming problems. The paper also provides empirical validations.

All reviewers are strongly positive about the research direction and the contributions of the results, which I agree with. I believe the paper contains multiple results (lower bounds via circuit theory, concrete transformer constructions, the dynamical programming problem framework, and the experiments) that are worthy to be highlighted in the community and may inspire future research. Therefore, I recommend acceptance with oral presentation, and congratulate the authors for the nice work.